# Q-Learning with Fine-Grained Gap-Dependent Regret

**Haochen Zhang, Zhong Zheng & Lingzhou Xue**[*]
Department of Statistics, The Pennsylvania State University
State College, PA, 16802, USA
{hqz5340,zvz5337,lzxue}@psu.edu

## ABSTRACT

We study fine-grained gap-dependent regret bounds for model-free reinforcement learning in episodic tabular Markov Decision Processes. Existing model-free algorithms achieve minimax worst-case regret, but their gap-dependent bounds remain coarse and fail to fully capture the structure of suboptimality gaps. To address this limitation, we establish fine-grained gap-dependent regret guarantees for both UCB-based and non-UCB-based algorithms. In the UCB-based setting, we develop a novel analytical framework that explicitly separates the analysis of optimal and suboptimal state-action pairs, yielding the first fine-grained regret upper bound for UCB-Hoeffding (Jin et al., 2018). In the non-UCB-based setting, we revisit the only existing algorithm, AMB (Xu et al., 2021), and identify two issues in its design and analysis: improper truncation in the $Q$-updates and violation of the martingale difference condition in the concentration argument. To resolve these issues, we propose two refinements of AMB: the UCB-based ULCB-Hoeffding and the non-UCB-based Refined AMB. For ULCB-Hoeffding, we establish the same fine-grained regret bound as UCB-Hoeffding by applying our fine-grained framework, highlighting its broad applicability. For Refined AMB, we derive a rigorous fine-grained gap-dependent regret bound in the non-UCB setting and demonstrate consistent empirical improvements over the original AMB.

## 1 INTRODUCTION

Reinforcement Learning (RL) (Sutton & Barto, 2018) is a sequential decision-making framework where an agent maximizes cumulative rewards through repeated interactions with the environment. RL algorithms are typically categorized as model-based or model-free methods. Model-free approaches directly learn value functions to optimize policies and are widely used in practice due to their simple implementation (Jin et al., 2018) and low memory requirements, which scale linearly with the number of states. In contrast, model-based methods require quadratic memory costs.

In this paper, we focus on model-free RL for episodic tabular Markov Decision Processes (MDPs) with inhomogeneous transition kernels. Specifically, we consider an episodic tabular MDP with $S$ states, $A$ actions, and $H$ steps per episode. For such MDPs, the minimax regret lower bound over $K$ episodes is $\Omega(\sqrt{H^2 SAT})$, where $T = KH$ is the total number of steps (Jin et al., 2018).

Many model-free algorithms achieve $\sqrt{T}$-type regret bounds (Jin et al., 2018; Zhang et al., 2020; Li et al., 2021; Xu et al., 2021; Zhang et al., 2025b), with two (Zhang et al., 2020; Li et al., 2021) matching the minimax bound up to logarithmic factors. Except for AMB (Xu et al., 2021), which uses a novel multi-step bootstrapping technique, all these methods rely on the Upper Confidence Bound (UCB) approach to drive exploration via optimistic value estimates.

In practice, RL algorithms often outperform their worst-case guarantees when positive suboptimality gaps exist, meaning the best action at each state is better than the others by some margin. In the model-free setting, for UCB-based algorithms, Yang et al. (2021) proved the first gap-dependent regret bound for UCB-Hoeffding (Jin et al., 2018), of order $\tilde{O}(H^6 SA/\Delta_{\min})$, where $\tilde{O}$ hides logarithmic factors and $\Delta_{\min}$ is the smallest positive suboptimality gap $\Delta_h(s,a)$ over all state-action-step

---

[*]Zhong Zheng and Lingzhou Xue are co-corresponding authors.

triples $(s, a, h)$. Later, Zheng et al. (2025b) improved the dependence on $H$ for UCB-Advantage (Zhang et al., 2020) and Q-EarlySettled-Advantage (Li et al., 2021). However, these results rely on a coarse-grained term $SA/\Delta_{\min}$ instead of the fine-grained $\Delta_h(s, a)$, limiting their tightness.

The only model-free, non-UCB-based algorithm, AMB, achieved the first fine-grained regret upper bound by incorporating two key components: Upper and Lower Confidence Bounds (ULCB) and multi-step bootstrapping. In particular, ULCB leverages both UCB and Lower Confidence Bound (LCB) techniques to select actions by maximizing the width of the confidence interval, and multi-step bootstrapping updates $Q$-estimates with rewards of multiple steps from settled optimal actions. However, as detailed in Section 4, the multi-step bootstrapping procedure encounters two issues in its algorithm design and analysis. Algorithmically, the improper truncation in the multi-step bootstrapping update (see lines 13-14 in Algorithm 1 of Xu et al. (2021)) breaks the key link between the $Q$-estimates and historical $V$-estimates (see their Equation (A.5)) that is essential for the analysis. Theoretically, the concentration inequalities are incorrectly applied by centering the estimators induced by multi-step bootstrapping on their expectations rather than on their conditional expectations (see their Equation (4.2) and Lemma 4.1), violating the required martingale difference conditions. These issues cast doubt on whether a rigorous fine-grained gap-dependent regret bound can be established for non-UCB-based AMB algorithms.

In contrast, recent model-based works (Simchowitz & Jamieson, 2019; Dann et al., 2021; Chen et al., 2025) have achieved fine-grained gap-dependent regret bounds of the following form:

$$\tilde{O}\left(\left(\sum_{h=1}^{H} \sum_{\Delta_h(s,a)>0} \frac{1}{\Delta_h(s,a)} + \frac{|Z_{\text{opt}}|}{\Delta_{\min}} + SA\right)\text{poly}(H)\right), \tag{1}$$

where $|Z_{\text{opt}}|$ denotes the number of optimal $(s, a, h)$ triples. These results incorporate individual suboptimality gaps $\Delta_h(s, a)$ and significantly reduce reliance on the global factor $1/\Delta_{\min}$. This progress naturally leads to the following open question:

*Can we establish rigorous fine-grained gap-dependent regret upper bounds for model-free RL with individual suboptimality gaps $\Delta_h(s, a)$ and improved dependence on $1/\Delta_{\min}$?*

Answering this question is challenging. **For UCB-based algorithms**, establishing fine-grained gap-dependent regret requires novel techniques, particularly in bounding the cumulative weighted estimation error of $Q$-estimates. Existing works (Yang et al., 2021; Zheng et al., 2025b) treat all state-action pairs uniformly in the analysis. However, it is insufficient for deriving fine-grained results, as optimal and suboptimal pairs exhibit significantly different visitation patterns: suboptimal pairs are typically visited only $\hat{O}(\log T)$ times (Zhang et al., 2025a), where $\hat{O}$ captures only the dependence on $T$. Ignoring this imbalance leads to loose bounds and an overly conservative dependence on $1/\Delta_{\min}$. **Regarding the non-UCB-based algorithm AMB**, it remains unclear whether the two estimators induced by multi-step bootstrapping jointly form an unbiased estimate of the optimal $Q$-function due to the randomness of the bootstrapping step. This property is crucial for the concentration analysis used to establish the optimism of AMB, yet it was not shown in Xu et al. (2021).

In this paper, we give an affirmative answer to the open question above by establishing **the first fine-grained gap-dependent regret upper bound for UCB-based algorithms**, along with rigorous counterparts for non-UCB-based algorithms. Our main contributions are summarized below:

**A Novel Fine-Grained Analytical Framework for UCB-Based Algorithms.** We develop a novel framework that explicitly distinguishes the visitation frequencies of optimal and suboptimal state-action pairs. Using this framework, we establish the first fine-grained, gap-dependent regret bound for a popular UCB-based algorithm, namely UCB-Hoeffding (Jin et al., 2018). As shown in Section 5, UCB-Hoeffding demonstrates improved empirical performance compared to AMB.

**Two Refinements of the AMB Algorithm with Rigorous Fine-Grained Analysis.** In Section 4, we revisit the AMB algorithm and identify both algorithmic and analytical issues that undermine its theoretical guarantees. We then propose two refinements of the AMB algorithm:

- **UCB-Based Refinement.** ULCB-Hoeffding in Section 4.1 simplifies AMB by removing its problematic multi-step bootstrapping and retaining only the ULCB mechanism. Using our UCB-based framework, we show that ULCB-Hoeffding achieves a fine-grained regret bound, demonstrating

that algorithms relying solely on the ULCB principle can also achieve fine-grained guarantees. This further underscores the generality of our UCB-based fine-grained analytical framework.

- **Non-UCB-Based Refinement.** We also propose Refined AMB, a non-UCB-based refinement that incorporates both ULCB and multi-step bootstrapping techniques. It has the following improvement: (i) removes improper truncations in the $Q$-updates, (ii) rigorously proves that the estimators induced by multi-step bootstrapping form an unbiased estimate of the optimal $Q$-function, (iii) ensures the martingale difference condition holds, which justifies applying concentration inequalities to these estimators, and (iv) establishes tighter confidence bounds. These refinements allow us to rigorously prove the fine-grained regret upper bound for a non-UCB-based algorithm and yield enhanced empirical performance, as shown in Section 5.

**Technical Novelty.** Our work introduces the following key technical innovations: (a) We analyze each state-action pair separately at every step, enabling a fine-grained upper bound on the cumulative weighted estimation error of the $Q$-estimates (Lemmas 3.2 and 3.3). (b) We establish a recursive relationship for cumulative weighted visitation counts across steps, supporting an inductive argument to obtain a fine-grained upper bound (Lemma 3.4), from which the expected regret upper bound follows (Lemma 3.1). (c) We perform a state-specific decomposition of conditional expectations in the concentration analysis of Refined AMB, enabling a recursive argument and induction over steps to show that the sum of two multi-step bootstrapping estimators is unbiased (Theorem 4.3 and Appendix G.3). The first two innovations, (a) and (b), form the core of our fine-grained analytical framework, extending its applicability to a wider range of model-free RL algorithms, while (c) offers a general technique for analyzing algorithms with multi-step bootstrapping.

## 2 PRELIMINARIES

In this paper, for any $C \in \mathbb{N}_+$, we denote by $[C]$ the set $\{1, 2, \ldots, C\}$. We write $\mathbb{I}[x]$ for the indicator function, which takes the value one if the event $x$ is true, and zero otherwise. We also set $\iota = \log(2SAT/p)$ with failure probability $p \in (0, 1)$ throughout this paper.

**Tabular Episodic Markov Decision Process (MDP).** A tabular episodic MDP is denoted as $\mathcal{M} := (\mathcal{S}, \mathcal{A}, H, \mathbb{P}, r)$, where $\mathcal{S}$ is the set of states with $|\mathcal{S}| = S$, $\mathcal{A}$ is the set of actions with $|\mathcal{A}| = A$, $H$ is the number of steps in each episode, $\mathbb{P} := \{\mathbb{P}_h\}_{h=1}^H$ is the transition kernel so that $\mathbb{P}_h(\cdot \mid s, a)$ characterizes the distribution over the next state given the state-action pair $(s, a)$ at step $h$, and $r := \{r_h\}_{h=1}^H$ are the deterministic reward functions with $r_h(s, a) \in [0, 1]$.

In each episode, an initial state $s_1$ is selected arbitrarily by an adversary. Then, at each step $h \in [H]$, an agent observes a state $s_h \in \mathcal{S}$, picks an action $a_h \in \mathcal{A}$, receives the reward $r_h = r_h(s_h, a_h)$ and then transits to the next state $s_{h+1}$. The episode ends when an absorbing state $s_{H+1}$ is reached.

**Policies and Value Functions.** A policy $\pi$ is a collection of $H$ functions $\left\{\pi_h : \mathcal{S} \to \Delta^{\mathcal{A}}\right\}_{h=1}^H$, where $\Delta^{\mathcal{A}}$ is the set of probability distributions over $\mathcal{A}$. A policy is deterministic if for any $s \in \mathcal{S}$, $\pi_h(s)$ concentrates all the probability mass on an action $a \in \mathcal{A}$. In this case, we denote $\pi_h(s) = a$. Let $V_h^\pi : \mathcal{S} \to \mathbb{R}$ denote the state value function at step $h$ under policy $\pi$. Formally, $V_h^\pi(s) := \sum_{h'=h}^H \mathbb{E}_{(s_{h'}, a_{h'}) \sim (\mathbb{P}, \pi)} [r_{h'}(s_{h'}, a_{h'}) \mid s_h = s]$. We also use $Q_h^\pi : \mathcal{S} \times \mathcal{A} \to \mathbb{R}$ to denote the state-action value function at step $h$ under policy $\pi$, defined as $Q_h^\pi(s, a) := r_h(s, a) + \sum_{h'=h+1}^H \mathbb{E}_{(s_{h'}, a_{h'}) \sim (\mathbb{P}, \pi)} [r_{h'}(s_{h'}, a_{h'}) \mid s_h = s, a_h = a]$. Azar et al. (2017) proved that there always exists an optimal policy $\pi^\star$ that achieves the optimal value $V_h^\star(s) = \sup_\pi V_h^\pi(s) = V_h^{\pi^\star}(s)$ and $Q_h^\star(s, a) = \sup_\pi Q_h^\pi(s, a) = Q_h^{\pi^\star}(s, a)$ for all $(s, h) \in \mathcal{S} \times [H]$. For any $(s, a, h)$, the following Bellman Equation and the Bellman Optimality Equation hold, with $V_{H+1}^\pi(s) = 0 = V_{H+1}^\star(s) = 0$:

$$\begin{cases} V_h^\pi(s) = \mathbb{E}_{a' \sim \pi_h(s)}[Q_h^\pi(s, a')] \\ Q_h^\pi(s, a) = r_h(s, a) + \mathbb{P}_{s,a,h} V_{h+1}^\pi \end{cases} \text{ and } \begin{cases} V_h^\star(s) = \max_{a' \in \mathcal{A}} Q_h^\star(s, a') \\ Q_h^\star(s, a) = r_h(s, a) + \mathbb{P}_{s,a,h} V_{h+1}^\star. \end{cases} \quad (2)$$

For any algorithm over $K$ episodes, let $\pi^k$ be the policy used in the $k$-th episode, and $s_1^k$ be the corresponding initial state. The regret over $T = HK$ steps is $\text{Regret}(T) := \sum_{k=1}^K \left(V_1^\star - V_1^{\pi^k}\right)(s_1^k)$.

**Suboptimality Gap.** For a given MDP, we define the suboptimality gap as follows.

**Definition 2.1.** *For any $(s, a, h)$, the suboptimality gap is defined as $\Delta_h(s, a) := V_h^\star(s) - Q_h^\star(s, a)$.*

Equation (2) ensures that $\Delta_h(s, a) \geq 0$ for any $(s, a, h) \in \mathcal{S} \times \mathcal{A} \times [H]$. Accordingly, we define the minimum gap at each step $h$ as follows.

**Definition 2.2.** *Define* $\Delta_{\min,h} := \inf\{\Delta_h(s, a) : \Delta_h(s, a) > 0, (s, a) \in \mathcal{S} \times \mathcal{A}\}$ *as the **minimum gap at step** $h$. If the set $\{\Delta_h(s, a) : \Delta_h(s, a) > 0, (s, a) \in \mathcal{S} \times \mathcal{A}\}$ is empty, we set $\Delta_{\min,h} = \infty$.*

Most gap-dependent works (Simchowitz & Jamieson, 2019; Xu et al., 2020; Dann et al., 2021; Yang et al., 2021; Zhang et al., 2025a) define a **minimum gap** as $\Delta_{\min} := \inf\{\Delta_h(s, a) : \Delta_h(s, a) > 0, (s, a, h) \in \mathcal{S} \times \mathcal{A} \times [H]\}$. By definition, it is obvious that $\Delta_{\min,h} \geq \Delta_{\min}$ for all $h \in [H]$.

## 3 FINE-GRAINED REGRET UPPER BOUND FOR UCB-BASED ALGORITHMS

In this section, we present the first fine-grained, gap-dependent regret analysis for a UCB-based algorithm—UCB-Hoeffding (Jin et al., 2018), using our novel framework introduced in Section 3.2. To demonstrate the generality of our approach, we introduce a new UCB-based algorithm, ULCB-Hoeffding, in Section 4.1 and establish a fine-grained regret bound for it with the same framework.

### 3.1 THEORETICAL GUARANTEES FOR UCB-HOEFFDING

We first review UCB-Hoeffding in Algorithm 1. At the start of any episode $k$, it keeps an upper bound $Q_h^k$ on $Q_h^\star$ for each $(s, a, h)$, and selects actions greedily. The update of $Q_h^k$ uses the standard Bellman update with step size $\eta_t = (H+1)/(H+t)$ and a Hoeffding bonus $b_t$. For convenience, for any $N \in \mathbb{N}_+$ and $1 \leq i \leq N$, we additionally define $\eta_0^0 = 1, \eta_0^N = 0$ and $\eta_i^N = \eta_i \prod_{i'=i+1}^{N}(1 - \eta_{i'})$.

---

**Algorithm 1** UCB-Hoeffding

1: Initialize $Q_h^1(s, a) \leftarrow H$ and $N_h^1(s, a) \leftarrow 0$ for all $(s, a, h)$.
2: **for** episode $k = 1, \ldots, K$, after receiving $s_1^k$ and setting $V_{H+1}^k = 0$, **do**
3:     **for** step $h = 1, \ldots, H$ **do**
4:         Take action $a_h^k = \arg\max_{a'} Q_h^k(s_h^k, a')$, and observe $s_{h+1}^k$.
5:         $t = N_h^{k+1}(s_h^k, a_h^k) \leftarrow N_h^k(s_h^k, a_h^k) + 1; \ b_t \leftarrow 2\sqrt{H^3\iota/t}$.
6:         $Q_h^{k+1}(s_h^k, a_h^k) = (1 - \eta_t)Q_h^k(s_h^k, a_h^k) + \eta_t[r_h(s_h^k, a_h^k) + V_{h+1}^k(s_{h+1}^k) + b_t]$.
7:         $V_h^{k+1}(s_h^k) = \min\{H, \max_{a' \in \mathcal{A}} Q_h^{k+1}(s_h^k, a')\}$.
8:         $Q_h^{k+1}(s, a) = Q_h^k(s, a), V_h^{k+1}(s) = V_h^k(s), N_h^{k+1}(s, a) = N_h^k(s, a), \forall(s, a) \neq (s_h^k, a_h^k)$.
9:     **end for**
10: **end for**

---

Next, we present the fine-grained gap-dependent regret upper bound for UCB-Hoeffding.

**Theorem 3.1.** *For UCB-Hoeffding (Algorithm 1), the expected regret $\mathbb{E}[\text{Regret}(T)]$ is bounded by*

$$O\left(\sum_{h=1}^{H} \sum_{\Delta_h(s,a)>0} \frac{H^5 \log(SAT)}{\Delta_h(s, a)} + \sum_{h=1}^{H} \frac{H^3 \left(\sum_{t=h+1}^{H} \sqrt{|Z_{\text{opt},t}|}\right)^2 \log(SAT)}{\Delta_{\min,h}} + SAH^3\right). \quad (3)$$

*Here for any $h \in [H]$, $Z_{\text{opt},h} = \{(s, a) \in \mathcal{S} \times \mathcal{A} | \Delta_h(s, a) = 0\}$ with $S \leq |Z_{\text{opt},h}| \leq SA$.*

In the ideal case where the MDP contains only a single suboptimal state-action-step triple $(s, a, h)$ with $h = H$, our result exhibits a significantly improved dependence on the minimum gap, namely $\tilde{O}(H^5/\Delta_{\min})$, compared to the $\tilde{O}(H^6SA/\Delta_{\min})$ dependence in Yang et al. (2021). Even in the worst scenario where all suboptimality gaps satisfy $\Delta_h(s, a) = \Delta_{\min}$, our result degrades gracefully to match the result in Yang et al. (2021). These findings demonstrate that our result outperforms that of Yang et al. (2021) in all cases for the UCB-Hoeffding algorithm.

By applying the Cauchy–Schwarz inequality and noting that $\Delta_{\min,h} \geq \Delta_{\min}$ for all $h \in [H]$, we can derive the following weaker but simpler upper bound on the expected regret from Equation (3):

$$O\left(\sum_{h=1}^{H} \sum_{\Delta_h(s,a)>0} \frac{H^5 \log(SAT)}{\Delta_h(s, a)} + \frac{H^5 |Z_{\text{opt}}| \log(SAT)}{\Delta_{\min}} + SAH^3\right),$$

where $Z_{\text{opt}} = \{(s, a, h) \in \mathcal{S} \times \mathcal{A} \times [H] | \Delta_h(s, a) = 0\}$ is the set of optimal state-action-step triples.

**Remark:** The lower bound established in Simchowitz & Jamieson (2019) shows that any UCB-based algorithm, such as UCB-Hoeffding, must incur a gap-dependent expected regret of at least

$$\tilde{\Omega}\left(\sum_{h=1}^{H}\sum_{\Delta_h(s,a)>0}\frac{1}{\Delta_h(s,a)}+\frac{S}{\Delta_{\min}}\right).$$

Our result matches this lower bound up to polynomial factors in $H$ in the ideal scenario where $|Z_{\text{opt}}|$ is independent of $A$, such as in MDPs with a constant number of optimal actions per state.

Xu et al. (2021) also provides a lower bound $\tilde{\Omega}(|Z_{\text{mul}}|/\Delta_{\min})$ for all types of algorithms when $HS \leq |Z_{\text{mul}}| \leq \frac{HSA}{2}$. Here, for any $h \in [H]$,

$$Z_{\text{mul}} = \{(s,a,h) \in \mathcal{S} \times \mathcal{A} \times [H] \mid \Delta_h(s,a) = 0, \ |Z_{\text{opt},h}(s)| > 1\},$$

where $Z_{\text{opt},h}(s) = \{a \in \mathcal{A} \mid \Delta_h(s,a) = 0\}$. When $HS \leq |Z_{\text{mul}}| \leq \frac{HSA}{2}$, it holds that $|Z_{\text{opt}}| \leq 2|Z_{\text{mul}}|$, and therefore the lower bound can be expressed as $\tilde{\Omega}(|Z_{\text{opt}}|/\Delta_{\min})$. This demonstrates the tightness of the dependence on $|Z_{\text{opt}}|/\Delta_{\min}$ in the second term of our result.

## 3.2 A Novel Fine-Grained Analytical Framework

In this subsection, we introduce the novel analytical framework used to derive fine-grained, gap-dependent regret upper bounds. Full proofs are deferred to Appendix E. The key ideas of our fine-grained analytical framework are summarized below:

(1) We first establish Lemma 3.1, which upper-bounds the regret by the expectation of the cumulative weighted visitation counts $\sum_{h=1}^{H}\sum_{s,a}\Delta_h(s,a)N_h^{K+1}(s,a)$ and further relates this term to the cumulative weighted estimation errors $\sum_{k=1}^{K}\omega_h^k(Q_h^k - Q_h^\star)(s_h^k, a_h^k)$.

(2) We then bound the cumulative weighted estimation errors by establishing a recursive relationship between consecutive steps (Lemma 3.2) and propagating it to the final step $H$ (Lemma 3.3).

(3) Using Lemmas 3.2 and 3.3, we derive a recursive relation for the cumulative weighted visitation counts $\sum_{s,a}\Delta_h(s,a)N_h^{K+1}(s,a)$ across steps, which enables an inductive argument to derive a fine-grained upper bound and subsequently bound the expected regret via Lemma 3.1.

### 3.2.1 Bounding Expected Regret with Cumulative Weighted Estimation Error

We begin with Lemma 3.1 that connects expected regret to suboptimality gaps:

**Lemma 3.1.** *For the UCB-Hoeffding algorithm with $K$ episodes and total $T = HK$ steps, we have:*

$$\mathbb{E}\left[\text{Regret}(T)\right] = \mathbb{E}\left(\sum_{h=1}^{H}\sum_{s,a}\Delta_h(s,a)N_h^{K+1}(s,a)\right).$$

Lemma 3.1 holds universally for any learning algorithm, as shown in Lemma E.2. Therefore, bounding the expected regret reduces to controlling $\sum_{h=1}^{H}\sum_{s,a}\Delta_h(s,a)N_h^{K+1}(s,a)$, which can further be bounded by the cumulative estimation error $\sum_{h=1}^{H}\sum_{k=1}^{K}\left(Q_h^k - Q_h^\star\right)(s_h^k, a_h^k)$. In particular, for any step $h$ and episode $k$, with high probability, we have

$$\left(Q_h^k - Q_h^\star\right)(s_h^k, a_h^k) \geq V_h^k(s_h^k) - Q_h^\star(s_h^k, a_h^k) \geq V_h^\star(s_h^k) - Q_h^\star(s_h^k, a_h^k) = \Delta_h(s_h^k, a_h^k). \quad (4)$$

Here, the first inequality follows from line 7 of Algorithm 1, and the second holds due to the optimism property $V_h^k \geq V_h^\star$ and $Q_h^k \geq Q_h^\star$ of UCB-Hoeffding (see Lemma E.1). With Equation (4), prior works (Yang et al., 2021; Zheng et al., 2025b) focused on bounding the cumulative weighted estimation error $\sum_{k=1}^{K}\omega_h^k(Q_h^k - Q_h^\star)(s_h^k, a_h^k)$ and established the following type of upper bound:

$$\sum_{k=1}^{K}\omega_h^k\left(Q_h^k - Q_h^\star\right)(s_h^k, a_h^k) \leq O\left(\sum_{h=h'}^{H}\sqrt{H^3 SA\|\omega_h(\cdot,h')\|_\infty\|\omega_h(\cdot,h')\|_1}\iota + \sum_{h=h'}^{H}C(h')\right), \quad (5)$$

where $\{\omega_h^k\}_{k=1}^{K}$ is a non-negative weight sequence and $C(h')$ collects the remaining terms at step $h'$. The norms $\|\omega_h(\cdot,h')\|_\infty$ and $\|\omega_h(\cdot,h')\|_1$ at step $h'$ are defined in Equation (7) later.

Equation (5) is obtained by applying the Cauchy–Schwarz inequality to the cumulative weighted bonus $\sum_k \omega_h^k b_{N_h^k}$ over all state-action pairs at any step $h$. However, as shown in Lemma 4.1 of Zhang et al. (2025a), in the gap-dependent setting, suboptimal state-action pairs $(s, a)$ at any step $h$ with $Q_h^\star(s, a) < V_h^\star(s)$ are visited at most $\hat{O}(\log T)$ times, whereas optimal pairs can be visited infinitely often. Thus, uniform analysis of all state-action pairs leads to loose bounds.

### 3.2.2 Separate Analysis for Each State-Action Pair

To address the looseness of uniform analysis, we analyze the cumulative weighted estimation error for **each state-action pair at every step**, enabling tighter control.

For any given step $h$ and non-negative weight sequence $\{\omega_h^k\}_{k=1}^K$, we define the following weights for any $k' \in [K]$, $h \le h' < H$:

$$\omega_h(k', h) := \omega_h^{k'}; \omega_h(k', h' + 1) := \sum_{i=N_{h'}^{k'+1}(s_{h'}^{k'}, a_{h'}^{k'})}^{N_{h'}^{K+1}(s_{h'}^{k'}, a_{h'}^{k'}) - 1} \omega_h(k^{i+1}(s_{h'}^{k'}, a_{h'}^{k'}, h'), h') \eta_{N_{h'}^{k'+1}(s_{h'}^{k'}, a_{h'}^{k'})}^i \quad (6)$$

with the norms

$$\|\omega_h(\cdot, h')\|_\infty := \max_{k' \in [K]} \omega_h(k', h'), \quad \|\omega_h(\cdot, h')\|_1 := \sum_{k'=1}^K \omega_h(k', h'). \quad (7)$$

Here, $k^i(s, a, h)$ denotes the episode index of the $i$-th visit to $(s, a, h)$ and $N_h^k(s, a)$ denotes the number of visits to $(s, a, h)$ before episode $k$. The weight $\omega_h(k', h' + 1)$ captures the contribution of the term $(Q_{h'+1}^{k'} - Q_{h'+1}^\star)(s_{h'+1}^{k'}, a_{h'+1}^{k'})$ when recursively bounding the cumulative weighted estimation error from step $h'$ to $h' + 1$ as shown in the second conclusion of Lemma 3.2 later.

For each state-action pair $(s, a)$, we define the state-action specific weight at any step $h \le h' \le H$ as $\omega_h(k', h', s, a) := \omega_h(k', h') \cdot \mathbb{I}[(s_{h'}^{k'}, a_{h'}^{k'}) = (s, a)]$ with the corresponding norms given by

$$\|\omega_h(\cdot, h', s, a)\|_\infty := \max_{k' \in [K]} \omega_h(k', h', s, a), \quad \|\omega_h(\cdot, h', s, a)\|_1 := \sum_{k'=1}^K \omega_h(k', h', s, a).$$

Additionally, for any state-action pair $(s, a)$, we also define

$$\tilde{\omega}_h(k', h' + 1, s, a) = \omega_h(k', h' + 1)\mathbb{I}[(s_{h'}^{k'}, a_{h'}^{k'}) = (s, a)].$$

The weight $\tilde{\omega}_h(k', h' + 1, s, a)$ characterizes the weight of the term $(Q_{h'+1}^{k'} - Q_{h'+1}^\star)(s_{h'+1}^{k'}, a_{h'+1}^{k'})$ when bounding the cumulative weighted estimation error for each state-action pair $(s, a)$ at step $h'$ as shown in the first conclusion of Lemma 3.2 later.

We are now ready to present Lemma 3.2, which bounds the cumulative weighted estimation error for each state-action pair $(s, a)$ at any subsequent steps $h' \in [h, H]$. It is derived by recursively using the $Q$-update (line 6 of Algorithm 1). The detailed statement is given in Lemma E.3, followed by its proof. Here, we use the shorthand $(s, a)_h^k = (s_h^k, a_h^k)$.

**Lemma 3.2.** *For the UCB-Hoeffding algorithm, with probability at least $1 - p$, for any non-negative weight sequence $\{\omega_h^k\}_k$ at step $h$, the following two conclusions hold simultaneously for any $(s, a) \in \mathcal{S} \times \mathcal{A}$ and subsequent step $h' \in [h, H]$:*

$$\sum_{k=1}^K \omega_h(k, h', s, a)(Q_{h'}^k - Q_{h'}^\star)(s_{h'}^k, a_{h'}^k) \le \sum_{k=1}^K \tilde{\omega}_h(k', h' + 1, s, a)(Q_{h'+1}^{k'} - Q_{h'+1}^\star)(s, a)_{h'+1}^{k'}$$

$$+ \|\omega_h(\cdot, h')\|_\infty H + 16\sqrt{H^3 \|\omega_h(\cdot, h')\|_\infty \|\omega_h(\cdot, h', s, a)\|_1 \iota},$$

*and*

$$\sum_{k=1}^K \omega_h(k, h') \left( Q_{h'}^k - Q_{h'}^\star \right)(s_{h'}^k, a_{h'}^k) \le \sum_{k=1}^K \omega_h(k', h' + 1)(Q_{h'+1}^{k'} - Q_{h'+1}^\star)(s_{h'+1}^{k'}, a_{h'+1}^{k'})$$

$$+ \|\omega_h(\cdot, h')\|_\infty SAH + 16 \sum_{s,a} \sqrt{H^3 \|\omega_h(\cdot, h')\|_\infty \|\omega_h(\cdot, h', s, a)\|_1 \iota}.$$

Iteratively applying recursions over steps $h' = h, \ldots, H$ in the second conclusion of Lemma 3.2, and using the recursively defined weights $\omega_h(k, h')$, we obtain the following Lemma 3.3:

**Lemma 3.3.** *For UCB-Hoeffding, with probability at least $1 - p$, for any non-negative weights $\{\omega_h^k\}_k$ at step $h$, it holds simultaneously for any $(s, a) \in \mathcal{S} \times \mathcal{A}$ and subsequent step $h' \in [h, H]$:*

$$\sum_{k=1}^{K} \omega_h(k, h') \left(Q_{h'}^k - Q_{h'}^\star\right) (s_{h'}^k, a_{h'}^k)$$

$$\leq \sum_{h_1=h'}^{H} \|\omega_h(\cdot, h_1)\|_\infty SAH + 16 \sum_{h_1=h'}^{H} \sum_{s,a} \sqrt{H^3 \|\omega_h(\cdot, h_1)\|_\infty \|\omega_h(\cdot, h_1, s, a)\|_1 \iota}.$$

The formal statement is presented in Lemma E.4, followed by its proof. Unlike the upper bound derived from the uniform analysis in Equation (5), Lemma 3.3 retains the individual contributions $\sqrt{H^3 \|\omega_h(\cdot, h_1)\|_\infty \|\omega_h(\cdot, h_1, s, a)\|_1 \iota}$. This allows a tighter upper bound under the uneven visitations across different triples in the gap-dependent analysis.

### 3.2.3  INDUCTIVE ANALYSIS FOR CUMULATIVE WEIGHTED VISITATION COUNTS

We partition the state-action pairs at each step $h'$ into two subsets: $Z_{\text{opt},h'}$ containing optimal state-action pairs, where $\Delta_{h'}(s, a) = 0$, and $Z_{\text{sub},h'} = \{(s, a)|\Delta_{h'}(s, a) > 0\}$ containing suboptimal state-action pairs. Then for any given step $h$, when Equation (4) holds, we set the weight as:

$$\omega_h^k := \mathbb{I}\left[(Q_h^k - Q_h^\star)(s_h^k, a_h^k) \geq \Delta_h(s_h^k, a_h^k), (s_h^k, a_h^k) \in Z_{\text{sub},h}\right] = \mathbb{I}\left[(s_h^k, a_h^k) \in Z_{\text{sub},h}\right] \leq 1.$$

The second equality follows directly from Equation (4). Using this choice, applying the first conclusion in Lemma 3.2 with $h' = h$, the bound $\|\omega_h(\cdot, h)\|_\infty \leq 1$, and the fact that $\|\omega_h(\cdot, h, s, a)\|_1 \leq N_h^{K+1}(s, a)$, we obtain the following inequalities for any state-action pair $(s, a) \in Z_{\text{sub},h}$:

$$\Delta_h(s, a) N_h^{K+1}(s, a) \leq \sum_{k=1}^{K} \omega_h^k (Q_h^k - Q_h^\star)(s_h^k, a_h^k) \mathbb{I}[(s_h^k, a_h^k) = (s, a)]$$

$$\leq \sum_{k'=1}^{K} \tilde{\omega}_h(k', h+1, s, a)(Q_{h+1}^{k'} - Q_{h+1}^\star)(s_{h+1}^{k'}, a_{h+1}^{k'}) + H + 16\sqrt{H^3 N_h^{K+1}(s, a) \iota}. \quad (8)$$

Solving this inequality for $\Delta_h(s, a) N_h^{K+1}(s, a)$ with $(s, a) \in Z_{\text{sub},h}$ and $\Delta_h(s, a) > 0$, we reach:

$$\Delta_h(s, a) N_h^{K+1}(s, a) \leq \frac{256 H^3 \iota}{\Delta_h(s, a)} + 2H + 2 \sum_{k'=1}^{K} \tilde{\omega}_h(k', h+1, s, a)(Q_{h+1}^{k'} - Q_{h+1}^\star)(s_{h+1}^{k'}, a_{h+1}^{k'}).$$

Define $\sum_{\text{sub}}$ as the summation over all suboptimal state-action pairs $(s, a) \in Z_{\text{sub},h}$. Summing the inequality above over all $(s, a) \in Z_{\text{sub},h}$, and noting that $\Delta_h(s, a) = 0$ for $(s, a) \notin Z_{\text{sub},h}$,

$$\sum_{\text{sub}} \tilde{\omega}_h(k', h+1, s, a) \leq \sum_{s,a} \tilde{\omega}_h(k', h+1, s, a) = \omega_h(k', h+1),$$

together with the optimism property $Q_{h+1}^k \geq Q_{h+1}^\star$ by Lemma E.1, we obtain:

$$\sum_{s,a} \Delta_h(s, a) N_h^{K+1}(s, a) \leq \sum_{\text{sub}} \frac{256 H^3 \iota}{\Delta_h(s, a)} + 2SAH + 2 \sum_{k'=1}^{K} \omega_h(k', h+1)(Q_{h+1}^{k'} - Q_{h+1}^\star)(s, a)_{h+1}^{k'}.$$

Applying Lemma 3.3 with $h' = h + 1$ to the last term in the equation above and defining $C'(h) = O(H^2 SA + \sum_{\text{sub}} H^3 \iota / \Delta_h(s, a))$ to collect the remaining terms, we have

$$\sum_{s,a} \Delta_h(s, a) N_h^{K+1}(s, a) \leq C'(h) + 32 \sum_{h'=h+1}^{H} \sum_{s,a} \sqrt{H^3 \|\omega_h(\cdot, h')\|_\infty \|\omega_h(\cdot, h', s, a)\|_1 \iota}. \quad (9)$$

To bound the last term in Equation (9), we apply the Cauchy–Schwarz inequality by distinguishing between optimal and suboptimal state-action pairs. Specifically, we apply the inequality **separately** to the optimal state-action pairs in $Z_{\text{opt},h'}$ for each step $h'$, and **collectively** to all suboptimal state-action pairs across steps $h < h' \leq H$. This separation enables a sharper bound of

$$O\left(\sum_{h'=h+1}^{H} \sqrt{H^3 |Z_{\text{opt},h'}| \|\omega_h(\cdot, h)\|_1 \iota} + \sqrt{H^3 \iota \sum_{\text{sub},h'} \frac{1}{\Delta_{h'}(s, a)} \sum_{\text{sub},h'} \Delta_{h'}(s, a) N_{h'}^{K+1}(s, a)}\right) \quad (10)$$

where the shorthand $\sum_{\text{sub},h'}$ denotes the summation over all $(s,a) \in Z_{\text{sub},h'}$ for $h < h' \leq H$. This result also relies on the following three properties proved in Equations (27) to (29) of Lemma E.5:

$$\|\omega_h(\cdot, h')\|_\infty \leq 3, \ \sum_{s,a} \|\omega_h(\cdot, h', s, a)\|_1 \leq \|\omega_h(\cdot, h)\|_1, \ \|\omega_h(\cdot, h', s, a)\|_1 \leq O\left(N_{h'}^{K+1}(s,a)\right).$$

Plugging the bound from Equation (10) into Equation (9) yields a recursive relation between $\sum_{s,a} \Delta_h(s,a) N_h^{K+1}(s,a)$ at step $h$ and future steps. Applying induction from $H$ down to 1, we obtain a fine-grained upper bound on the cumulative weighted visitation $\sum_{s,a} \Delta_h(s,a) N_h^{K+1}(s,a)$.

**Lemma 3.4.** *For UCB-Hoeffding algorithm and a sufficiently large constant $c_1 > 0$, with probability at least $1 - p$, it holds simultaneously for any $h \in [H]$ that:*

$$\sum_{s,a} \frac{\Delta_h(s,a) N_h^{K+1}(s,a)}{c_1} \leq SAH^2 + \sum_{h'=h}^{H} \sum_{\Delta_{h'}(s,a)>0} \frac{H^4 \iota}{\Delta_{h'}(s,a)} + \frac{H^3 \left(\sum_{t=h+1}^{H} \sqrt{|Z_{\text{opt},t}|}\right)^2 \iota}{\Delta_{\min,h}}$$

$$+ \sum_{h'=h+1}^{H} \frac{H^2 \left(\sum_{t=h'+1}^{H} \sqrt{|Z_{\text{opt},t}|}\right)^2 \iota}{\Delta_{\min,h'}}.$$

The full proof is provided in Lemma E.5. By combining this result with Lemma 3.1, we complete the proof of Theorem 3.1, establishing the desired fine-grained, gap-dependent regret upper bound.

## 4 FINE-GRAINED GAP-DEPENDENT REGRET UPPER BOUND FOR AMB

The AMB algorithm (Xu et al., 2021) was proposed to establish a fine-grained, gap-dependent regret bound. However, we identify issues in both its algorithmic design and theoretical analysis that prevent it from achieving valid fine-grained guarantees. We first summarize these issues below.

**Improper Truncation of $Q$-Estimates in Algorithm Design.** AMB maintains upper and lower estimates on the optimal $Q$-value functions, denoted by $\overline{Q}$ and $\underline{Q}$, respectively. However, during multi-step bootstrapping updates of these estimates, it applies truncations at $H$ and 0 (see lines 13-14 in Algorithm 3). This design breaks the recursive structure linking $Q$-estimates to historical $V$-estimates. In particular, it invalidates their Equation (A.5), which is essential for establishing the theoretical guarantee on the optimism and pessimism of $Q$-estimates $\overline{Q}$ and $\underline{Q}$, respectively.

**Violation of Martingale Difference Conditions in Concentration Analysis.** AMB uses multi-step bootstrapping and constructs $Q$-estimates by decomposing the $Q$-function into two parts: rewards accumulated along states with determined optimal actions, and those collected from the first state with undetermined optimal actions. When proving optimism and pessimism of the $Q$-estimates (see their Lemma 4.2), Xu et al. (2021) attempt to bound the deviation between the $Q$-estimates and $Q^\star$ using Azuma–Hoeffding inequalities. However, when analyzing the two estimators arising from the $Q$-function decomposition (see their Equation (4.2) and Lemma 4.1), each term is improperly centered around an incorrect expectation because the randomness of the bootstrapping step is ignored, thereby violating the martingale-difference condition required for the Azuma–Hoeffding inequality.

These issues compromise the claimed optimism and pessimism guarantees for the $Q$-estimates and invalidate the stated fine-grained gap-dependent regret upper bound in Xu et al. (2021). A detailed analysis is provided in Appendix G.1. To address these issues, we propose two refinements: a UCB-based refinement, ULCB-Hoeffding, and a non-UCB-based refinement, Refined AMB.

### 4.1 UCB-BASED REFINEMENT: ULCB-HOEFFDING

In this subsection, we introduce the UCB-based refinement of AMB, ULCB-Hoeffding. It also achieves a fine-grained regret upper bound and demonstrates improved empirical performance over AMB. Importantly, our fine-grained analytical framework presented in Section 3.2 naturally extends to this variant, demonstrating the framework's flexibility and generality.

The ULCB-Hoeffding algorithm is presented in Algorithm 2. At the start of each episode $k$, ULCB-Hoeffding maintains upper and lower bounds, $\overline{Q}_h^k(s,a)$ and $\underline{Q}_h^k(s,a)$, of the optimal value func-

tion $Q_h^\star(s,a)$ for any $(s,a,h)$. It then constructs a candidate action set $A_h^k(s)$ by eliminating actions that are considered suboptimal (see line 14 in Algorithm 2). Specifically, if action $a$ satisfies $\overline{Q}_h^{k+1}(s,a) < \underline{V}_h^{k+1}(s)$, then by line 9 in Algorithm 2, there exists another action $a'$ such that $Q_h^\star(s,a) \leq \overline{Q}_h^{k+1}(s,a) < \underline{V}_h^{k+1}(s) \leq \underline{Q}_h^{k+1}(s,a') \leq Q_h^\star(s,a')$, which confirms that the action $a$ is suboptimal. At the end of episode $k$, the new policy $\pi_h^{k+1}(s)$ is chosen to maximize the width of the confidence interval $(\overline{Q}_h^{k+1} - \underline{Q}_h^{k+1})(s,a)$, which measures the uncertainty in the $Q$-estimates.

---

**Algorithm 2** ULCB-Hoeffding

---

1: **Initialize:** Set the failure probability $p \in (0,1)$, $\overline{Q}_h^1(s,a) = \overline{V}_h^1(s) \leftarrow H$, $\underline{Q}_h^1(s,a) = \underline{V}_h^1(s) = N_h^1(s,a) \leftarrow 0$ and $A_h^1(s) = \mathcal{A}$ for any $(s,a,h) \in \mathcal{S} \times \mathcal{A} \times [H]$.

2: **for** episode $k = 1, \ldots, K$, after receiving $s_1^k$ and setting $\overline{V}_{H+1}^k = \underline{V}_{H+1}^k(s) = 0$, **do**

3:     **for** step $h = 1, \ldots, H$ **do**

4:         Choose $a_h^k \triangleq \begin{cases} \arg\max_{a \in A_h^k(s)}(\overline{Q}_h^k - \underline{Q}_h^k)(s_h^k, a), & \text{if } |A_h^k(s_h^k)| > 1 \\ \text{the only element in } A_h^k(s_h^k), & \text{if } |A_h^k(s_h^k)| = 1 \end{cases}$ and get $s_{h+1}^k$.

5:         Set $t = N_h^{k+1}(s_h^k, a_h^k) \leftarrow N_h^k(s_h^k, a_h^k) + 1$ and the bonus $b_t = 2\sqrt{H^3\iota/t}$, and update:

6:         $\overline{Q}_h^{k+1}(s_h^k, a_h^k) = (1 - \eta_t)\overline{Q}_h^k(s_h^k, a_h^k) + \eta_t \left[ r_h(s_h^k, a_h^k) + \overline{V}_{h+1}^k(s_{h+1}^k) + b_t \right]$.

7:         $\underline{Q}_h^{k+1}(s_h^k, a_h^k) = (1 - \eta_t)\underline{Q}_h^k(s_h^k, a_h^k) + \eta_t \left[ r_h(s_h^k, a_h^k) + \underline{V}_{h+1}^k(s_{h+1}^k) - b_t \right]$.

8:         $\overline{V}_h^{k+1}(s_h^k) = \min \left\{ H, \max_{a \in A_h^k(s_h^k)} \overline{Q}_h^{k+1}(s_h^k, a) \right\}$.

9:         $\underline{V}_h^{k+1}(s_h^k) = \max \left\{ 0, \max_{a \in A_h^k(s_h^k)} \underline{Q}_h^{k+1}(s_h^k, a) \right\}$.

10:     **end for**

11:     **for** $(s,a,h) \in \mathcal{S} \times A \times [H] \setminus \{(s_h^k, a_h^k)\}_{h=1}^H$ **do**

12:         $\left(\overline{Q}_h^{k+1}, \underline{Q}_h^{k+1}, N_h^{k+1}\right)(s,a) = \left(\overline{Q}_h^k, \underline{Q}_h^k, N_h^k\right)(s,a), \left(\overline{V}_h^{k+1}, \overline{V}_h^{k+1}\right)(s) = \left(\overline{V}_h^k, \underline{V}_h^k\right)(s)$.

13:     **end for**

14:     $\forall (s,h) \in \mathcal{S} \times [H]$, update $A_h^{k+1}(s) = \{a \in A_h^k(s) : \overline{Q}_h^{k+1}(s,a) \geq \underline{V}_h^{k+1}(s)\}$.

15: **end for**

---

The main difference between ULCB-Hoeffding and AMB lies in the updates of $Q$-estimates. ULCB-Hoeffding uses the standard Bellman update (lines 6–7 of Algorithm 2), similar to UCB-Hoeffding (line 6 of Algorithm 1), which is essential to prove a fine-grained regret upper bound. In contrast, AMB uses a multi-step bootstrapping update, which will be detailed in Section 4 and Appendix G.1.

We now present both worst-case and gap-dependent regret upper bounds for ULCB-Hoeffding.

**Theorem 4.1.** *For any $p \in (0,1)$, let $\iota = \log(2SAT/p)$. Then with probability at least $1 - p$, ULCB-Hoeffding (Algorithm 2) satisfies $\mathrm{Regret}(T) \leq O(\sqrt{H^4SAT\iota})$.*

The proof of Theorem 4.1 is provided in Appendix F.2. The resulting bound matches the $\sqrt{T}$-type worst-case regret guarantee of UCB-Hoeffding (Jin et al., 2018).

**Theorem 4.2.** *For ULCB-Hoeffding (Algorithm 2), the expected regret is upper bounded by (3).*

The proof of Theorem 4.2 is given in Appendix F.3 using the fine-grained analytical framework developed in Section 3.2. ULCB-Hoeffding attains the same fine-grained gap-dependent regret upper bound as UCB-Hoeffding. As discussed in Section 3.1, the guarantee in Equation (3) matches the lower bound established by Simchowitz & Jamieson (2019) for UCB-based algorithms, with a tight dependence on $|Z_{\mathrm{opt}}|/\Delta_{\min}$ that also aligns with the lower bound in Xu et al. (2021), up to polynomial factors in $H$.

### 4.2 Non-UCB-Based Refinement: Refined AMB

Since multi-step bootstrapping can improve the regret's dependence on $1/\Delta_{\min}$, we propose a non-UCB-based refinement, Refined AMB, which incorporates both ULCB and multi-step bootstrapping. The following key modifications are introduced to address the two issues of AMB:

**(a) Revising Update Rules.** We remove the truncations in the updates of $Q$-estimates and instead apply them to the corresponding $V$-estimates. This preserves the crucial recursive structure linking $Q$-estimates to historical $V$-estimates used in the theoretical analysis.

**(b) Establishing Unbiasedness of Multi-Step Bootstrapping.** We rigorously prove that the estimators from multi-step bootstrapping form an unbiased estimate of the optimal value function $Q^\star$.

**(c) Ensuring Martingale Difference Condition.** We ensure the validity of Azuma–Hoeffding inequalities by centering the multi-step bootstrapping estimators around true conditional expectations.

**(d) Tightening Confidence Bounds.** By jointly analyzing the concentration of both estimators, we tighten the confidence interval and halve the bonus, leading to improved empirical performance.

These modifications not only ensure theoretical validity but also yield improved empirical performance. The refined algorithm is presented in Algorithms 4 and 5 of Appendix G.2. We further establish the following optimism and pessimism properties for its $Q$-estimates.

**Theorem 4.3** (Informal). *For the Refined AMB algorithm, with high probability, $\overline{Q}_h^k(s, a) \geq Q_h^\star(s, a) \geq \underline{Q}_h^k(s, a)$ holds simultaneously for all $(s, a, h, k) \in \mathcal{S} \times \mathcal{A} \times [H] \times [K]$.*

The formal statement is given in Theorem G.1, with its proof in Appendix G.3. Based on this result, we can follow the remaining analysis of Xu et al. (2021) to prove the following regret upper bound:

$$O\left( \sum_{h=1}^{H} \sum_{\Delta_h(s,a)>0} \frac{H^5 \log(SAT)}{\Delta_h(s, a)} + \frac{H^5 |Z_{\text{mul}}| \log(SAT)}{\Delta_{\min}} \right). \tag{11}$$

## 5 NUMERICAL EXPERIMENTS

In this section, we present numerical experiments[1] conducted in synthetic environments, evaluating four algorithms: AMB, Refined AMB, UCB-Hoeffding, and ULCB-Hoeffding. We consider four **experiment scales** with $(H, S, A, K) = (2, 3, 3, 10^5), (5, 5, 5, 6 \times 10^5), (7, 8, 6, 5 \times 10^6)$, and $(10, 15, 10, 2 \times 10^7)$. For each $(s, a, h)$, rewards $r_h(s, a)$ are sampled independently from the uniform distribution over $[0, 1]$, and transition kernels $\mathbb{P}_h(\cdot \mid s, a)$ are drawn uniformly from the $S$-dimensional probability simplex. The initial state of each episode is sampled uniformly at random from the state space.

We also set $\iota = 1$ and the bonus coefficient $c = 1$ for UCB-Hoeffding, ULCB-Hoeffding, and Refined AMB, and $c = 2$ for AMB. This is because AMB applies concentration inequalities separately to the two estimators induced by multi-step bootstrapping. In contrast, all other algorithms, including the Refined AMB that combines the concentration analysis for multi-step bootstrapping, apply the concentration inequality only once, resulting in a bonus term with half the constant.

To report uncertainty, we collect 10 sample trajectories per algorithm under the same MDP instance. In Figure 1 of Appendix B, we plot $\text{Regret}(T) / \log(K + 1)$ versus the number of episodes $K$. Solid lines indicate the median regret, and shaded regions represent the 10th-90th percentile intervals.

The results show that ULCB-Hoeffding and Refined AMB achieve comparable performance, both outperforming the original AMB, while UCB-Hoeffding performs the best overall. In all settings, the regret curves for all algorithms except AMB flatten as $K$ increases, indicating logarithmic growth in regret, which is consistent with the fine-grained theoretical guarantees.

## 6 CONCLUSION

This work establishes rigorous fine-grained gap-dependent regret bounds for model-free RL in episodic tabular MDPs. In the UCB-based setting, we introduce a novel analytical framework that enables the first fine-grained regret analysis of UCB-Hoeffding. For the non-UCB-based AMB algorithm, we propose two refinements, ULCB-Hoeffding and Refined AMB, that address its algorithmic and analytical issues. In particular, for Refined AMB, we establish a rigorous fine-grained regret bound in the non-UCB setting, while also demonstrating improved empirical performance.

---

[1] All experiments were conducted on a desktop equipped with an Intel Core i7-14700F processor and completed within 12 hours. The code is included in the supplementary materials.

## ACKNOWLEDGMENT

The work of Haochen Zhang, Zhong Zheng, and Lingzhou Xue was supported by the U.S. National Science Foundation under the grants DMS-1953189 and CCF-2007823 and by the U.S. National Institutes of Health under the grant 1R01GM152812.

## ETHICS STATEMENT

This work is primarily theoretical and does not involve human subjects, personal data, or any experiments requiring ethical approval. We have followed all guidelines outlined in the ICLR Code of Ethics, ensuring transparency, integrity, and fairness throughout the research process. There are no foreseeable ethical concerns or potential harms related to this study.

## REPRODUCIBILITY STATEMENT

To ensure reproducibility, we provide detailed theoretical analyses, including a clearly defined tabular MDP framework and assumptions in Section 2, as well as proof sketch outlines in Sections 3 and 4. Full proofs are included in the appendix. For the empirical results included in Section 5, all experiments were conducted on a desktop equipped with an Intel Core i7-14700F processor and completed within 12 hours. The complete source code is provided in the supplementary materials to support independent verification.

## USE OF LARGE LANGUAGE MODELS

In this work, the use of large language models was strictly limited to text polishing and language refinement. All core scientific ideas, problem formulation, methodology design, and experimental planning were independently developed by the authors.

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

In the appendix, Appendix A reviews related work, and Appendix B presents the experimental results. Appendix C presents a comparison of results and techniques with model-based works. Appendix D introduces several lemmas that support our proofs. Appendix E establishes the fine-grained gap-dependent regret upper bound for the UCB-Hoeffding algorithm, marking the first such result for a UCB-based method. Appendix F provides proofs for both the worst-case and fine-grained gap-dependent regret bounds of the ULCB-Hoeffding algorithm. Finally, Appendix G offers a detailed analysis of both algorithmic and technical issues of the original AMB algorithm and presents a proof of the fine-grained regret bound for our Refined AMB algorithm.

## A  RELATED WORK

**Online RL for Tabular Episodic MDPs with Worst-Case Regret.** There are mainly two types of algorithms for reinforcement learning: model-based and model-free algorithms. Model-based algorithms learn a model from past experience and make decisions based on this model, while model-free algorithms only maintain a group of value functions and take the induced optimal actions. Due to these differences, model-free algorithms are usually more space-efficient and time-efficient compared to model-based algorithms. However, model-based algorithms may achieve better learning performance by leveraging the learned model.

Next, we discuss the literature on model-based and model-free algorithms for finite-horizon tabular MDPs with worst-case regret. Auer et al. (2008), Agrawal & Jia (2017), Azar et al. (2017), Kakade et al. (2018), Agarwal et al. (2020), Dann et al. (2019), Zanette & Brunskill (2019),Zhang et al. (2021), Zhou et al. (2023) and Zhang et al. (2024b) worked on model-based algorithms. Notably, Zhang et al. (2024b) provided an algorithm that achieves a regret of $\tilde{O}(\min\{\sqrt{SAH^2T}, T\})$, which matches the information lower bound. Jin et al. (2018), Zhang et al. (2025b), Zhang et al. (2020), Li et al. (2021) and Ménard et al. (2021) work on model-free algorithms. The latter three have introduced algorithms that achieve minimax regret of $\tilde{O}(\sqrt{SAH^2T})$. There are also several works focusing on online federated RL settings, such as Zheng et al. (2024), Labbi et al. (2024), Zheng et al. (2025a), and Zhang et al. (2025b). Notably, the last three works all achieve minimax regret bounds up to logarithmic factors.

**Suboptimality Gap.** When there exists a strictly positive suboptimality gap, logarithmic regret becomes achievable. Early studies established asymptotic logarithmic regret bounds (Auer & Ortner, 2007; Tewari & Bartlett, 2008). More recently, non-asymptotic bounds have been developed (Jaksch et al., 2010; Ok et al., 2018; Simchowitz & Jamieson, 2019; He et al., 2021). Specifically, Jaksch et al. (2010) designed a model-based algorithm whose regret bound depends on the policy gap instead of the action gap studied in this paper. Ok et al. (2018) derived problem-specific logarithmic-type lower bounds for both structured and unstructured MDPs. Simchowitz & Jamieson (2019) extended the model-based algorithm proposed by Zanette & Brunskill (2019) and obtained logarithmic regret bounds. More recently, Chen et al. (2025) further improved model-based gap-dependent results. Logarithmic regret bounds have also been established in the linear function approximation setting (He et al., 2021; Papini et al., 2021; Zhang et al., 2024a; 2026). Nguyen-Tang et al. (2023) provided gap-dependent guarantees for offline RL with linear function approximation.

Specifically, for model-free algorithms, Yang et al. (2021) demonstrated that the UCB-Hoeffding algorithm proposed in Jin et al. (2018) achieves a gap-dependent regret bound of $\tilde{O}(H^6SA/\Delta_{\min})$. This result was later improved by Xu et al. (2021), who introduced the Adaptive Multi-step Bootstrap (AMB) algorithm to achieve tighter bounds. Furthermore, Zheng et al. (2025b) provided gap-dependent analyses for algorithms with reference-advantage decomposition (Zhang et al., 2022; Li et al., 2021; Zheng et al., 2025a). More recently, Zhang et al. (2025a) and Zhang et al. (2025b) extended gap-dependent analysis to federated $Q$-learning settings.

There are also some other works focusing on gap-dependent sample complexity bounds (Jonsson et al., 2020; Al Marjani & Proutiere, 2020; Al Marjani et al., 2021; Tirinzoni et al., 2022; Wagenmaker et al., 2022b; Wagenmaker & Jamieson, 2022; Wang et al., 2022; Tirinzoni et al., 2023).

**Other Problem-Dependent Performance.** In practice, RL algorithms often outperform what their worst-case performance guarantees would suggest. This motivates a recent line of works that investigate optimal performance in various problem-dependent settings (Fruit et al., 2018; Jin et al., 2020; Talebi & Maillard, 2018; Wagenmaker et al., 2022a; Zhao et al., 2023; Zhou et al., 2023).

## B    EXPERIMENTAL RESULTS

This section provides the four numerical plots for four experiment scales with $(H, S, A, K) = (2, 3, 3, 10^5), (5, 5, 5, 6 \times 10^5), (7, 8, 6, 5 \times 10^6)$, and $(10, 15, 10, 2 \times 10^7)$ in Section 5. The algorithms evaluated are AMB, represented by the blue curve; ULCB-Hoeffding, shown in purple; Refined AMB, depicted in green; and UCB-Hoeffding, indicated by the red curve.

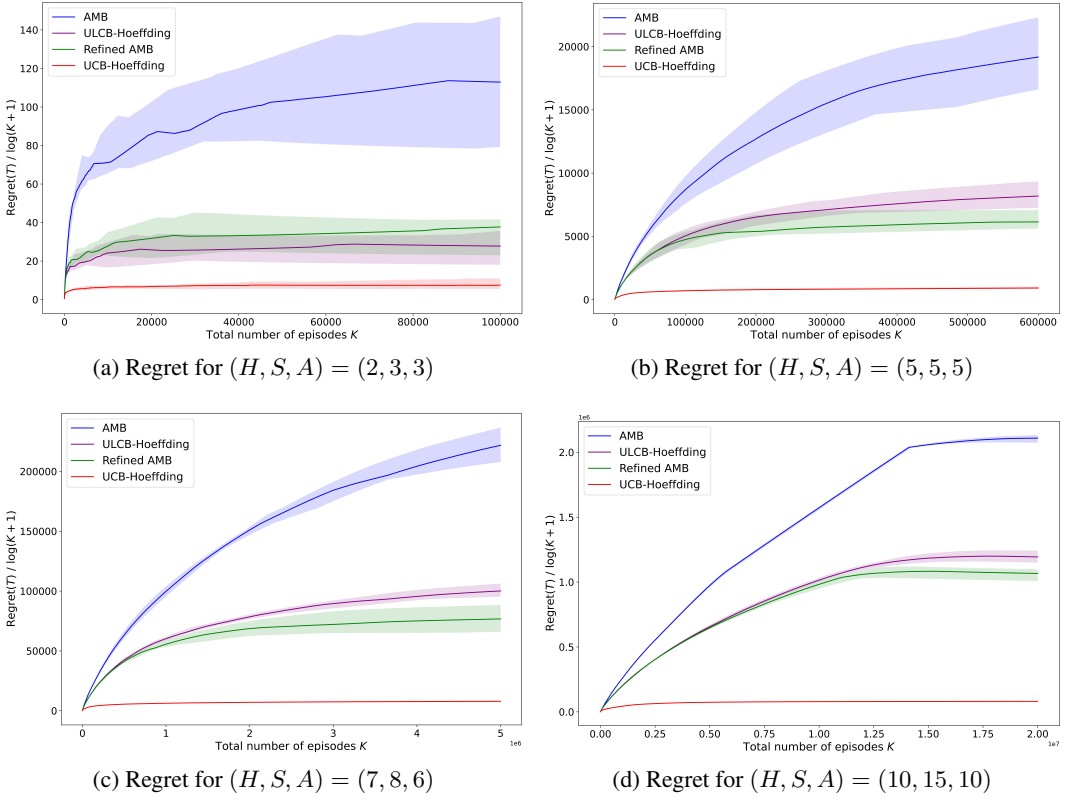

(a) Regret for $(H, S, A) = (2, 3, 3)$

(b) Regret for $(H, S, A) = (5, 5, 5)$

(c) Regret for $(H, S, A) = (7, 8, 6)$

(d) Regret for $(H, S, A) = (10, 15, 10)$

Figure 1: Regret Comparison of Different Algorithms.

Each plot displays the comparative performance of four distinct algorithms. In each plot, we collect 10 sample trajectories per algorithm under the same MDP instance and plot the results of $\text{Regret}(T)/\log(K + 1)$ versus the number of episodes $K$. Solid lines represent the median regret, while shaded regions show the range between the 10th and 90th percentiles.

## C    COMPARISON WITH MODEL-BASED WORKS

### C.1    REGRET UPPER BOUND COMPARISON

We first compare our fine-grained gap-dependent regret upper bound with those obtained in model-based works. Among them, the state-of-the-art result in Chen et al. (2025) establishes the following regret upper bound:

$$\bar{O}\left(\sum_{h=1}^{H} \sum_{\Delta_h(s,a)>0} \frac{H^2}{\Delta_h(s,a)} + \frac{H^2|Z_{\text{opt}}|}{\Delta_{\min}} + SAH^4 \max\{S, H\}\right),$$

where $\bar{O}$ hides variance-dependent and logarithmic factors.

Compared with our results in Equation (3) for UCB-Hoeffding and ULCB-Hoeffding, as well as in Equation (11) for Refined AMB, this result achieves an improved $H$-dependence in the gap-dependent terms, while maintaining a comparable $H$-dependence in the last term. However, the final

term scales as $SAH^4 \max\{S, H\}$, which introduces an additional $S^2$ dependence when $S \geq H$. Such a quadratic dependence on the state space dimensions can be prohibitive in large-scale MDPs.

This improved $H$-dependence stems from the strong performance of the model-based algorithm MVP (Zhang et al., 2024b). MVP achieves sharper regret guarantees by incorporating variance-based bonuses and explicitly estimating the transition kernel, thereby enabling more accurate value function estimation. However, as is typical for model-based methods, maintaining transition-kernel estimates incurs a memory cost of order $O(S^2AH)$, which is larger than the $O(SAH)$ memory requirement of model-free methods.

Our fine-grained analytical framework developed in Section 3.2 can potentially further improve the $H$-dependence in the fine-grained regret upper bound when applied to more advanced model-free RL algorithms, such as UCB-Bernstein (Jin et al., 2018), UCB-Advantage (Zhang et al., 2020), and Q-EarlySettled-Advantage (Li et al., 2021).

## C.2 TECHNICAL COMPARISON

All model-based works (Simchowitz & Jamieson, 2019; Dann et al., 2021; Chen et al., 2025) establish fine-grained gap-dependent regret upper bounds whose forms are similar to that in Equation (1). However, their analytical techniques cannot be directly applied to the model-free setting, particularly to the UCB-Hoeffding algorithm. To clarify this distinction, we next describe the common analytical approach underlying these model-based works.

All three works begin by using the following relationship to upper bound the expected regret:

$$\mathbb{E}(\text{Regret}(T)) \leq \mathbb{E}\left[\sum_{k=1}^{K}\sum_{h=1}^{H} E_h^k(s, a)\right],$$

where

$$E_h^k(s, a) = Q_h^k(s, a) - \left(r_h(s, a) + \mathbb{E}_{s' \sim \mathbb{P}_h(\cdot|s,a)}[V_{h+1}^k(s')]\right) \qquad (12)$$

denotes the surplus at $(s, a, h, k) \in \mathcal{S} \times \mathcal{A} \times [H] \times [K]$. The surplus is then bounded in terms of the suboptimality gap $\Delta_h(s, a)$ to derive a fine-grained regret bound.

For all these model-based works, the $Q$-estimates are updated as

$$Q_h^k(s, a) = r_h(s, a) + \mathbb{E}_{s' \sim \hat{\mathbb{P}}_h^k(\cdot|s,a)}[V_{h+1}^k(s')] + b_h^k(s, a),$$

where $\hat{\mathbb{P}}_h^k(\cdot \mid s, a)$ is the empirical estimate of the transition kernel $\mathbb{P}_h(\cdot \mid s, a)$ at the beginning of episode $k$. Substituting this update into Equation (12) gives

$$E_h^k(s, a) = b_h^k(s, a) + \mathbb{E}_{s' \sim \hat{\mathbb{P}}_h^k(\cdot|s,a)}[V_{h+1}^k(s')] - \mathbb{E}_{s' \sim \mathbb{P}_h(\cdot|s,a)}[V_{h+1}^k(s')]).$$

The two conditional expectation terms share the same input $V_{h+1}^k$, and $\hat{\mathbb{P}}_h^k(\cdot \mid s, a)$ is an unbiased estimator of the transition kernel $\mathbb{P}_h(\cdot \mid s, a)$. Therefore, their difference can be controlled using standard concentration inequalities and empirical process techniques.

The situation is fundamentally different for model-free methods, which do not maintain an empirical estimate of the transition kernel. In particular, for UCB-Hoeffding, the $Q$-estimates are updated as

$$Q_h^k(s, a) = r_h(s, a) + \sum_{i=1}^{N_h^k(s_h^k, a_h^k)} \eta_i^{N_h^k(s_h^k, a_h^k)} V_{h+1}^{k^i(s_h^k, a_h^k, h)}(s_{h+1}^{k^i}) + \beta_{N_h^k(s_h^k, a_h^k)},$$

where $\beta_n$ denotes the cumulative bonus for all $n \in \mathbb{N}_+$ defined in Lemma E.1 later. Accordingly,

$$E_h^k(s, a) = b_h^k(s, a) + \sum_{i=1}^{N_h^k(s_h^k, a_h^k)} \eta_i^{N_h^k(s_h^k, a_h^k)} \left(V_{h+1}^{k^i(s_h^k, a_h^k, h)}(s_{h+1}^{k^i}) - \mathbb{E}_{s' \sim \mathbb{P}_h(\cdot|s,a)}[V_{h+1}^k(s')]\right).$$

A key distinction is that the two terms inside the difference now involve different value function estimates: the empirical term uses historical estimates $V_{h+1}^{k^i}$, whereas the conditional expectation term uses the current estimate $V_{h+1}^k$. This temporal mismatch introduces additional bias, which prevents the analytical framework developed for model-based methods from being directly extended to the model-free setting.

## D  GENERAL LEMMAS

**Lemma D.1.** (Azuma-Hoeffding Inequality). *Suppose* $\{X_k\}_{k=0}^{\infty}$ *is a martingale and* $|X_k - X_{k-1}| \leq c_k$, $\forall k \in \mathbb{N}_+$, *almost surely. Then for any* $N \in \mathbb{N}_+$ *and* $\epsilon > 0$, *it holds that:*

$$\mathbb{P}\left(|X_N - X_0| \geq \epsilon\right) \leq 2 \exp\left(-\frac{\epsilon^2}{2\sum_{k=1}^{N} c_k^2}\right).$$

Based on the definition of $\eta_n^N$, it can be easily verified that $\sum_{n=1}^{N} \eta_n^N = \mathbb{I}[N > 0]$. We also have the following properties proved in Lemma 1 of Li et al. (2021).

**Lemma D.2.** *For any integer* $N > 0$, *the following properties hold:*

*(a) For any* $n \in \mathbb{N}_+$,

$$\sum_{N=n}^{\infty} \eta_n^N \leq 1 + \frac{1}{H}.$$

*(b) For any* $N \in \mathbb{N}_+$,

$$\sum_{n=1}^{N} (\eta_n^N)^2 \leq \frac{2H}{N}.$$

*(c) For any* $t \in \mathbb{N}_+$ *and* $\alpha \in (0, 1)$,

$$\frac{1}{t^{\alpha}} \leq \sum_{i=1}^{t} \frac{\eta_i^t}{i^{\alpha}} \leq \frac{2}{t^{\alpha}}.$$

The following lemma summarizes some basic but useful properties of the defined weights. When $(s, a, h)$ is clear from context, we also write $k^i := k^i(s, a, h)$ and $N_h^k := N_h^k(s, a)$ for simplicity.

**Lemma D.3.** *For any given non-negative weight sequence* $\{\omega_h^k\}_{k \in [K]}$ *at step* $h$, *the following relationships hold for any* $k' \in [K]$ *and* $h \leq h' < H$:

*(a)* $\sum_{s,a} \tilde{\omega}_h(k', h'+1, s, a) = \omega_h(k', h'+1)$.

*(b)* $\|\omega_h(\cdot, h', s, a)\|_{\infty} \leq \|\omega_h(\cdot, h')\|_{\infty}$.

*(c)* $\|\omega_h(\cdot, h', s, a)\|_1 \leq \|\omega_h(\cdot, h')\|_{\infty} N_{h'}^{K+1}(s, a)$.

*(d)* $\|\omega_h(\cdot, h')\|_1 = \sum_{s,a} \|\omega_h(\cdot, h', s, a)\|_1$.

*(e)* $\|\omega_h(\cdot, h'+1)\|_{\infty} \leq \left(1 + \frac{1}{H}\right) \|\omega_h(\cdot, h')\|_{\infty}$, $\|\omega_h(\cdot, h'+1)\|_1 \leq \|\omega_h(\cdot, h')\|_1$.

*Proof.* (a) is because

$$\sum_{s,a} \tilde{\omega}_h(k', h'+1, s, a) = \sum_{s,a} \omega_h(k', h'+1) \mathbb{I}[(s_{h'}^{k'}, a_{h'}^{k'}) = (s, a)] = \omega_h(k', h'+1).$$

(b) is because for any $k' \in [K]$

$$\omega_h(k', h', s, a) = \omega_h(k', h') \cdot \mathbb{I}\left[(s_{h'}^{k'}, a_{h'}^{k'}) = (s, a)\right] \leq \omega_h(k', h') \leq \|\omega_h(\cdot, h')\|_{\infty}.$$

(c) is because

$$\|\omega_h(\cdot, h', s, a)\|_1 \leq \|\omega_h(\cdot, h')\|_{\infty} \sum_{k'=1}^{K} \mathbb{I}\left[(s_{h'}^{k'}, a_{h'}^{k'}) = (s, a)\right] = \|\omega_h(\cdot, h')\|_{\infty} N_{h'}^{K+1}(s, a).$$

(d) is because

$$\sum_{s,a} \|\omega_h(\cdot, h', s, a)\|_1 = \sum_{s,a} \sum_{k'=1}^{K} \omega_h(k', h') \cdot \mathbb{I}[(s_{h'}^{k'}, a_{h'}^{k'}) = (s, a)] = \sum_{k'=1}^{K} \omega_h(k', h') = \|\omega_h(\cdot, h')\|_1.$$

For (e), we first prove that

$$\omega_h(k', h'+1) = \sum_{i=N_{h'}^{k'+1}(s_{h'}^{k'}, a_{h'}^{k'})}^{N_{h'}^{K+1}(s_{h'}^{k'}, a_{h'}^{k'})-1} \omega_h(k^{i+1}(s_{h'}^{k'}, a_{h'}^{k'}, h'), h')\eta_{N_{h'}^{k'+1}(s_{h'}^{k'}, a_{h'}^{k'})}^i$$

$$= \sum_{k=1}^{K} \omega_h(k, h') \sum_{j=1}^{N_{h'}^k(s_{h'}^k, a_{h'}^k)} \eta_j^{N_{h'}^k(s_{h'}^k, a_{h'}^k)} \mathbb{I}\left[k^j(s_{h'}^k, a_{h'}^k, h') = k'\right]. \quad (13)$$

This is because according to the definition of $k^j$, $\mathbb{I}\left[k^j(s_{h'}^k, a_{h'}^k, h') = k'\right] = 1$ if and only if $(s_{h'}^{k'}, a_{h'}^{k'}) = (s_{h'}^k, a_{h'}^k)$, $k' \leq k-1$ and $j = N_{h'}^{k'+1}(s_{h'}^{k'}, a_{h'}^{k'})$ and then we have:

$$\sum_{k=1}^{K} \omega_h(k, h') \sum_{j=1}^{N_{h'}^k(s_{h'}^k, a_{h'}^k)} \eta_j^{N_{h'}^k(s_{h'}^k, a_{h'}^k)} \mathbb{I}\left[k^j(s_{h'}^k, a_{h'}^k, h') = k'\right]$$

$$= \sum_{k=k'+1}^{K} \omega_h(k, h')\eta_{N_{h'}^{k'+1}(s_{h'}^{k'}, a_{h'}^{k'})}^{N_{h'}^k(s_{h'}^k, a_{h'}^k)} \mathbb{I}\left[(s_{h'}^{k'}, a_{h'}^{k'}) = (s_{h'}^k, a_{h'}^k)\right]$$

$$= \sum_{i=N_{h'}^{k'+1}(s_{h'}^{k'}, a_{h'}^{k'})}^{N_{h'}^{K+1}(s_{h'}^{k'}, a_{h'}^{k'})-1} \omega_h(k^{i+1}(s_{h'}^{k'}, a_{h'}^{k'}, h'), h')\eta_{N_{h'}^{k'+1}(s_{h'}^{k'}, a_{h'}^{k'})}^i$$

The last equation is because for $i = N_{h'}^k(s_{h'}^{k'}, a_{h'}^{k'})$ and $(s_{h'}^{k'}, a_{h'}^{k'}) = (s_{h'}^k, a_{h'}^k)$, we have $k = k^{i+1}(s_{h'}^{k'}, a_{h'}^{k'}, h')$. Moreover, due to the indicator $\mathbb{I}[(s_{h'}^{k'}, a_{h'}^{k'}) = (s_{h'}^k, a_{h'}^k)]$, the summation in the second equation above only includes episodes in which $(s_{h'}^{k'}, a_{h'}^{k'})$ is visited. Therefore, it terminates at the episode of the last visit to $(s_{h'}^{k'}, a_{h'}^{k'})$ with $i = N_{h'}^{K+1}(s_{h'}^{k'}, a_{h'}^{k'}) - 1$. Then for any $k' \in [K]$,

$$\omega_h(k', h'+1) \leq \|\omega_h(\cdot, h')\|_{\infty} \sum_{i=N_{h'}^{k'+1}(s_{h'}^{k'}, a_{h'}^{k'})}^{N_{h'}^{K+1}(s_{h'}^{k'}, a_{h'}^{k'})-1} \eta_{N_{h'}^{k'+1}(s_{h'}^{k'}, a_{h'}^{k'})}^i \leq \left(1 + \frac{1}{H}\right)\|\omega_h(\cdot, h')\|_{\infty}.$$

This proves the first conclusion. The second conclusion is proved by Equation (13) and

$$\sum_{k'=1}^{K} \omega_h(k', h'+1) = \sum_{k=1}^{K} \omega_h(k, h')\left(\sum_{i=1}^{N_{h'}^k} \eta_i^{N_{h'}^k}\right) \leq \sum_{k=1}^{K} \omega_h(k, h') = \|\omega_h(\cdot, h')\|_1.$$

$\square$

**Lemma D.4.** *For any non-negative weight sequence $\{\omega_h^k\}_k$ at step $h \in [H]$, any subsequent step $h' \in [h, H]$, any state-action pair $(s, a) \in \mathcal{S} \times \mathcal{A}$, and any $\alpha \in (0, 1)$, it holds that:*

$$\sum_{k=1, N_{h'}^k>0}^{K} \frac{\omega_h(k, h', s, a)}{N_{h'}^k(s_{h'}^k, a_{h'}^k)^\alpha} \leq \frac{1}{1-\alpha}\|\omega_h(\cdot, h')\|_{\infty}^{\alpha}\|\omega_h(\cdot, h', s, a)\|_1^{1-\alpha},$$

*and*

$$\sum_{k=1, N_{h'}^k>0}^{K} \frac{\omega_h(k, h')}{N_{h'}^k(s_{h'}^k, a_{h'}^k)^\alpha} \leq \frac{1}{1-\alpha}(SA\|\omega_h(\cdot, h')\|_{\infty})^{\alpha}\|\omega_h(\cdot, h')\|_1^{1-\alpha},$$

*Proof.* We first note that

$$\sum_{k=1, N_{h'}^k>0}^{K} \frac{\omega_h(k, h', s, a)}{N_{h'}^k(s_{h'}^k, a_{h'}^k)^\alpha} = \sum_{k=1, N_{h'}^k>0}^{K} \frac{\omega_h(k, h')\mathbb{I}[(s_{h'}^k, a_{h'}^k) = (s, a)]}{N_{h'}^k(s_{h'}^k, a_{h'}^k)^\alpha}$$

$$= \sum_{i=1}^{N_{h'}^K(s,a)} \frac{\omega_h(k^{i+1}(s, a, h'), h')}{i^\alpha}. \quad (14)$$

Then we have
$$\sum_{i=1}^{N_{h'}^K(s,a)} \omega_h(k^{i+1}(s,a,h'),h') \leq \|\omega_h(\cdot,h',s,a)\|_1.$$

Given the term on RHS of Equation (14), when the weights $\omega_h(k^{i+1}(s,a,h'),h')$ concentrate on the former terms with smaller index $i \geq 1$, we can obtain the largest value. Let

$$c_{s,a,h'} = \left\lceil \frac{\|\omega_h(\cdot,h',s,a)\|_1}{\|\omega_h(\cdot,h')\|_\infty} \right\rceil \quad \text{and} \quad d_{s,a,h'} = \|\omega_h(\cdot,h',s,a)\|_1 - (c_{s,a,h'}-1)\|\omega_h(\cdot,h')\|_\infty.$$

Then we have:

$$\sum_{k=1,N_{h'}^k>0}^{K} \frac{\omega_h(k,h',s,a)}{N_{h'}^k(s_{h'}^k,a_{h'}^k)^\alpha}$$

$$\leq \sum_{i=1}^{c_{s,a,h'}-1} \frac{\|\omega_h(\cdot,h')\|_\infty}{i^\alpha} + \frac{d_{s,a,h'}}{c_{s,a,h'}^\alpha}$$

$$\leq \|\omega_h(\cdot,h')\|_\infty \sum_{i=1}^{c_{s,a,h'}-1} \frac{i^{1-\alpha}-(i-1)^{1-\alpha}}{1-\alpha} + \frac{d_{s,a,h'}}{c_{s,a,h'}^\alpha} \quad (15)$$

$$= \frac{\|\omega_h(\cdot,h')\|_\infty(c_{s,a,h'}-1)^{1-\alpha}}{1-\alpha} + \frac{d_{s,a,h'}}{c_{s,a,h'}^\alpha}$$

$$= \|\omega_h(\cdot,h')\|_\infty^\alpha \left( \frac{[(c_{s,a,h'}-1)\|\omega_h(\cdot,h')\|_\infty]^{1-\alpha}}{1-\alpha} + \frac{d_{s,a,h'}}{(c_{s,a,h'}\|\omega_h(\cdot,h')\|_\infty)^\alpha} \right)$$

$$\leq \|\omega_h(\cdot,h')\|_\infty^\alpha \left( \frac{[(c_{s,a,h'}-1)\|\omega_h(\cdot,h')\|_\infty]^{1-\alpha}}{1-\alpha} + \frac{d_{s,a,h'}}{\|\omega_h(\cdot,h',s,a)\|_1^\alpha} \right). \quad (16)$$

Here the last inequality is because $c_{s,a,h'}\|\omega_h(\cdot,h')\|_\infty \geq \|\omega_h(\cdot,h',s,a)\|_1$. Equation (15) is because for any $0 < y < x$ and $\alpha \in (0,1)$, we have:

$$\frac{x-y}{x^\alpha} \leq \frac{1}{1-\alpha}(x^{1-\alpha}-y^{1-\alpha}).$$

Then, let $x = i$ and $y = i-1$, it holds that:

$$\frac{1}{i^\alpha} \leq \frac{1}{1-\alpha}(i^{1-\alpha}-(i-1)^{1-\alpha}).$$

Also let $x = \|\omega_h(\cdot,h',s,a)\|_1$ and $y = (c_{s,a,h'}-1)\|\omega_h(\cdot,h')\|_\infty$, we have:

$$\frac{d_{s,a,h'}}{\|\omega_h(\cdot,h',s,a)\|_1^\alpha} + \frac{[(c_{s,a,h'}-1)\|\omega_h(\cdot,h')\|_\infty]^{1-\alpha}}{1-\alpha} \leq \frac{\|\omega_h(\cdot,h',s,a)\|_1^{1-\alpha}}{1-\alpha}.$$

Applying this inequality to Equation (16), we have:

$$\sum_{k=1,N_{h'}^k>0}^{K} \frac{\omega_h(k,h')\mathbb{I}[(s_{h'}^k,a_{h'}^k)=(s,a)]}{N_{h'}^k(s_{h'}^k,a_{h'}^k)^\alpha} \leq \frac{1}{1-\alpha}\|\omega_h(\cdot,h')\|_\infty^\alpha \|\omega_h(\cdot,h',s,a)\|_1^{1-\alpha}.$$

Therefore, we have proved the first conclusion. By summing this conclusion for all state-action pairs $(s,a)$, we reach:

$$\sum_{k=1,N_{h'}^k>0}^{K} \frac{\omega_h(k,h')}{N_{h'}^k(s_{h'}^k,a_{h'}^k)^\alpha} \leq \sum_{s,a} \frac{1}{1-\alpha}\|\omega_h(\cdot,h')\|_\infty^\alpha \|\omega_h(\cdot,h',s,a)\|_1^{1-\alpha}$$

$$\leq \frac{1}{1-\alpha}(SA\|\omega_h(\cdot,h')\|_\infty)^\alpha\|\omega_h(\cdot,h')\|_1^{1-\alpha}.$$

The last inequality is by Hölder's inequality, as

$$\sum_{s,a} \|\omega_h(\cdot,h',s,a)\|_1^{1-\alpha} \leq (SA)^\alpha\|\omega_h(\cdot,h')\|_1^{1-\alpha}.$$

$\square$

# E PROOF OF THEOREM 3.1

## E.1 PROOF OF LEMMAS IN SECTION 3.2

Before proceeding to the proof, we will provide several key lemmas. By Lemma 4.3 of Jin et al. (2018), we have the following conclusion.

**Lemma E.1.** *Using $\forall(s,a,h,k)$ as the simplified notation for $\forall(s,a,h,k) \in \mathcal{S} \times \mathcal{A} \times [H] \times [K]$. With probability at least $1-p$, and $\beta_0 = 0$ and $\beta_t = 8\sqrt{\frac{H^3\iota}{t}}$ for $t \in \mathbb{N}_+$, the following event holds:*

$$\mathcal{E} = \left\{ 0 \leq (Q_h^k - Q_h^\star)(s,a) \leq \eta_0^{N_h^k} H + \sum_{i=1}^{N_h^k} \eta_i^{N_h^k} (V_{h+1}^{k^i} - V_{h+1}^\star)(s_{h+1}^{k^i}) + \beta_{N_h^k}, \forall(s,a,h,k) \right\}.$$

We now proceed to prove the lemmas used in Section 3.2. We begin with the proof of Lemma 3.1. In fact, this result holds for any learning algorithm.

**Lemma E.2** (Formal statement of Lemma 3.1). *For any learning algorithm with $K$ episodes and $T = HK$ steps, the expected regret is bounded as*

$$\mathbb{E}\left[\text{Regret}(T)\right] \leq \mathbb{E}\left( \sum_{h=1}^{H} \sum_{s,a} \Delta_h(s,a) N_h^{K+1}(s,a) \right).$$

*Proof.*

$$\begin{aligned}
\left(V_1^\star - V_1^{\pi^k}\right)(s_1^k) &= V_1^\star(s_1^k) - Q_1^\star(s_1^k, a_1^k) + \left(Q_1^\star - Q_1^{\pi^k}\right)(s_1^k, a_1^k) \\
&= \Delta_1(s_1^k, a_1^k) + \mathbb{E}\left[ \left(V_2^\star - V_2^{\pi^k}\right)(s_2^k) \mid s_2^k \sim P_1(\cdot \mid s_1^k, a_1^k) \right] \\
&= \mathbb{E}\left[ \Delta_1(s_1^k, a_1^k) + \Delta_2(s_2^k, a_2^k) \mid s_2^k \sim P_1(\cdot \mid s_1^k, a_1^k) \right] \\
&\quad + \mathbb{E}\left[ \left(Q_2^\star - Q_2^{\pi^k}\right)(s_2^k, a_2^k) \mid s_2^k \sim P_1(\cdot \mid s_1^k, a_1^k) \right] \\
&= \cdots = \mathbb{E}\left[ \sum_{h=1}^{H} \Delta_h\left(s_h^k, a_h^k\right) \,\middle|\, s_{h+1}^k \sim P_h(\cdot \mid s_h^k, a_h^k),\ h \in [H-1] \right].
\end{aligned}$$

Here, the second equation is from the Bellman Equation and the Bellman Optimality Equation in Equation (2). Therefore, we can get another expression of expected regret:

$$\mathbb{E}\left(\text{Regret}(T)\right) = \mathbb{E}\left[ \sum_{k=1}^{K} \left(V_1^\star - V_1^{\pi^k}\right)(s_1^k) \right] = \mathbb{E}\left[ \sum_{k=1}^{K} \sum_{h=1}^{H} \Delta_h(s_h^k, a_h^k) \right].$$

Note that

$$\begin{aligned}
\mathbb{E}(\text{Regret}(T)) &= \mathbb{E}\left( \sum_{h=1}^{H} \sum_{k=1}^{K} \Delta_h(s_h^k, a_h^k) \right) \\
&= \mathbb{E}\left( \sum_{h=1}^{H} \sum_{k=1}^{K} \sum_{s,a} \Delta_h(s,a) \mathbb{I}[(s_h^k, a_h^k) = (s,a)] \right) \\
&= \mathbb{E}\left( \sum_{h=1}^{H} \sum_{s,a} \Delta_h(s,a) N_h^{K+1}(s,a) \right).
\end{aligned}$$

We finish the proof of the lemma. $\qquad\square$

We then prove Lemma 3.2 by bounding the cumulative weighted estimation error

$$\sum_{k=1}^{K} \omega_h^k \left(Q_h^k - Q_h^\star\right)(s_h^k, a_h^k)$$

for each state-action pair $(s,a)$.

**Lemma E.3** (Formal statement of Lemma 3.2). *For UCB-Hoeffding, under event $\mathcal{E}$ in Lemma E.1, for any non-negative weight sequence $\{\omega_h^k\}_k$ at step $h$, it holds simultaneously for any $(s,a) \in \mathcal{S} \times \mathcal{A}$ and subsequent step $h' \in [h, H]$ that:*

$$\sum_{k=1}^{K} \omega_h(k, h', s, a)(Q_{h'}^k - Q_{h'}^\star)(s_{h'}^k, a_{h'}^k) \leq \sum_{k'=1}^{K} \tilde{\omega}_h(k', h'+1, s, a)(Q_{h'+1}^{k'} - Q_{h'+1}^\star)(s, a)_{h'+1}^{k'}$$
$$+ \|\omega_h(\cdot, h')\|_\infty H + 16\sqrt{H^3 \|\omega_h(\cdot, h')\|_\infty \|\omega_h(\cdot, h', s, a)\|_1 \iota}. \tag{17}$$

*and*

$$\sum_{k=1}^{K} \omega_h(k, h') \left(Q_{h'}^k - Q_{h'}^\star\right)(s_{h'}^k, a_{h'}^k) \leq \sum_{k'=1}^{K} \omega_h(k', h'+1)(Q_{h'+1}^{k'} - Q_{h'+1}^\star)(s_{h'+1}^{k'}, a_{h'+1}^{k'})$$
$$+ \|\omega_h(\cdot, h')\|_\infty SAH + 16 \sum_{s,a} \sqrt{H^3 \|\omega_h(\cdot, h')\|_\infty \|\omega_h(\cdot, h', s, a)\|_1 \iota}. \tag{18}$$

*Proof.* Under the event $\mathcal{E}$ in Lemma E.1, we have the following relationship

$$\sum_{k=1}^{K} \omega_h(k, h', s, a) \left(Q_{h'}^k - Q_{h'}^\star\right)(s_{h'}^k, a_{h'}^k) \leq \sum_{k=1}^{K} \omega_h(k, h', s, a)\eta_0^{N_{h'}^k} H$$
$$+ \sum_{k=1}^{K} \omega_h(k, h', s, a) \sum_{i=1}^{N_{h'}^k} \eta_i^{N_{h'}^k}(V_{h'+1}^{k^i} - V_{h'+1}^\star)(s_{h'+1}^{k^i}) + \sum_{k=1}^{K} \omega_h(k, h', s, a)\beta_{N_{h'}^k}. \tag{19}$$

Here $k^i = k^i(s_{h'}^k, a_{h'}^k, h')$. For the first term in Equation (19), we have

$$\sum_{k=1}^{K} \omega_h(k, h', s, a)\eta_0^{N_{h'}^k} H \leq \|\omega_h(\cdot, h', s, a)\|_\infty H \sum_{k=1}^{K} \mathbb{I}\left[(s_{h'}^k, a_{h'}^k) = (s, a), N_{h'}^k(s, a) = 0\right]$$
$$\leq \|\omega_h(\cdot, h')\|_\infty H. \tag{20}$$

The last inequality is because $\|\omega_h(\cdot, h', s, a)\|_\infty \leq \|\omega_h(\cdot, h')\|_\infty$ by (b) of Lemma D.3.

For the second term in Equation (19), we have

$$\sum_{k=1}^{K} \omega_h(k, h', s, a) \sum_{i=1}^{N_{h'}^k} \eta_i^{N_{h'}^k}(V_{h'+1}^{k^i} - V_{h'+1}^\star)(s_{h'+1}^{k^i})$$
$$= \sum_{k=1}^{K} \omega_h(k, h')\mathbb{I}\left[(s_{h'}^k, a_{h'}^k) = (s, a)\right] \sum_{i=1}^{N_{h'}^k} \eta_i^{N_{h'}^k}(V_{h'+1}^{k^i} - V_{h'+1}^\star)(s_{h'+1}^{k^i}) \left(\sum_{k'=1}^{K} \mathbb{I}[k^i = k']\right)$$
$$= \sum_{k'=1}^{K} (V_{h'+1}^{k'} - V_{h'+1}^\star)(s_{h'+1}^{k'}) \left(\sum_{k=1}^{K} \sum_{i=1}^{N_{h'}^k} \omega_h(k, h')\mathbb{I}\left[(s_{h'}^k, a_{h'}^k) = (s, a)\right] \eta_i^{N_{h'}^k} \mathbb{I}[k^i = k']\right)$$
$$\leq \sum_{k'=1}^{K} (Q_{h'+1}^{k'} - Q_{h'+1}^\star)(s_{h'+1}^{k'}, a_{h'+1}^{k'}) \left(\sum_{k=1}^{K} \sum_{i=1}^{N_{h'}^k} \omega_h(k, h')\eta_i^{N_{h'}^k} \mathbb{I}\left[k^i = k', (s_{h'}^k, a_{h'}^k) = (s, a)\right]\right)$$
$$= \sum_{k'=1}^{K} \tilde{\omega}_h(k', h'+1, s, a)(Q_{h'+1}^{k'} - Q_{h'+1}^\star)(s_{h'+1}^{k'}, a_{h'+1}^{k'}). \tag{21}$$

The inequality is by $Q_{h'+1}^{k'}(s_{h'+1}^{k'}, a_{h'+1}^{k'}) \geq V_{h'+1}^{k'}(s_{h'+1}^{k'})$, $Q_{h'+1}^\star(s_{h'+1}^{k'}, a_{h'+1}^{k'}) \leq V_{h'+1}^\star(s_{h'+1}^{k'})$. For Equation (21), by the definition of $k^i$, $\mathbb{I}\left[k^i(s_{h'}^k, a_{h'}^k, h') = k'\right] = 1$ holds only when $(s_{h'}^k, a_{h'}^k) =$

$(s_{h'}^k, a_{h'}^k)$ and then we have:

$$\sum_{k=1}^{K} \sum_{i=1}^{N_{h'}^k} \omega_h(k, h') \eta_i^{N_{h'}^k} \mathbb{I}\left[k^i = k', (s_{h'}^k, a_{h'}^k) = (s, a)\right]$$

$$= \mathbb{I}\left[(s_{h'}^{k'}, a_{h'}^{k'}) = (s, a)\right] \sum_{k=1}^{K} \sum_{i=1}^{N_{h'}^k} \omega_h(k, h') \eta_i^{N_{h'}^k} \mathbb{I}\left[k^i = k'\right] = \tilde{\omega}_h(k', h' + 1, s, a).$$

The last equation is because of Equation (13) and the definition of $\tilde{\omega}_h(k', h' + 1, s, a)$.

For the last term of Equation (19), by Lemma D.4, it holds that

$$\sum_{k=1}^{K} \omega_h(k, h', s, a) \beta_{N_{h'}^k} \leq 8\sqrt{H^3 \iota} \sum_{k=1, N_{h'}^k > 0}^{K} \omega_h(k, h', s, a) \sqrt{\frac{1}{N_{h'}^k(s_{h'}^k, a_{h'}^k)}}$$

$$\leq 16\sqrt{H^3 \|\omega_h(\cdot, h')\|_\infty \|\omega_h(\cdot, h', s, a)\|_1 \iota}. \tag{22}$$

Combining the results of Equation (20), Equation (21) and Equation (22), we finish the proof of Equation (17). Summing this conclusion over all state-action pairs $(s, a)$, and noting that $\sum_{s,a} \tilde{\omega}_h(k', h' + 1, s, a) = \omega_h(k', h' + 1)$, we prove Equation (18). □

Lemma 3.3 then follows immediately from a recursive application of the results established above.

**Lemma E.4** (Formal statement of Lemma 3.3). *For UCB-Hoeffding, under event $\mathcal{E}$ in Lemma E.1, for any non-negative weight sequence $\{\omega_h^k\}_k$ at step $h$, it holds simultaneously for any $(s, a) \in \mathcal{S} \times \mathcal{A}$ and subsequent step $h' \in [h, H]$ that:*

$$\sum_{k=1}^{K} \omega_h(k, h') \left(Q_{h'}^k - Q_{h'}^\star\right)(s_{h'}^k, a_{h'}^k)$$

$$\leq \sum_{h_1=h'}^{H} \|\omega_h(\cdot, h_1)\|_\infty SAH + 16 \sum_{h_1=h'}^{H} \sum_{s,a} \sqrt{H^3 \|\omega_h(\cdot, h_1)\|_\infty \|\omega_h(\cdot, h_1, s, a)\|_1 \iota}.$$

*Proof.* By applying recursion on steps $h', h' + 1, ..., H$ in Equation (18), since $Q_{H+1}^k(s, a) = Q_{H+1}^\star(s, a) = 0$ for any $(s, a, k) \in \mathcal{S} \times \mathcal{A} \times [K]$, the proof is complete. □

Building on the previous lemma, we now establish a novel upper bound on cumulative weighted visitation counts

$$\sum_{s,a} \Delta_h(s, a) N_h^{K+1}(s, a),$$

which then enables the final bound on expected regret through Lemma E.2.

**Lemma E.5** (Formal statement of Lemma 3.4). *For UCB-Hoeffding algorithm and $c_1 = 20736$, under the event $\mathcal{E}$ in Lemma E.1, it holds simultaneously for any $h \in [H]$ that:*

$$\sum_{s,a} \frac{\Delta_h(s, a) N_h^{K+1}(s, a)}{c_1} \leq SAH^2 + \sum_{h'=h}^{H} \sum_{\Delta_{h'}(s,a)>0} \frac{H^4 \iota}{\Delta_{h'}(s, a)} + \frac{H^3 \left(\sum_{t=h+1}^{H} \sqrt{|Z_{\text{opt},t}|}\right)^2 \iota}{\Delta_{\min,h}}$$

$$+ \sum_{h'=h+1}^{H} \frac{H^2 \left(\sum_{t=h'+1}^{H} \sqrt{|Z_{\text{opt},t}|}\right)^2 \iota}{\Delta_{\min,h'}}.$$

*Proof.* We use mathematical induction to prove this conclusion. For step $h$, let

$$\omega_h^k = \mathbb{I}\left[Q_h^k(s_h^k, a_h^k) - Q_h^\star(s_h^k, a_h^k) \geq \Delta_h(s_h^k, a_h^k), (s_h^k, a_h^k) \in Z_{\text{sub},h}\right]$$

$$= \mathbb{I}\left[(s_h^k, a_h^k) \in Z_{\text{sub},h}\right] \leq 1.$$

The second equation is because for any given $(h,k) \in [H] \times [k]$, if $(s_h^k, a_h^k) \in Z_{\text{sub},h}$, we have

$$Q_h^k(s_h^k, a_h^k) - Q_h^\star(s_h^k, a_h^k) \geq V_h^k(s_h^k) - Q_h^\star(s_h^k, a_h^k) \geq V_h^\star(s_h^k) - Q_h^\star(s_h^k, a_h^k) = \Delta_h(s_h^k, a_h^k) > 0.$$

The first inequality holds because $Q_h^k(s_h^k, a_h^k) \geq V_h^k(s_h^k)$, as guaranteed by the update rule in line 8 of Algorithm 1. The second inequality follows directly from the $\mathcal{E}$ in Lemma E.1, which ensures that $Q_h^k(s_h^k, a) \geq Q_h^\star(s_h^k, a)$ for all $(a, h, k) \in \mathcal{A} \times [H] \times [K]$ and thus

$$V_h^k(s_h^k) = \min\left\{H, \max_a Q_h^k(s_h^k, a)\right\} \geq \min\left\{H, \max_a Q_h^\star(s_h^k, a)\right\} = \max_a Q_h^\star(s_h^k, a) = V_h^\star(s_h^k).$$

Based on the definition of $\omega_h^k$, for any $(s,a) \in Z_{\text{sub},h}$, we have

$$\|\omega_h(\cdot, h, s, a)\|_1 = \sum_{k=1}^K \mathbb{I}\left[(s_h^k, a_h^k) = (s,a)\right] = N_h^{K+1}(s,a)$$

and $\|\omega_h(\cdot, h, s, a)\|_1 = 0$ for $(s,a) \in Z_{\text{opt},h}$. By Lemma E.3, for any $(s,a) \in Z_{\text{sub},h}$, it holds that,

$$\sum_{k=1}^K \omega_h^k \left(Q_h^k - Q_h^\star\right)(s_h^k, a_h^k)\mathbb{I}[(s_h^k, a_h^k) = (s,a)] = \sum_{k=1}^K \omega_h(k, h, s, a)\left(Q_h^k - Q_h^\star\right)(s_h^k, a_h^k)$$

$$\leq H + 16\sqrt{H^3 N_h^{K+1}(s,a)\iota} + \sum_{k'=1}^K \tilde{\omega}_h(k', h+1, s, a)(Q_{h+1}^{k'} - Q_{h+1}^\star)(s_{h+1}^{k'}, a_{h+1}^{k'}). \tag{23}$$

Also note that for any $(s,a) \in Z_{\text{sub},h}$, we have

$$\sum_{k=1}^K \omega_h^k \left(Q_h^k - Q_h^\star\right)(s_h^k, a_h^k)\mathbb{I}[(s_h^k, a_h^k) = (s,a)] \geq \Delta_h(s,a)\sum_{k=1}^K \omega_h^k \mathbb{I}[(s_h^k, a_h^k) = (s,a)]$$

$$= \Delta_h(s,a)N_h^{K+1}(s,a). \tag{24}$$

Combining the results of Equation (23) and Equation (24), it holds for any $(s,a) \in Z_{\text{sub},h}$ that,

$$\Delta_h(s,a)N_h^{K+1}(s,a)$$

$$\leq H + 16\sqrt{H^3 N_h^{K+1}(s,a)\iota} + \sum_{k'=1}^K \tilde{\omega}_h(k', h+1, s, a)(Q_{h+1}^{k'} - Q_{h+1}^\star)(s_{h+1}^{k'}, a_{h+1}^{k'}).$$

Solving this inequality, we can derive the following conclusion for any $(s,a) \in Z_{\text{sub},h}$:

$$\Delta_h(s,a)N_h^{K+1}(s,a) \leq \frac{256H^3\iota}{\Delta_h(s,a)} + 2H + 2\sum_{k'=1}^K \tilde{\omega}_h(k', h+1, s, a)(Q_{h+1}^{k'} - Q_{h+1}^\star)(s_{h+1}^{k'}, a_{h+1}^{k'}).$$

Since $\Delta_h(s,a) = 0$ for $(s,a) \notin Z_{\text{sub},h}$ and $Q_{h+1}^k(s,a) \geq Q_{h+1}^\star(s,a)$ for any $(s,a,h,k) \in \mathcal{S} \times \mathcal{A} \times [H] \times [K]$, by summing the inequality above over all state-action pairs $(s,a) \in Z_{\text{sub},h}$, we reach:

$$\sum_{s,a} \Delta_h(s,a)N_h^{K+1}(s,a)$$

$$\leq \sum_{\Delta_h(s,a)>0} \frac{256H^3\iota}{\Delta_h(s,a)} + 2SAH + 2\sum_{k'=1}^K \omega_h(k', h+1)(Q_{h+1}^{k'} - Q_{h+1}^\star)(s_{h+1}^{k'}, a_{h+1}^{k'}). \tag{25}$$

Here we use

$$\sum_{(s,a)\in Z_{\text{sub},h}} \tilde{\omega}_h(k', h+1, s, a) \leq \omega_h(k', h+1)$$

by (a) of Lemma D.3.

Let $h = H$, since $Q_{H+1}^k(s,a) = Q_{H+1}^\star(s,a) = 0$ for any $(s,a,k) \in \mathcal{S} \times \mathcal{A} \times [K]$, we prove the lemma for $h = H$ with Equation (25). Assuming the conclusion holds for steps $h+1, \ldots, H$, we now prove it for step $h$.

By Lemma E.4, we have

$$\sum_{k'=1}^{K} \omega_h(k', h+1)(Q_{h+1}^{k'} - Q_{h+1}^{\star})(s_{h+1}^{k'}, a_{h+1}^{k'})$$

$$\leq \sum_{h'=h+1}^{H} \|\omega_h(\cdot, h')\|_\infty SAH + 16 \sum_{h'=h+1}^{H} \sum_{s,a} \sqrt{H^3 \|\omega_h(\cdot, h')\|_\infty \|\omega_h(\cdot, h', s, a)\|_1 \iota} \qquad (26)$$

with

$$\|\omega_h(\cdot, h')\|_\infty \leq \left(1 + \frac{1}{H}\right)^{h'-h} \|\omega_h(\cdot, h)\|_\infty \leq 3, \qquad (27)$$

and

$$\|\omega_h(\cdot, h')\|_1 \leq \|\omega_h(\cdot, h)\|_1 = \sum_{s,a} \|\omega_h(\cdot, h, s, a)\|_1 = \sum_{\Delta_h(s,a)>0} N_h^{K+1}(s, a). \qquad (28)$$

by part (e) of Lemma D.3. In this case, by Equation (27) and part (c) of Lemma D.3, we further obtain the following bound:

$$\|\omega_h(\cdot, h', s, a)\|_1 \leq \|\omega_h(\cdot, h')\|_\infty N_{h'}^{K+1}(s, a) \leq 3N_{h'}^{K+1}(s, a). \qquad (29)$$

Furthermore, by Equation (27), for the first term in Equation (26), we have:

$$\sum_{h'=h+1}^{H} \|\omega_h(\cdot, h')\|_\infty SAH \leq 3SAH^2.$$

For the second term in Equation (26), we divide the state-action pairs $(s, a)$ at each step $h'$ into two categories: $Z_{\text{opt}, h'}$, where $\Delta_{h'}(s, a) = 0$, and $Z_{\text{sub}, h'}$, where $\Delta_{h'}(s, a) > 0$. We apply the Cauchy–Schwarz inequality to all sub-optimal state-action pairs **jointly across all steps**, and to optimal state-action pairs **individually at each step** $h'$.

$$16 \sum_{h'=h+1}^{H} \sum_{s,a} \sqrt{H^3 \|\omega_h(\cdot, h')\|_\infty \|\omega_h(\cdot, h', s, a)\|_1 \iota}$$

$$\leq 16\sqrt{3} \sqrt{H^3 \iota \left( \sum_{h'=h+1}^{H} \sum_{\Delta_{h'}(s,a)>0} \frac{1}{\Delta_{h'}(s,a)} \right) \left( \sum_{h'=h+1}^{H} \sum_{\Delta_{h'}(s,a)>0} \Delta_{h'}(s,a) \|\omega_h(\cdot, h', s, a)\|_1 \right)}$$

$$+ 16\sqrt{3} \sum_{h'=h+1}^{H} \sqrt{H^3 |Z_{\text{opt}, h'}| \iota \sum_{(s,a) \in Z_{\text{opt}, h'}} \|\omega_h(\cdot, h', s, a)\|_1}$$

$$\leq 48 \sqrt{H^3 \iota \left( \sum_{h'=h+1}^{H} \sum_{\Delta_{h'}(s,a)>0} \frac{1}{\Delta_{h'}(s,a)} \right) \left( \sum_{h'=h+1}^{H} \sum_{\Delta_{h'}(s,a)>0} \Delta_{h'}(s,a) N_{h'}^{K+1}(s,a) \right)}$$

$$+ 16\sqrt{3} \left( \sum_{h'=h+1}^{H} \sqrt{H^3 |Z_{\text{opt}, h'}| \iota} \right) \sqrt{\sum_{\Delta_h(s,a)>0} N_h^{K+1}(s, a)}. \qquad (30)$$

The last inequality is because $\|\omega_h(\cdot, h', s, a)\|_1 \leq 3N_{h'}^{K+1}(s, a)$ by Equation (29) and

$$\sum_{(s,a) \in Z_{\text{opt}, h'}} \|\omega_h(\cdot, h', s, a)\|_1 \leq \|\omega_h(\cdot, h')\|_1 \leq \sum_{\Delta_h(s,a)>0} N_h^{K+1}(s, a),$$

where the first inequality follows from part (d) of Lemma D.3, and the second from Equation (28).

For the first term in Equation (30), by AM-GM inequality, we have:

$$48 \sqrt{H^3 \iota \left( \sum_{h'=h+1}^{H} \sum_{\Delta_{h'}(s,a)>0} \frac{1}{\Delta_{h'}(s,a)} \right) \left( \sum_{h'=h+1}^{H} \sum_{\Delta_{h'}(s,a)>0} \Delta_{h'}(s,a) N_{h'}^{K+1}(s,a) \right)}$$

$$\leq 24\sqrt{c_1} \sum_{h'=h+1}^{H} \sum_{\Delta_{h'}(s,a)>0} \frac{H^4 \iota}{\Delta_{h'}(s,a)} + 24\sqrt{c_1} \sum_{h'=h+1}^{H} \sum_{\Delta_{h'}(s,a)>0} \frac{\Delta_{h'}(s,a) N_{h'}^{K+1}(s,a)}{Hc_1}. \qquad (31)$$

By the induction hypothesis, the lemma holds for all steps $h + 1 \leq h' \leq H$. Therefore, we obtain:

$$\sum_{s,a} \frac{\Delta_{h'}(s,a) N_{h'}^{K+1}(s,a)}{c_1} \leq SAH^2 + \sum_{i=h'}^{H} \sum_{\Delta_i(s,a)>0} \frac{H^4 \iota}{\Delta_i(s,a)}$$
$$+ \frac{H^3 \left(\sum_{t=h'+1}^{H} \sqrt{|Z_{\text{opt},t}|}\right)^2 \iota}{\Delta_{\min,h'}} + \sum_{i=h'+1}^{H} \frac{H^2 \left(\sum_{t=i+1}^{H} \sqrt{|Z_{\text{opt},t}|}\right)^2 \iota}{\Delta_{\min,i}}.$$

By summing this inequality for $h + 1 \leq h' \leq H$, it holds that:

$$\sum_{h'=h+1}^{H} \sum_{\Delta_{h'}(s,a)>0} \frac{\Delta_{h'}(s,a) N_{h'}^{K+1}(s,a)}{H c_1} \leq \sum_{h'=h+1}^{H} SAH + \sum_{h'=h+1}^{H} \sum_{i=h'}^{H} \sum_{\Delta_i(s,a)>0} \frac{H^3 \iota}{\Delta_i(s,a)}$$
$$+ \sum_{h'=h+1}^{H} \frac{H^2 \left(\sum_{t=h'+1}^{H} \sqrt{|Z_{\text{opt},t}|}\right)^2 \iota}{\Delta_{\min,h'}} + \sum_{h'=h+1}^{H} \sum_{i=h'+1}^{H} \frac{H \left(\sum_{t=i+1}^{H} \sqrt{|Z_{\text{opt},t}|}\right)^2 \iota}{\Delta_{\min,i}}$$
$$\leq SAH^2 + \sum_{h'=h+1}^{H} \sum_{\Delta_{h'}(s,a)>0} \frac{H^4 \iota}{\Delta_{h'}(s,a)} + 2 \sum_{h'=h+1}^{H} \frac{H^2 \left(\sum_{t=h'+1}^{H} \sqrt{|Z_{\text{opt},t}|}\right)^2 \iota}{\Delta_{\min,h'}}.$$

Applying the above inequality to Equation (31) and substituting it into Equation (30), we obtain:

$$16 \sum_{h'=h+1}^{H} \sum_{s,a} \sqrt{H^3 \|\omega_h(\cdot,h')\|_\infty \|\omega_h(\cdot,h',s,a)\|_1 \iota}$$
$$\leq 24\sqrt{c_1} \left(SAH^2 + 2 \sum_{h'=h+1}^{H} \sum_{\Delta_{h'}(s,a)>0} \frac{H^4 \iota}{\Delta_{h'}(s,a)} + 2 \sum_{h'=h+1}^{H} \frac{H^2 \left(\sum_{t=h'+1}^{H} \sqrt{|Z_{\text{opt},t}|}\right)^2 \iota}{\Delta_{\min,h'}}\right)$$
$$+ 16\sqrt{3} \left(\sum_{h'=h+1}^{H} \sqrt{H^3 |Z_{\text{opt},h'}| \iota}\right) \sqrt{\sum_{\Delta_h(s,a)>0} N_h^{K+1}(s,a)}. \tag{32}$$

By applying this inequality to Equation (26) and substituting the result into Equation (25), and using the bound $\|\omega_h(\cdot,h')\|_\infty \leq 3$ from Equation (27), we conclude that the following inequality holds:

$$\sum_{s,a} \Delta_h(s,a) N_h^{K+1}(s,a)$$
$$\leq 96\sqrt{c_1} \left(SAH^2 + \sum_{h'=h}^{H} \sum_{\Delta_{h'}(s,a)>0} \frac{H^4 \iota}{\Delta_{h'}(s,a)} + \sum_{h'=h+1}^{H} \frac{H^2 \left(\sum_{t=h'+1}^{H} \sqrt{|Z_{\text{opt},t}|}\right)^2 \iota}{\Delta_{\min,h'}}\right)$$
$$+ 32\sqrt{3} \left(\sum_{h'=h+1}^{H} \sqrt{H^3 |Z_{\text{opt},h'}| \iota}\right) \sqrt{\sum_{\Delta_h(s,a)>0} N_h^{K+1}(s,a)}. \tag{33}$$

Note that if $|Z_{\text{sub},h}| > 0$, which means $\Delta_{\min,h} > 0$, we have

$$\sum_{s,a} \Delta_h(s,a) N_h^{K+1}(s,a) = \sum_{\Delta_h(s,a)>0} \Delta_h(s,a) N_h^{K+1}(s,a) \geq \Delta_{\min,h} \sum_{\Delta_h(s,a)>0} N_h^{K+1}(s,a).$$

Define

$$b = \Delta_{\min,h}, \ c = 32\sqrt{3} \left(\sum_{h'=h+1}^{H} \sqrt{H^3 |Z_{\text{opt},h'}| \iota}\right), \ x = \sqrt{\sum_{\Delta_h(s,a)>0} N_h^{K+1}(s,a)}$$

and let the first term on the right-hand side of Equation (33) be denoted by $d$. Then Equation (33) can be rewritten as:

$$bx^2 - cx - d \leq 0.$$

When $b > 0$, solving the inequality yields:

$$x \leq \frac{c + \sqrt{c^2 + 4bd}}{2b}.$$

Applying this upper bound to Equation (33), by AM-GM inequality, we obtain

$$\sum_{s,a} \Delta_h(s,a) N_h^{K+1}(s,a) \leq cx + d \leq \frac{c^2 + c\sqrt{c^2 + 4bd}}{2b} + d \leq \frac{3c^2}{2b} + \frac{3d}{2}$$

$$\leq 144\sqrt{c_1} \left( SAH^2 + \sum_{h'=h}^{H} \sum_{\Delta_{h'}(s,a)>0} \frac{H^4\iota}{\Delta_{h'}(s,a)} + \sum_{h'=h+1}^{H} \frac{H^2 \left( \sum_{t=h'+1}^{H} \sqrt{|Z_{\mathrm{opt},t}|} \right)^2 \iota}{\Delta_{\min,h'}} \right)$$

$$+ 4608 \frac{H^3 \iota \left( \sum_{t=h+1}^{H} \sqrt{|Z_{\mathrm{opt},t}|} \right)^2}{\Delta_{\min,h}}.$$

If $|Z_{\mathrm{sub},h}| = 0$, then $\Delta_{\min,h} = \infty$ and $\sum_{s,a} \Delta_h(s,a) N_h^{K+1}(s,a) = 0$. In this case, the conclusion holds trivially. Therefore, the result is established for step $h$, completing the proof. $\square$

### E.2 BOUNDING THE EXPECTED REGRET

Now we bound the gap-dependent expected regret. Let $p = \frac{1}{T}$, then $\mathcal{E}$ holds with probability at least $1 - \frac{1}{T}$ and $\iota \leq O(\log(SAT))$. Therefore, by Lemma E.2, we have

$$\mathbb{E}(\mathrm{Regret}(T)) = \mathbb{E} \left( \sum_{h=1}^{H} \sum_{s,a} \Delta_h(s,a) N_h^{K+1}(s,a) \right)$$

$$= \mathbb{E} \left( \sum_{h=1}^{H} \sum_{s,a} \Delta_h(s,a) N_h^{K+1}(s,a) \Big| \mathcal{E} \right) \mathbb{P}(\mathcal{E}) + \mathbb{E} \left( \sum_{h=1}^{H} \sum_{s,a} \Delta_h(s,a) N_h^{K+1}(s,a) \Big| \mathcal{E}^c \right) \mathbb{P}(\mathcal{E}^c)$$

$$\leq O \left( \sum_{h=1}^{H} \sum_{\Delta_h(s,a)>0} \frac{H^5 \log(SAT)}{\Delta_h(s,a)} + \sum_{h=1}^{H} \frac{H^3 \left( \sum_{t=h+1}^{H} \sqrt{|Z_{\mathrm{opt},t}|} \right)^2 \log(SAT)}{\Delta_{\min,h}} + SAH^3 \right)$$

$$\leq O \left( \sum_{h=1}^{H} \sum_{\Delta_h(s,a)>0} \frac{H^5 \log(SAT)}{\Delta_h(s,a)} + \frac{H^5 |Z_{\mathrm{opt}}| \log(SAT)}{\Delta_{\min}} + SAH^3 \right).$$

The first inequality is because under the event $\mathcal{E}$, by Lemma E.5, we have

$$\sum_{h=1}^{H} \sum_{s,a} \Delta_h(s,a) N_h^{K+1}(s,a)$$

$$\leq \sum_{h=1}^{H} SAH^2 + \sum_{h=1}^{H} \sum_{h'=h}^{H} \sum_{\Delta_{h'}(s,a)>0} \frac{H^4\iota}{\Delta_{h'}(s,a)} + \sum_{h=1}^{H} \frac{H^3 \left( \sum_{t=h+1}^{H} \sqrt{|Z_{\mathrm{opt},t}|} \right)^2 \iota}{\Delta_{\min,h}}$$

$$+ \sum_{h=1}^{H} \sum_{h'=h+1}^{H} \frac{H^2 \left( \sum_{t=h'+1}^{H} \sqrt{|Z_{\mathrm{opt},t}|} \right)^2 \iota}{\Delta_{\min,h'}}$$

$$\leq O \left( \sum_{h=1}^{H} \sum_{\Delta_h(s,a)>0} \frac{H^5 \log(SAT)}{\Delta_h(s,a)} + \sum_{h=1}^{H} \frac{H^3 \left( \sum_{t=h+1}^{H} \sqrt{|Z_{\mathrm{opt},t}|} \right)^2 \log(SAT)}{\Delta_{\min,h}} + SAH^3 \right)$$

by Lemma E.5 and under the event $\mathcal{E}^c$,

$$\sum_{h=1}^{H} \sum_{s,a} \Delta_h(s,a) N_h^{K+1}(s,a) \leq HT.$$

The last inequality uses Cauchy-Schwarz inequality and $\Delta_{\min,h} \geq \Delta_{\min}$ for any $h \in [H]$.

# F  PROOF OF REGRET UPPER BOUNDS FOR ULCB-HOEFFDING

## F.1  AUXILIARY LEMMAS

We first validate the upper bounds $\overline{Q}$ and lower bounds $\underline{Q}$ introduced in Algorithm 2. For simplicity, we denote $\mathbb{P}_{s,a,h}f = \mathbb{E}_{s_{h+1}\sim\mathbb{P}_h(\cdot|s,a)}(f(s_{h+1})|s_h = s, \overline{a}_h = a)$ and $\mathbb{1}_s f = f(s)$ for any $(s,a,h) \in \mathcal{S} \times \mathcal{A} \times [H]$ and function $f : \mathcal{S} \to \mathbb{R}$. We first prove some probability events to facilitate our proof.

**Lemma F.1.** *For the ULCB-Hoeffding algorithm (Algorithm 2), we have the following conclusions:*

*(a) With probability at least $1 - p$, the following event holds:*

$$\mathcal{G}_1 = \left\{ \left| \sum_{i=1}^{N_h^k} \eta_i^{N_h^k} \left( \mathbb{1}_{s_{h+1}^{k^i}} - \mathbb{P}_{s,a,h} \right) V_{h+1}^\star \right| \leq 2\sqrt{\frac{H^3\iota}{N_h^k(s,a)}}, \ \forall(s,a,h,k) \right\}.$$

*(b) With probability at least $1 - p$, the following event holds:*

$$\mathcal{G}_2 = \left\{ \sum_{h=1}^{H}\sum_{k=1}^{K} \left(1 + \frac{1}{H}\right)^{h-1} \left( \mathbb{P}_{s_h^k,a_h^k,h} - \mathbb{1}_{s_{h+1}^k} \right) \left( V_{h+1}^\star - V_{h+1}^{\pi^k} \right) \leq 27\sqrt{2H^2T\iota} \right\}.$$

*Proof.* (a) The sequence

$$\left\{ \sum_{i=1}^{N} \eta_i^N \left( \mathbb{1}_{s_{h+1}^{k^i}} - \mathbb{P}_{s,a,h} \right) V_{h+1}^\star \right\}_{N\in\mathbb{N}^+}$$

is a martingale sequence with

$$\left| \sum_{i=1}^{N} \eta_i^N \left( \mathbb{1}_{s_{h+1}^{k^i}} - \mathbb{P}_{s,a,h} \right) V_{h+1}^\star \right| \leq \eta_i^N H.$$

Then according to Azuma-Hoeffding inequality and (b) of Lemma D.2, for any $p \in (0,1)$, with probability at least $1 - \frac{p}{SAT}$, it holds for given $N_h^k(s,a) = N \in \mathbb{N}_+$ that:

$$\left| \sum_{i=1}^{N} \eta_i^N \left( \mathbb{1}_{s_{h+1}^{k^i}} - \mathbb{P}_{s,a,h} \right) V_{h+1}^\star \right| \leq 2\sqrt{\frac{H^3\iota}{N}}.$$

For any $(s,a,h,k) \in \mathcal{S} \times \mathcal{A} \times [H] \times [K]$, we have $N_h^k(s,a) \in [\frac{T}{H}]$. Considering all the possible combinations $(s,a,h,N) \in \mathcal{S}\times\mathcal{A}\times[H]\times[\frac{T}{H}]$, with probability at least $1-p$, it holds simultaneously for all $(s,a,h,k) \in \mathcal{S} \times \mathcal{A} \times [H] \times [K]$ that:

$$\left| \sum_{i=1}^{N_h^k} \eta_i^{N_h^k} \left( \mathbb{1}_{s_{h+1}^{k^i}} - \mathbb{P}_{s,a,h} \right) V_{h+1}^\star \right| \leq 2\sqrt{\frac{H^3\iota}{N_h^k(s,a)}}.$$

(b) For $\mathcal{G}_2$, the sequence

$$\left\{ \left(1 + \frac{1}{H}\right)^{h-1} \left( \mathbb{P}_{s_h^k,a_h^k,h} - \mathbb{1}_{s_{h+1}^k} \right) \left( V_{h+1}^\star - V_{h+1}^{\pi^k} \right) \right\}_{k,h}$$

can be reordered to a martingale sequence based on the "episode first, step second" rule. The absolute values of the sequence are bounded by $27H$. According to Azuma-Hoeffding inequality, for any $p \in (0,1)$, with probability at least $1 - p$, it holds that:

$$\sum_{h=1}^{H}\sum_{k=1}^{K} \left(1 + \frac{1}{H}\right)^{h-1} \left( \mathbb{P}_{s_h^k,a_h^k,h} - \mathbb{1}_{s_{h+1}^k} \right) \left( V_{h+1}^\star - V_{h+1}^{\pi^k} \right) \leq 27\sqrt{2H^2T\iota}.$$

$\square$

**Lemma F.2.** *For all $(s, a, h, k) \in \mathcal{S} \times \mathcal{A} \times [H] \times [K]$, when event $\mathcal{G}_1$ in Lemma F.1 happens, the upper and lower confidence bounds in Algorithm 2 are valid:*

$$\overline{V}_h^k(s) \geq V_h^\star(s) \geq \underline{V}_h^k(s) \quad \text{and} \quad \overline{Q}_h^k(s, a) \geq Q_h^\star(s, a) \geq \underline{Q}_h^k(s, a).$$

*Proof.* We use mathematical induction on $k$ to prove this lemma. For $k = 1$, the lemma holds based on the initialization in line 2 of Algorithm 2. Assuming the conclusion holds for all $1, 2, ..., k - 1$, we will prove the conclusion for $k + 1$ at episode $k$.

If $(s, a, h) \in \mathcal{S} \times A \times [H] \setminus \{(s_h^k, a_h^k)\}_{h=1}^H$, then we have

$$\overline{V}_h^{k+1}(s) = \overline{V}_h^k(s) \geq V_h^\star(s) \geq \underline{V}_h^k(s) = \underline{V}_h^{k+1}(s).$$

and

$$\overline{Q}_h^{k+1}(s, a) = \overline{Q}_h^k(s, a) \geq Q_h^\star(s, a) \geq \underline{Q}_h^k(s, a) = \underline{Q}_h^{k+1}(s, a).$$

For $(s_h^k, a_h^k, h)$, based on the update rule in line 12 and line 13 in Algorithm 2, we have

$$\overline{Q}_h^{k+1}(s_h^k, a_h^k) = \eta_0^{N_h^{k+1}} H + \sum_{i=1}^{N_h^{k+1}} \eta_i^{N_h^{k+1}} \left( r_h(s_h^k, a_h^k) + \overline{V}_{h+1}^{k^i}(s_{h+1}^{k^i}) + b_i \right)$$

$$\geq \eta_0^{N_h^{k+1}} H + \sum_{i=1}^{N_h^{k+1}} \eta_i^{N_h^{k+1}} \left( r_h(s_h^k, a_h^k) + \overline{V}_{h+1}^{k^i}(s_{h+1}^{k^i}) \right) + 2\sqrt{\frac{H^3 \iota}{N_h^{k+1}}}. \tag{34}$$

$$\underline{Q}_h^{k+1}(s_h^k, a_h^k) = \sum_{i=1}^{N_h^{k+1}} \eta_i^{N_h^{k+1}} \left( r_h(s_h^k, a_h^k) + \underline{V}_{h+1}^{k^i}(s_{h+1}^{k^i}) - b_i \right).$$

$$\leq \sum_{i=1}^{N_h^{k+1}} \eta_i^{N_h^{k+1}} \left( r_h(s_h^k, a_h^k) + \underline{V}_{h+1}^{k^i}(s_{h+1}^{k^i}) \right) - 2\sqrt{\frac{H^3 \iota}{N_h^{k+1}}}. \tag{35}$$

These two inequalities are because

$$\sum_{i=1}^{N_h^{k+1}} \eta_i^{N_h^{k+1}} b_i = 2 \sum_{i=1}^{N_h^{k+1}} \eta_i^{N_h^{k+1}} \sqrt{\frac{H^3 \iota}{i}} \geq 2\sqrt{\frac{H^3 \iota}{N_h^{k+1}}}$$

by (c) of Lemma D.2. Furthermore, by the Bellman Optimality Equation, it holds that:

$$Q_h^\star(s_h^k, a_h^k) = r_h(s_h^k, a_h^k) + \mathbb{P}_{s_h^k, a_h^k, h} V_{h+1}^\star.$$

Combining with Equation (34) and Equation (35), we can derive the following conclusion:

$$\left( \overline{Q}_h^{k+1} - Q_h^\star \right)(s_h^k, a_h^k)$$

$$\geq \sum_{i=1}^{N_h^{k+1}} \eta_i^{N_h^{k+1}} \left( \overline{V}_{h+1}^{k^i}(s_{h+1}^{k^i}) - \mathbb{P}_{s_h^k, a_h^k, h} V_{h+1}^\star \right) + 2\sqrt{\frac{H^3 \iota}{N_h^{k+1}}}$$

$$= \sum_{i=1}^{N_h^{k+1}} \eta_i^{N_h^{k+1}} (\overline{V}_{h+1}^{k^i} - V_{h+1}^\star)(s_{h+1}^{k^i}) + \sum_{i=1}^{N_h^{k+1}} \eta_i^{N_h^{k+1}} (\mathbb{1}_{s_{h+1}^{k^i}} - \mathbb{P}_{s_h^k, a_h^k, h}) V_{h+1}^\star + 2\sqrt{\frac{H^3 \iota}{N_h^{k+1}}} \geq 0.$$

The last inequality is because $\overline{V}_{h+1}^{k^i}(s_{h+1}^{k^i}) \geq V_{h+1}^\star(s_{h+1}^{k^i})$ for $k^i \leq k$ and the event $\mathcal{G}_1$. Similarly,

$$\left( \underline{Q}_h^{k+1} - Q_h^\star \right)(s_h^k, a_h^k)$$

$$\leq \sum_{i=1}^{N_h^{k+1}} \eta_i^{N_h^{k+1}} \left( \underline{V}_{h+1}^{k^i}(s_{h+1}^{k^i}) - \mathbb{P}_{s_h^k, a_h^k, h} V_{h+1}^\star \right) - 2\sqrt{\frac{H^3 \iota}{N_h^{k+1}}}$$

$$= \sum_{i=1}^{N_h^{k+1}} \eta_i^{N_h^{k+1}} (\underline{V}_{h+1}^{k^i} - V_{h+1}^\star)(s_{h+1}^{k^i}) + \sum_{i=1}^{N_h^{k+1}} \eta_i^{N_h^{k+1}} (\mathbb{1}_{s_{h+1}^{k^i}} - \mathbb{P}_{s_h^k, a_h^k, h}) V_{h+1}^\star - 2\sqrt{\frac{H^3 \iota}{N_h^{k+1}}} \leq 0.$$

The last inequality is because $\underline{V}_{h+1}^{k^i}(s_{h+1}^{k^i}) \leq V_{h+1}^{\star}(s_{h+1}^{k^i})$ for $k^i \leq k$ and the event $\mathcal{G}_1$. Now we have proved that $\overline{Q}_h^{k+1}(s,a) \geq Q_h^{\star}(s,a) \geq \underline{Q}_h^{k+1}(s,a)$. Therefore, by noting that

$$\overline{V}_h^{k+1}(s) = \min\left\{H, \max_{a\in A_h^k(s)} \overline{Q}_h^{k+1}(s,a)\right\} \geq \max_{a\in A_h^k(s)} Q_h^{\star}(s,a) = V_h^{\star}(s)$$

and

$$\underline{V}_h^{k+1}(s) = \max\left\{0, \max_{a\in A_h^k(s)} \underline{Q}_h^{k+1}(s,a)\right\} \leq \max_a Q_h^{\star}(s,a) = V_h^{\star}(s),$$

we prove the conclusion for $k+1$ and thus complete the proof. □

**Lemma F.3.** *When event $\mathcal{G}_1$ in Lemma F.1 happens, for any $(h,k) \in [H] \times [K]$, we have that:*

$$\overline{V}_h^k(s_h^k) - \underline{V}_h^k(s_h^k) \leq \overline{Q}_h^k(s_h^k, a_h^k) - \underline{Q}_h^k(s_h^k, a_h^k).$$

*Proof.* If $|A_h^k(s_h^k)| = 1$, based on the definition of $A_h^k(s_h^k)$, we have

$$\overline{V}_h^k(s_h^k) \leq \max_{a\in A_h^{k-1}(s_h^k)} \overline{Q}_h^k(s_h^k, a) = \overline{Q}_h^k(s_h^k, a_h^k).$$

and

$$\underline{V}_h^k(s_h^k) \geq \max_{a\in A_h^{k-1}(s_h^k)} \underline{Q}_h^k(s_h^k, a) = \underline{Q}_h^k(s_h^k, a_h^k).$$

Therefore, we prove the conclusion. If $|A_h^k(s_h^k)| > 1$, define:

$$\hat{a} = \operatorname*{arg\,max}_{a\in A_h^{k-1}(s_h^k)} \overline{Q}_h^k(s_h^k, a), \quad \tilde{a} = \operatorname*{arg\,max}_{a\in A_h^{k-1}(s_h^k)} \underline{Q}_h^k(s_h^k, a)$$

Then we have

$$\begin{aligned}
\overline{V}_h^k(s_h^k) - \underline{V}_h^k(s_h^k) &\leq \overline{Q}_h^k(s_h^k, \hat{a}) - \underline{Q}_h^k(s_h^k, \tilde{a}) \\
&= \left(\overline{Q}_h^k(s_h^k, \hat{a}) - \underline{Q}_h^k(s_h^k, \hat{a})\right) + \left(\underline{Q}_h^k(s_h^k, \hat{a}) - \underline{Q}_h^k(s_h^k, \tilde{a})\right) \\
&\leq \overline{Q}_h^k(s_h^k, a_h^k) - \underline{Q}_h^k(s_h^k, a_h^k).
\end{aligned}$$

The last inequality is because

$$a_h^k = \operatorname*{arg\,max}_{a\in A_h^k(s)} \overline{Q}_h^k(s_h^k, a) - \underline{Q}_h^k(s_h^k, a)$$

when $|A_h^k(s_h^k)| > 1$ and $\underline{Q}_h^k(s_h^k, \hat{a}) \leq \underline{Q}_h^k(s_h^k, \tilde{a})$ based on the definition of $\tilde{a}$. □

**Lemma F.4.** *When event $\mathcal{G}_1$ in Lemma F.1 happens, for any $(h,k) \in [H] \times [K]$, we have that:*

$$\overline{Q}_h^k(s_h^k, a_h^k) - \underline{Q}_h^k(s_h^k, a_h^k) \geq \frac{\Delta_h(s_h^k, a_h^k)}{2}.$$

*Proof.* If $|A_h^k(s_h^k)| = 1$, based on the definition of $A_h^k(s_h^k)$, we have

$$V_h^{\star}(s_h^k) \leq \overline{V}_h^k(s_h^k) \leq \max_{a\in A_h^{k-1}(s_h^k)} \overline{Q}_h^k(s_h^k, a) = \overline{Q}_h^k(s_h^k, a_h^k).$$

Combining the result with $\underline{Q}_h^k(s_h^k, a_h^k) \leq Q_h^{\star}(s_h^k, a_h^k)$ by Lemma F.2, it holds that:

$$\overline{Q}_h^k(s_h^k, a_h^k) - \underline{Q}_h^k(s_h^k, a_h^k) \geq V_h^{\star}(s_h^k) - Q_h^{\star}(s_h^k, a_h^k) = \Delta_h(s_h^k, a_h^k).$$

Therefore, we prove the conclusion. If $|A_h^k(s_h^k)| > 1$, define:

$$\hat{a} = \operatorname*{arg\,max}_{a\in A_h^{k-1}(s_h^k)} \overline{Q}_h^k(s_h^k, a).$$

Then the conclusion follows from the following analysis:

$$
\begin{aligned}
&\Delta_h(s_h^k, a_h^k) \\
&= V_h^\star(s_h^k) - Q_h^\star(s_h^k, a_h^k) \\
&\leq \overline{V}_h^k(s_h^k) - \underline{Q}_h^k(s_h^k, a_h^k) \tag{36} \\
&= \overline{Q}_h^k(s_h^k, \hat{a}) - \underline{Q}_h^k(s_h^k, a_h^k) \\
&= \left(\overline{Q}_h^k(s_h^k, \hat{a}) - \underline{Q}_h^k(s_h^k, \hat{a})\right) + \left(\underline{Q}_h^k(s_h^k, \hat{a}) - \overline{Q}_h^k(s_h^k, a_h^k)\right) + \left(\overline{Q}_h^k(s_h^k, a_h^k) - \underline{Q}_h^k(s_h^k, a_h^k)\right) \tag{37} \\
&\leq 2\left(\overline{Q}_h^k(s_h^k, a_h^k) - \underline{Q}_h^k(s_h^k, a_h^k)\right).
\end{aligned}
$$

Here, Equation (36) is because of $\overline{V}_h^k(s_h^k) \geq V_h^\star(s_h^k)$ and $\underline{Q}_h^k(s_h^k, a_h^k) \leq Q_h^\star(s_h^k, a_h^k)$ by event $\mathcal{G}_1$ of Lemma F.2. Equation (37) is because

$$
a_h^k = \underset{a \in A_h^k(s)}{\arg\max}\, \overline{Q}_h^k(s_h^k, a) - \underline{Q}_h^k(s_h^k, a)
$$

when $|A_h^k(s_h^k)| > 1$ and $\underline{Q}_h^k(s_h^k, \hat{a}) \leq \underline{V}_h^k(s_h^k) \leq \overline{Q}_h^k(s_h^k, a_h^k)$ since $a_h^k \in A_h^k(s_h^k)$. $\qquad\square$

## F.2 Proof of Theorem 4.1

In this section, we bound the worst-case regret under the event $\mathcal{G}_1 \cap \mathcal{G}_2$ in Lemma F.1.

For $h \in [H+1]$, denote:

$$
\delta_h^k = \left(\overline{V}_h^k - V_h^\star\right)(s_h^k), \quad \zeta_h^k = \left(\overline{V}_h^k - V_h^{\pi^k}\right)(s_h^k).
$$

Here, $\delta_{H+1}^k = \zeta_{H+1}^k = 0$. Because $V_h^\star(s) = \sup_\pi V_h^\pi(s)$, we have $\delta_h^k \leq \zeta_h^k$ for any $h \in [H+1]$. In addition, as $\overline{V}_h^k(s) \geq V_h^\star(s)$ for all $(s, h, k) \in \mathcal{S} \times [H] \times [K]$ by Lemma F.2, we have:

$$
\text{Regret}(T) = \sum_{k=1}^K \left(V_1^\star(s_1^k) - V_1^{\pi^k}(s_1^k)\right) \leq \sum_{k=1}^K \left(\overline{V}_1^k(s_1^k) - V_1^{\pi^k}(s_1^k)\right) = \sum_{k=1}^K \zeta_1^k.
$$

Thus, we only need to bound $\sum_{k=1}^K \zeta_1^k$. Noting that

$$
\begin{aligned}
\sum_{k=1}^K \zeta_h^k &\leq \sum_{k=1}^K (\overline{Q}_h^k - Q_h^{\pi^k})(s_h^k, a_h^k) \\
&= \sum_{k=1}^K (\overline{Q}_h^k - Q_h^\star)(s_h^k, a_h^k) + \sum_{k=1}^K (Q_h^\star - Q_h^{\pi^k})(s_h^k, a_h^k) \\
&= \sum_{k=1}^K (\overline{Q}_h^k - Q_h^\star)(s_h^k, a_h^k) + \sum_{k=1}^K \mathbb{P}_{s_h^k, a_h^k, h}\left(V_{h+1}^\star - V_{h+1}^{\pi^k}\right). \tag{38}
\end{aligned}
$$

In the last inequality, we use the Bellman Equation (Equation (2)):

$$
Q_h^\star(s, a) = r_h(s, a) + \mathbb{P}_{s, a, h} V_{h+1}^\star, \quad Q_h^{\pi^k}(s, a) = r_h(s, a) + \mathbb{P}_{s, a, h} V_{h+1}^{\pi^k}.
$$

Next, we will bound $\sum_{k=1}^K (\overline{Q}_h^k - Q_h^\star)(s_h^k, a_h^k)$. Using Bellman Optimality Equation, we know

$$
\begin{aligned}
(\overline{Q}_h^k - Q_h^\star)(s_h^k, a_h^k) &\leq \eta_0^{N_h^k} H + \sum_{i=1}^{N_h^k} \eta_i^{N_h^k}\left(\overline{V}_{h+1}^{k^i}(s_{h+1}^i) - \mathbb{P}_{s_h^k, a_h^k, h} V_{h+1}^\star\right) + \sum_{i=1}^{N_h^k} \eta_i^{N_h^k} b_i \\
&\leq \eta_0^{N_h^k} H + \sum_{i=1}^{N_h^k} \eta_i^{N_h^k}\left(\overline{V}_{h+1}^{k^i}(s_{h+1}^i) - \mathbb{P}_{s_h^k, a_h^k, h} V_{h+1}^\star\right) + 4\sqrt{\frac{H^3 \iota}{N_h^k}} \\
&\leq \eta_0^{N_h^k} H + \sum_{i=1}^{N_h^k} \eta_i^{N_h^k}\left(\overline{V}_{h+1}^{k^i} - V_{h+1}^\star\right)(s_{h+1}^i) + 6\sqrt{\frac{H^3 \iota}{N_h^k}}. \tag{39}
\end{aligned}
$$

The first inequality uses:

$$\sum_{i=1}^{N_h^k} \eta_i^{N_h^k} b_i = 2 \sum_{i=1}^{N_h^k} \eta_i^{N_h^k} \sqrt{\frac{H^3 \iota}{i}} \le 4 \sqrt{\frac{H^3 \iota}{N_h^k}}.$$

by Lemma D.2. The last inequality is by the event $\mathcal{G}_1$ in Lemma F.1. By summing Equation (39) over $k \in [K]$, we reach

$$\sum_{k=1}^{K} (\overline{Q}_h^k - Q_h^\star)(s_h^k, a_h^k)$$

$$\le \sum_{k=1}^{K} \eta_0^{N_h^k} H + \sum_{k=1, N_h^k>0}^{K} \sum_{i=1}^{N_h^k} \eta_i^{N_h^k} (\overline{V}_{h+1}^{k^i} - V_{h+1}^\star)(s_{h+1}^{k^i}) + \sum_{k=1, N_h^k>0}^{K} 6 \sqrt{\frac{H^3 \iota}{N_h^k}}. \qquad (40)$$

For the first term of Equation (40), we have:

$$\sum_{k=1}^{K} \eta_0^{N_h^k} H = H \sum_{s,a} \sum_{k=1}^{K} \mathbb{I}[N_h^k = 0, (s_h^k, a_h^k) = (s, a)] \le HSA. \qquad (41)$$

For the second term of Equation (40), by Lemma D.4, it holds that:

$$\sum_{k=1, N_h^k>0}^{K} 6 \sqrt{\frac{H^3 \iota}{N_h^k}} \le 12 \sqrt{H^2 SAT\iota}. \qquad (42)$$

For the last term of Equation (40), similar to proof of Equation (4.7) in Jin et al. (2018), it holds that:

$$\sum_{k=1, N_h^k>0}^{K} \sum_{i=1}^{N_h^k} \eta_i^{N_h^k} (\overline{V}_{h+1}^{k^i} - V_{h+1}^\star)(s_{h+1}^{k^i}) \le \left(1 + \frac{1}{H}\right) \sum_{k=1}^{K} \delta_{h+1}^k. \qquad (43)$$

Taking the above results Equation (41), Equation (42) and Equation (43) together with Equation (40), back to Equation (38) to reach

$$\sum_{k=1}^{K} \zeta_h^k \le \left(1 + \frac{1}{H}\right) \sum_{k=1}^{K} \delta_{h+1}^k + 12 \sqrt{H^2 SAT\iota} + HSA + \sum_{k=1}^{K} \mathbb{P}_{s_h^k, a_h^k, h} \left(V_{h+1}^\star - V_{h+1}^{\pi^k}\right)$$

$$\le \left(1 + \frac{1}{H}\right) \sum_{k=1}^{K} \zeta_{h+1}^k + 12 \sqrt{H^2 SAT\iota} + HSA + \sum_{k=1}^{K} \left(\mathbb{P}_{s_h^k, a_h^k, h} - \mathbb{1}_{s_{h+1}^k}\right) \left(V_{h+1}^\star - V_{h+1}^{\pi^k}\right).$$

By recursion on $h$, since $\zeta_{H+1}^k = 0$, we can get the following conclusion:

$$\text{Regret}(T) \le \sum_{k=1}^{K} \zeta_1^k$$

$$\le O\left(\sqrt{H^4 SAT\iota} + H^2 SA + \sum_{h=1}^{H} \sum_{k=1}^{K} \left(1 + \frac{1}{H}\right)^{h-1} \left(\mathbb{P}_{s_h^k, a_h^k, h} - \mathbb{1}_{s_{h+1}^k}\right) \left(V_{h+1}^\star - V_{h+1}^{\pi^k}\right)\right)$$

$$\le O\left(\sqrt{H^4 SAT\iota} + H^2 SA\right).$$

The last inequality is because of the event $\mathcal{G}_2$ in Lemma F.1. We note that when $T \ge \sqrt{H^4 SAT\iota}$, we have $\sqrt{H^4 SAT\iota} \ge H^2 SA$, and when $T \le \sqrt{H^4 SAT\iota}$, we have $\sum_{k=1}^{K} \delta_1^k \le HK = T \le \sqrt{H^4 SAT\iota}$. Therefore, we can remove the $H^2 SA$ term in the regret upper bound.

To summarize, with probability at least $1 - 2p$, we have $\text{Regret}(T) \le O(\sqrt{H^4 SAT\iota})$. Rescaling $p$ to $p/2$ finishes the proof.

### F.3 PROOF OF THEOREM 4.2

In this section, we derive the fine-grained gap-dependent regret bound for ULCB-Hoeffding following a similar line of reasoning as in UCB-Hoeffding. Let $p = \frac{1}{T}$, then the event $\mathcal{G}_1$ holds with probability at least $1 - \frac{1}{T}$ and $\iota \leq O(\log(SAT))$. Therefore, by Lemma E.2, we have

$$\mathbb{E}(\text{Regret}(T))$$

$$= \mathbb{E}\left(\sum_{h=1}^{H}\sum_{s,a}\Delta_h(s,a)N_h^{K+1}(s,a)\right)$$

$$= \mathbb{E}\left(\sum_{h=1}^{H}\sum_{s,a}\Delta_h(s,a)N_h^{K+1}(s,a)\Big|\mathcal{G}_1\right)\mathbb{P}(\mathcal{G}_1) + \mathbb{E}\left(\sum_{h=1}^{H}\sum_{s,a}\Delta_h(s,a)N_h^{K+1}(s,a)\Big|\mathcal{G}_1^c\right)\mathbb{P}(\mathcal{G}_1^c)$$

$$\leq O\left(\sum_{h=1}^{H}\sum_{\Delta_h(s,a)>0}\frac{H^5\log(SAT)}{\Delta_h(s,a)} + \sum_{h=1}^{H}\frac{H^3\left(\sum_{t=h+1}^{H}\sqrt{|Z_{\text{opt},t}|}\right)^2\log(SAT)}{\Delta_{\min,h}} + SAH^3\right).$$

The last inequality is because under the event $\mathcal{G}_1$, by Lemma F.7, we have

$$\sum_{h=1}^{H}\sum_{s,a}\Delta_h(s,a)N_h^{K+1}(s,a)$$

$$\leq O\left(\sum_{h=1}^{H}\sum_{\Delta_h(s,a)>0}\frac{H^5\log(SAT)}{\Delta_h(s,a)} + \sum_{h=1}^{H}\frac{H^3\left(\sum_{t=h+1}^{H}\sqrt{|Z_{\text{opt},t}|}\right)^2\log(SAT)}{\Delta_{\min,h}} + SAH^3\right).$$

and under the event $\mathcal{G}_1^c$,

$$\sum_{h=1}^{H}\sum_{s,a}\Delta_h(s,a)N_h^{K+1}(s,a) \leq HT.$$

Now we only need to prove Lemma F.7. Using the same fine-grained analytical framework, we first bound the cumulative weighted estimation error for each state-action pair $(s,a)$ at step $h$.

**Lemma F.5.** *For the ULCB-Hoeffding algorithm (Algorithm 2), under the event $\mathcal{G}_1$ in Lemma F.1, for any non-negative weight sequence $\{\omega_h^k\}_k$ at step $h$, it holds simultaneously for any $(s,a) \in \mathcal{S} \times \mathcal{A}$ and subsequent step $h' \in [h, H]$ that:*

$$\sum_{k=1}^{K}\omega_h(k,h',s,a)\left(\overline{Q}_{h'}^k - \underline{Q}_{h'}^k\right)(s_{h'}^k, a_{h'}^k) \leq \sum_{k'=1}^{K}\tilde{\omega}_h(k',h'+1,s,a)\left(\overline{Q}_{h'+1}^{k'} - \underline{Q}_{h'+1}^{k'}\right)(s,a)_{h'+1}^{k'}$$

$$+ \|\omega_h(\cdot,h')\|_\infty H + 16\sqrt{H^3\|\omega_h(\cdot,h')\|_\infty\|\omega_h(\cdot,h',s,a)\|_1\iota}. \tag{44}$$

*and*

$$\sum_{k=1}^{K}\omega_h(k,h')\left(\overline{Q}_{h'}^k - \underline{Q}_{h'}^k\right)(s_{h'}^k, a_{h'}^k) \leq \sum_{k'=1}^{K}\omega_h(k',h'+1)\left(\overline{Q}_{h'+1}^{k'} - \underline{Q}_{h'+1}^{k'}\right)(s_{h'+1}^{k'}, a_{h'+1}^{k'})$$

$$+ \|\omega_h(\cdot,h')\|_\infty SAH + 16\sum_{s,a}\sqrt{H^3\|\omega_h(\cdot,h')\|_\infty\|\omega_h(\cdot,h',s,a)\|_1\iota}. \tag{45}$$

*Proof.* Under the event $\mathcal{G}_1$ in Lemma F.1, by Equation (34) and Equation (35), we have

$$\overline{Q}_{h'}^k(s_{h'}^k, a_{h'}^k) = \eta_0^{N_{h'}^k}H + \sum_{i=1}^{N_{h'}^k}\eta_i^{N_{h'}^k}\left(r_{h'}(s_{h'}^k, a_{h'}^k) + \overline{V}_{h'+1}^{k^i}(s_{h'+1}^{k^i}) + b_i\right)$$

$$\leq \eta_0^{N_{h'}^k}H + \sum_{i=1}^{N_{h'}^k}\eta_i^{N_{h'}^k}\left(r_{h'}(s_{h'}^k, a_{h'}^k) + \overline{V}_{h'+1}^{k^i}(s_{h'+1}^{k^i})\right) + 4\sqrt{\frac{H^3\iota}{N_{h'}^k}}, \tag{46}$$

and

$$\underline{Q}_{h'}^k(s_{h'}^k, a_{h'}^k) = \sum_{i=1}^{N_{h'}^k} \eta_i^{N_{h'}^k} \left( r_{h'}(s_{h'}^k, a_{h'}^k) + \underline{V}_{h'+1}^{k^i}(s_{h'+1}^{k^i}) - b_i \right).$$

$$\geq \sum_{i=1}^{N_{h'}^k} \eta_i^{N_{h'}^k} \left( r_{h'}(s_{h'}^k, a_{h'}^k) + \underline{V}_{h'+1}^{k^i}(s_{h'+1}^{k^i}) \right) - 4\sqrt{\frac{H^3\iota}{N_{h'}^k}}. \tag{47}$$

These two inequalities are because

$$\sum_{i=1}^{N_{h'}^k} \eta_i^{N_{h'}^k} b_i = 2 \sum_{i=1}^{N_{h'}^k} \eta_i^{N_{h'}^k} \sqrt{\frac{H^3\iota}{i}} \leq 4\sqrt{\frac{H^3\iota}{N_{h'}^k}}$$

by (c) of Lemma D.2. Therefore, by taking the difference between Equation (46) and Equation (47), we reach

$$\sum_{k=1}^K \omega_h(k, h', s, a) \left( \overline{Q}_{h'}^k - \underline{Q}_{h'}^k \right)(s_{h'}^k, a_{h'}^k) \leq \sum_{k=1}^K \omega_h(k, h', s, a) \eta_0^{N_{h'}^k} H$$

$$+ \sum_{k=1}^K \omega_h(k, h', s, a) \sum_{i=1}^{N_{h'}^k} \eta_i^{N_{h'}^k} \left( \overline{V}_{h'+1}^{k^i} - \underline{V}_{h'+1}^{k^i} \right)(s_{h'+1}^{k^i}) + \sum_{k=1}^K \omega_h(k, h', s, a) \beta_{N_{h'}^k}. \tag{48}$$

Same as Equation (20) and Equation (22), we have

$$\sum_{k=1}^K \omega_h(k, h', s, a) \eta_0^{N_{h'}^k} H \leq \|\omega_h(\cdot, h')\|_\infty H.$$

and

$$\sum_{k=1}^K \omega_h(k, h', s, a) \beta_{N_{h'}^k} \leq 16\sqrt{H^3 \|\omega_h(\cdot, h')\|_\infty \|\omega_h(\cdot, h', s, a)\|_1 \iota}.$$

For the second term in Equation (48), similar to Equation (21), we have

$$\sum_{k=1}^K \omega_h(k, h', s, a) \sum_{i=1}^{N_{h'}^k} \eta_i^{N_{h'}^k} \left( \overline{V}_{h'+1}^{k^i} - \underline{V}_{h'+1}^{k^i} \right)(s_{h'+1}^{k^i})$$

$$= \sum_{k=1}^K \omega_h(k, h') \mathbb{I}\left[ (s_{h'}^k, a_{h'}^k) = (s, a) \right] \sum_{i=1}^{N_{h'}^k} \eta_i^{N_{h'}^k} \left( \overline{V}_{h'+1}^{k^i} - \underline{V}_{h'+1}^{k^i} \right)(s_{h'+1}^{k^i}) \left( \sum_{k'=1}^K \mathbb{I}[k^i = k'] \right)$$

$$= \sum_{k'=1}^K \left( \overline{V}_{h'+1}^{k'} - \underline{V}_{h'+1}^{k'} \right)(s_{h'+1}^{k'}) \left( \sum_{k=1}^K \sum_{i=1}^{N_{h'}^k} \omega_h(k, h') \mathbb{I}\left[ (s_{h'}^k, a_{h'}^k) = (s, a) \right] \eta_i^{N_{h'}^k} \mathbb{I}[k^i = k'] \right)$$

$$\leq \sum_{k'=1}^K \left( \overline{Q}_{h'+1}^{k'} - \underline{Q}_{h'+1}^{k'} \right)(s_{h'+1}^{k'}, a_{h'+1}^{k'}) \left( \sum_{k=1}^K \sum_{i=1}^{N_{h'}^k} \omega_h(k, h') \eta_i^{N_{h'}^k} \mathbb{I}\left[ k^i = k', (s_{h'}^k, a_{h'}^k) = (s, a) \right] \right)$$

$$= \sum_{k'=1}^K \tilde{\omega}_h(k', h'+1, s, a) \left( \overline{Q}_{h'+1}^{k'} - \underline{Q}_{h'+1}^{k'} \right)(s_{h'+1}^{k'}, a_{h'+1}^{k'}).$$

The inequality follows from Lemma F.3. Combining the upper bounds for each term in Equation (48), we finish the proof of Equation (44). Summing this conclusion over all state-action pairs $(s, a)$, and noting that $\sum_{s,a} \tilde{\omega}_h(k', h'+1, s, a) = \omega_h(k', h'+1)$, we prove Equation (45). □

Building on the lemma above, we can establish the following result. The proof follows the same argument as in Lemma E.4.

**Lemma F.6.** *For the ULCB-Hoeffding algorithm (Algorithm 2), under the event $\mathcal{G}_1$ in Lemma F.1, for any non-negative weight sequence $\{\omega_h^k\}_k$ at step $h$, it holds simultaneously for any $(s, a) \in \mathcal{S} \times \mathcal{A}$ and subsequent step $h' \in [h, H]$ that:*

$$\sum_{k=1}^K \omega_h(k, h') \left(\overline{Q}_{h'}^k - \underline{Q}_{h'}^k\right)(s_{h'}^k, a_{h'}^k)$$

$$\leq \sum_{h_1=h'}^H \|\omega_h(\cdot, h_1)\|_\infty SAH + 16 \sum_{h_1=h'}^H \sum_{s,a} \sqrt{H^3 \|\omega_h(\cdot, h_1)\|_\infty \|\omega_h(\cdot, h_1, s, a)\|_1 \iota}.$$

The following lemma bounds the summation $\sum_{s,a} \Delta_h(s, a) N_h^{K+1}(s, a)$, which directly contributes to bounding the expected regret via Lemma E.2. The proof largely mirrors that of Lemma E.5, with only minor differences; we therefore focus on the distinctions and omit the unchanged parts.

**Lemma F.7.** *For the ULCB-Hoeffding algorithm (Algorithm 2), under the event $\mathcal{G}_1$ in Lemma F.1, it holds for any $h \in [H]$ and $c_2 = 82944$ that:*

$$\sum_{s,a} \frac{\Delta_h(s, a) N_h^{K+1}(s, a)}{c_2} \leq SAH^2 + \sum_{h'=h}^H \sum_{\Delta_{h'}(s,a)>0} \frac{H^4 \iota}{\Delta_{h'}(s, a)} + \frac{H^3 \left(\sum_{t=h+1}^H \sqrt{|Z_{\mathrm{opt},t}|}\right)^2 \iota}{\Delta_{\min,h}}$$

$$+ \sum_{h'=h+1}^H \frac{H^2 \left(\sum_{t=h'+1}^H \sqrt{|Z_{\mathrm{opt},t}|}\right)^2 \iota}{\Delta_{\min,h'}}.$$

*Proof.* We use mathematical induction to prove this conclusion. For step $h$, let

$$\omega_h^k = \mathbb{I}\left[\overline{Q}_h^k(s_h^k, a_h^k) - \underline{Q}_h^k(s_h^k, a_h^k) \geq \frac{\Delta_h(s_h^k, a_h^k)}{2}, (s_h^k, a_h^k) \in Z_{\mathrm{sub},h}\right]$$

$$= \mathbb{I}\left[(s_h^k, a_h^k) \in Z_{\mathrm{sub},h}\right] \leq 1.$$

The second equation is by Lemma F.4. Based on the definition of $\omega_h^k$, for any $(s, a) \in Z_{\mathrm{sub},h}$,

$$\|\omega_h(\cdot, h, s, a)\|_1 = \sum_{k=1}^K \mathbb{I}\left[(s_h^k, a_h^k) = (s, a)\right] = N_h^{K+1}(s, a)$$

and $\|\omega_h(\cdot, h, s, a)\|_1 = 0$ for $(s, a) \in Z_{\mathrm{opt},h}$. By Lemma F.5, for any $(s, a) \in Z_{\mathrm{sub},h}$, it holds that,

$$\sum_{k=1}^K \omega_h^k \left(\overline{Q}_h^k - \underline{Q}_h^k\right)(s_h^k, a_h^k) \mathbb{I}[(s_h^k, a_h^k) = (s, a)] = \sum_{k=1}^K \omega_h(k, h, s, a) \left(\overline{Q}_h^k - \underline{Q}_h^k\right)(s_h^k, a_h^k)$$

$$\leq H + 16\sqrt{H^3 N_h^{K+1}(s, a) \iota} + \sum_{k'=1}^K \tilde{\omega}_h(k', h+1, s, a) \left(\overline{Q}_{h+1}^{k'} - \underline{Q}_{h+1}^{k'}\right)(s_{h+1}^{k'}, a_{h+1}^{k'}). \quad (49)$$

Also note that for any $(s, a) \in Z_{\mathrm{sub},h}$ with $\Delta_h(s, a) > 0$, we have

$$\sum_{k=1}^K \omega_h^k \left(\overline{Q}_h^k - \underline{Q}_h^\star\right)(s_h^k, a_h^k) \mathbb{I}[(s_h^k, a_h^k) = (s, a)]$$

$$\geq \frac{\Delta_h(s, a)}{2} \sum_{k=1}^K \omega_h^k \mathbb{I}[(s_h^k, a_h^k) = (s, a)] = \frac{\Delta_h(s, a) N_h^{K+1}(s, a)}{2}. \quad (50)$$

Combining the results of Equation (49) and Equation (50), it holds for any $(s, a) \in Z_{\mathrm{sub},h}$ that,

$$\frac{\Delta_h(s, a) N_h^{K+1}(s, a)}{2}$$

$$\leq H + 16\sqrt{H^3 N_h^{K+1}(s, a) \iota} + \sum_{k'=1}^K \tilde{\omega}_h(k', h+1, s, a)(\overline{Q}_{h+1}^{k'} - \underline{Q}_{h+1}^{k'})(s_{h+1}^{k'}, a_{h+1}^{k'}).$$

Solving this inequality, we can derive the following conclusion for any $(s, a) \in Z_{\text{sub},h}$:

$$\Delta_h(s,a)N_h^{K+1}(s,a) \leq \frac{1024H^3\iota}{\Delta_h(s,a)} + 4H + 4\sum_{k'=1}^{K} \tilde{\omega}_h(k', h+1, s, a)(\overline{Q}_{h+1}^{k'} - \underline{Q}_{h+1}^{k'})(s_{h+1}^{k'}, a_{h+1}^{k'}).$$

Since $\Delta_h(s,a) = 0$ for $(s,a) \notin Z_{\text{sub},h}$ and $\overline{Q}_{h+1}^k(s,a) \geq \underline{Q}_{h+1}^k(s,a)$ for any $(s,a,h,k) \in \mathcal{S} \times \mathcal{A} \times [H] \times [K]$, by summing the inequality above over all state-action pairs $(s,a) \in Z_{\text{sub},h}$, we reach:

$$\sum_{s,a} \Delta_h(s,a)N_h^{K+1}(s,a)$$

$$\leq \sum_{\Delta_h(s,a)>0} \frac{1024H^3\iota}{\Delta_h(s,a)} + 4SAH + 4\sum_{k'=1}^{K} \omega_h(k', h+1)(\overline{Q}_{h+1}^{k'} - \underline{Q}_{h+1}^{k'})(s_{h+1}^{k'}, a_{h+1}^{k'}). \tag{51}$$

Let $h = H$. Since $Q_{H+1}^k(s,a) = Q_{H+1}^\star(s,a) = 0$ for all $(s,a,k) \in \mathcal{S} \times \mathcal{A} \times [K]$, the base case $h = H$ follows immediately from Equation (51).

Now, assume the lemma holds for steps $h+1, \ldots, H$. Using the same inductive argument as in Lemma E.5, we prove the case for step $h$. From Lemma F.6, we have:

$$\sum_{k'=1}^{K} \omega_h(k', h+1)(\overline{Q}_{h+1}^{k'} - \underline{Q}_{h+1}^{k'})(s_{h+1}^{k'}, a_{h+1}^{k'})$$

$$\leq \sum_{h'=h+1}^{H} \|\omega_h(\cdot, h')\|_\infty SAH + 16\sum_{h'=h+1}^{H}\sum_{s,a} \sqrt{H^3\|\omega_h(\cdot,h')\|_\infty\|\omega_h(\cdot,h',s,a)\|_1\iota}. \tag{52}$$

Similar to the proof of Equation (32), we can derive that

$$16\sum_{h'=h+1}^{H}\sum_{s,a} \sqrt{H^3\|\omega_h(\cdot,h')\|_\infty\|\omega_h(\cdot,h',s,a)\|_1\iota}$$

$$\leq 24\sqrt{c_2}\left(SAH^2 + 2\sum_{h'=h+1}^{H}\sum_{\Delta_{h'}(s,a)>0} \frac{H^4\iota}{\Delta_{h'}(s,a)} + 2\sum_{h'=h+1}^{H} \frac{H^2\left(\sum_{t=h'+1}^{H}\sqrt{|Z_{\text{opt},t}|}\right)^2\iota}{\Delta_{\min,h'}}\right)$$

$$+ 16\sqrt{3}\left(\sum_{h'=h+1}^{H}\sqrt{H^3|Z_{\text{opt},h'}|\iota}\right)\sqrt{\sum_{\Delta_h(s,a)>0} N_h^{K+1}(s,a)}.$$

By applying this inequality to Equation (52) and substituting the result into Equation (51), and using the bound $\|\omega_h(\cdot, h')\|_\infty \leq 3$ from Equation (27), we conclude that the following inequality holds:

$$\sum_{s,a} \Delta_h(s,a)N_h^{K+1}(s,a)$$

$$\leq 192\sqrt{c_1}\left(SAH^2 + \sum_{h'=h}^{H}\sum_{\Delta_{h'}(s,a)>0} \frac{H^4\iota}{\Delta_{h'}(s,a)} + \sum_{h'=h+1}^{H} \frac{H^2\left(\sum_{t=h'+1}^{H}\sqrt{|Z_{\text{opt},t}|}\right)^2\iota}{\Delta_{\min,h'}}\right)$$

$$+ 64\sqrt{3}\left(\sum_{h'=h+1}^{H}\sqrt{H^3|Z_{\text{opt},h'}|\iota}\right)\sqrt{\sum_{\Delta_h(s,a)>0} N_h^{K+1}(s,a)}.$$

By applying the same method used to solve Equation (33), we can similarly establish that

$$\sum_{s,a} \Delta_h(s,a)N_h^{K+1}(s,a) \leq 18432\frac{H^3\iota\left(\sum_{t=h+1}^{H}\sqrt{|Z_{\text{opt},t}|}\right)^2}{\Delta_{\min,h}}$$

$$+ 288\sqrt{c_2}\left(SAH^2 + \sum_{h'=h}^{H}\sum_{\Delta_{h'}(s,a)>0} \frac{H^4\iota}{\Delta_{h'}(s,a)} + \sum_{h'=h+1}^{H} \frac{H^2\left(\sum_{t=h'+1}^{H}\sqrt{|Z_{\text{opt},t}|}\right)^2\iota}{\Delta_{\min,h'}}\right).$$

This establishes the result for step $h$, thereby completing the proof. $\qquad\square$

# G PROOF OF FINE-GRAINED GAP-DEPENDENT REGRET BOUND FOR AMB

## G.1 REVIEW OF AMB ALGORITHM

We first review the AMB algorithm (Xu et al., 2021) in Algorithm 3.

---

**Algorithm 3** Adaptive Multi-step Bootstrap (AMB)

---

1: **Input:** $p \in (0,1)$ (failure probability), $H, A, S, K \geq 1$
2: **Initialization:** For any $\forall (s,a,h) \in \mathcal{S} \times \mathcal{A} \times [H]$, initialize $\overline{Q}_h^1(s,a) \leftarrow H$, $\underline{Q}_h^1(s,a) \leftarrow 0$, $G_h^1 = \emptyset$, $A_h^1(s) \leftarrow \mathcal{A}$ and $\overline{V}_h^1(s) = \underline{V}_h^1(s) = 0$.
3: **for** $k = 1, 2, \ldots, K$ **do**
4:     **Step 1: Collect data:**
5:     Rollout from a random initial state $s_1^k \sim \mu$ using policy $\pi_k = \{\pi_h^k\}_{h=1}^H$, defined as:

$$\pi_h^k(s) \triangleq \begin{cases} \arg\max_{a \in A_h^k(s)} \overline{Q}_h^k(s,a) - \underline{Q}_h^k(s,a), & \text{if } |A_h^k(s)| > 1 \\ \text{the element in } A_h^k(s), & \text{if } |A_h^k(s)| = 1 \end{cases}$$

6:     and obtain an episode $\left\{ (s_h^k, a_h^k, r_h^k = r_h(s_h^k, a_h^k)) \right\}_{h=1}^H$.
7:     **Step 2: Update $Q$-function:**
8:     **for** $h = H, H-1, \ldots, 1$ **do**
9:         **if** $s_h^k \notin G_h^k$ **then**
10:             Let $n = N_h^{k+1}(s,a)$ be the number of visits to $(s,a)$ at step $h$ in the first $k$ episodes.
11:             Let $h' = h'(k,h)$ be the first index after step $h$ in episode $k$ such that $s_{h'}^k \notin G_{h'}^k$. (If such a state does not exist, set $h' = H + 1$ and $\overline{V}_{H+1}^k = \underline{V}_{H+1}^k(s) = 0$.)
12:             Compute $b_n' = 4\sqrt{H^3 \log(2SAT/p)/n}$ and $\hat{Q}_h^{k,d}(s,a) = \sum_{h \leq i < h'} r_i^k$.
13:             $\overline{Q}_h^{k+1}(s_h^k, a_h^k) = \min\left\{ H, (1-\eta_n)\overline{Q}_h^k(s_h^k, a_h^k) + \eta_n\left(\hat{Q}_h^{k,d}(s_h^k, a_h^k) + \overline{V}_{h'}^k(s_{h'}^k) + b_n'\right) \right\}$.
14:             $\underline{Q}_h^{k+1}(s_h^k, a_h^k) = \max\left\{ 0, (1-\eta_n)\underline{Q}_h^k(s_h^k, a_h^k) + \eta_n\left(\hat{Q}_h^{k,d}(s_h^k, a_h^k) + \underline{V}_{h'}^k(s_{h'}^k) - b_n'\right) \right\}$.
15:             $\overline{V}_h^{k+1}(s_h^k) = \max_{a' \in A_h^k(s_h^k)} \overline{Q}_h^{k+1}(s_h^k, a')$.
16:             $\underline{V}_h^{k+1}(s_h^k) = \max_{a' \in A_h^k(s_h^k)} \underline{Q}_h^{k+1}(s_h^k, a')$.
17:         **end if**
18:     **end for**
19:     **for** $(s,a,h) \in \mathcal{S} \times \mathcal{A} \times [H] \setminus \{(s_h^k, a_h^k) | 1 \leq h \leq H, s_h^k \notin G_h^k\}_{h=1}^H$ **do**
20:         $\overline{Q}_h^{k+1}(s,a) = \overline{Q}_h^k(s,a), \underline{Q}_h^{k+1}(s,a) = \underline{Q}_h^k(s,a), \overline{V}_h^{k+1}(s) = \overline{V}_h^k(s), \underline{V}_h^{k+1}(s) = \underline{V}_h^k(s)$.
21:     **end for**
22:     **Step 3: Eliminate the sub-optimal actions:**
23:     $\forall s \in \mathcal{S}, h \in [H]$, set $A_h^{k+1}(s) = \left\{ a \in A_h^k(s) : \overline{Q}_h^k(s,a) \geq \underline{V}_h^k(s) \right\}$.
24:     Set $G_h^{k+1} = \{s \in \mathcal{S} : |A_h^{k+1}(s)| = 1\}$.
25: **end for**

---

AMB maintains upper and lower bounds $\overline{Q}_h^k(s,a)$ and $\underline{Q}_h^k(s,a)$ for each state-action-step triple $(s,a,h)$ at the beginning of episode $k$. The policy $\pi^k$ is selected by maximizing the confidence interval length $\overline{Q} - \underline{Q}$. Based on these bounds, for each state $s$ and step $h$, AMB constructs a set of candidate optimal actions, denoted by $A_h^k(s)$, by eliminating any action $a$ whose upper bound is lower than the lower bound of some other action. If $|A_h^k(s)| = 1$, the optimal action is identified, denoted by $\pi_h^\star(s)$, and $s$ is referred to as a *decided state*; otherwise, $s$ is called an *undecided state*. Let $G_h^k = \{s \mid |A_h^k(s)| = 1\}$ denote the set of all decided states at step $h$ in episode $k$.

Let $\mathcal{F}_{h,k}$ denote the filtration generated by the trajectory up to and including step $h$ in episode $k$. In particular, $\mathcal{F}_{h,k}$ contains the policy $\pi^k$ and the realized state-action pair $(s_h^k, a_h^k)$. AMB constructs upper and lower bounds of the $Q$-function by decomposing the $Q$-function into two parts: the rewards accumulated within the decided states and those from the undecided states. Formally, starting

from state $s_h^k$ at step $h$ and following the policy $\pi^k$, we observe the trajectory $\{(s_{h'}^k, a_{h'}^k, r_{h'}^k)\}_{h'=h}^H$. Let $h' = h'(k,h) > h$ denote the first index such that $s_{h'}^k \notin G_{h'}^k$. Then, the optimal $Q$-value function $Q_h^\star(s,a)$ can be decomposed as:

$$Q_h^{k,d}(s,a) \triangleq \mathbb{E}\left[\sum_{l=h}^{h'-1} r_l(s_l^k, \pi_l^\star(s_l^k)) \mid \mathcal{F}_{h,k}, (s_h^k, a_h^k) = (s,a)\right]$$

and

$$Q_h^{k,ud}(s,a) \triangleq \mathbb{E}\left[V_{h'}^\star(s_{h'}^k) \mid \mathcal{F}_{h,k}, (s_h^k, a_h^k) = (s,a)\right],$$

where $Q_h^{k,d}$ and $Q_h^{k,ud}$ represent the contributions from the decided and undecided parts, respectively. To estimate $Q_h^{k,d}(s_h, a_h)$, AMB uses the sum of empirical rewards in episode $k$:

$$\hat{Q}_h^{k,d}(s,a) = \sum_{l=h}^{h'-1} r_l(s_l^k, a_l^k).$$

To estimate $Q_h^{k,ud}(s_h, a_h)$, AMB performs bootstrapping using the existing upper-bound $V$-estimate $\overline{V}_h^k(s_{h'}^k)$. The resulting update rules of the $Q$-estimates are:

$$\overline{Q}_h^{k+1}(s_h^k, a_h^k) = \min\left\{H, (1-\eta_n)\overline{Q}_h^k(s_h^k, a_h^k) + \eta_n\left(\hat{Q}_h^{k,d}(s_h^k, a_h^k) + \overline{V}_{h'}^k(s_{h'}^k) + b_n'\right)\right\}. \quad (53)$$

$$\underline{Q}_h^{k+1}(s_h^k, a_h^k) = \max\left\{0, (1-\eta_n)\underline{Q}_h^k(s_h^k, a_h^k) + \eta_n\left(\hat{Q}_h^{k,d}(s_h^k, a_h^k) + \underline{V}_{h'}^k(s_{h'}^k) - b_n'\right)\right\}. \quad (54)$$

The learning rate $\eta_n = \frac{H+1}{H+n}$, where $n = N_h^{k+1}(s_h^k, a_h^k)$ represents the number of visits to state-action pair $(s_h^k, a_h^k)$ at step $h$ within the first $k$ episodes. By unrolling the recursion in $h$, we obtain:

$$\overline{Q}_h^k(s_h^k, a_h^k) \leq \min\left\{H, \eta_0^{N_h^k} H + \sum_{i=1}^{N_h^k} \eta_i^{N_h^k}\left(\hat{Q}_h^{k^i,d}(s_h^k, a_h^k) + \overline{V}_{h'}^{k^i}(s_{h'}^{k^i}) + b_i'\right)\right\}, \quad (55)$$

$$\overline{Q}_h^k(s_h^k, a_h^k) \geq \max\left\{0, \eta_0^{N_h^k} H + \sum_{i=1}^{N_h^k} \eta_i^{N_h^k}\left(\hat{Q}_h^{k^i,d}(s_h^k, a_h^k) + \underline{V}_{h'}^{k^i}(s_{h'}^{k^i}) - b_i'\right)\right\}. \quad (56)$$

To ensure the optimism of the $Q$-estimates $\overline{Q}$ and the pessimism of $\underline{Q}$, Xu et al. (2021) adopt the equality forms of Equation (55) and Equation (56) in their Equation (A.5). However, **these equalities do not hold under the actual update rules** in Equation (53) and Equation (54), due to the presence of truncations at $H$ and $0$. In fact, only the inequalities in Equation (55) and Equation (56) can be rigorously derived from the updates. This creates a fundamental inconsistency: to establish optimism and pessimism of $Q$-estimates, we require an upper bound on $\overline{Q}$ and a lower bound on $\underline{Q}$, which are the reverse of the inequalities implied by the truncated updates. Therefore, the truncations at $H$ and $0$ in the update rules Equation (53) and Equation (54) in the AMB algorithm are theoretically improper and should be removed to ensure analytical correctness.

Moreover, the bonus term $b_n'$ is derived by bounding the deviation between $\overline{Q}_h^k(s,a)$ and $Q_h^\star(s,a)$. This analysis relies on applying the Azuma–Hoeffding inequality to two martingale difference terms:

$$\sum_{i=1}^{N_h^k} \eta_i^{N_h^k}\left(\hat{Q}_h^{k^i,d}(s_h^k, a_h^k) - Q_h^{k^i,d}(s_h^k, a_h^k)\right) \quad \text{and} \quad \sum_{i=1}^{N_h^k} \eta_i^{N_h^k}\left(V_{h'}^\star(s_{h'}^{k^i}) - Q_h^{k^i,ud}(s_h^k, a_h^k)\right),$$

based on the following assumed decomposition:

$$Q_h^{k,d}(s_h^k, a_h^k) + Q_h^{k,ud}(s_h^k, a_h^k) = Q_h^\star(s_h^k, a_h^k). \quad (57)$$

This decomposition implies that the sum of the estimators $\hat{Q}_h^{k,d}(s,a)$ and $V_{h'}^\star(s_{h'}^{k^i})$ in multi-step bootstrapping forms an unbiased estimate of $Q_h^\star(s,a)$.

However, Xu et al. (2021) incorrectly apply the Azuma–Hoeffding inequality by centering the estimators $\hat{Q}_h^{k,d}(s,a)$ and $\overline{V}_{h'}^k(s_{h'}^{k^i})$ around their **expectations** (see their Equation (4.2) and Lemma

4.1), rather than around their corresponding **conditional expectations** $Q_h^{k,d}(s,a)$ and $Q_h^{k,ud}(s,a)$. Moreover, the unbiasedness of multi-step bootstrapping implied by Equation (57) requires formal justification. These issues compromise the claimed optimism and pessimism properties of the $Q$-estimators, thereby invalidating the corresponding fine-grained regret guarantees.

To address these issues, we introduce the following key modifications:

**(a) Revising update rules.** We move the truncations at $H$ and 0 in Equation (53) and Equation (54) to the corresponding $V$-estimates (lines 15–16 in Algorithm 4), retaining only the multi-step bootstrapping updates. This allows us to recover the equalities in Equation (55) and Equation (56).

**(b) Proving unbiasedness of multi-step bootstrapping.** We rigorously prove Equation (57), showing that $\hat{Q}_h^{k,d}(s,a)$ and $\overline{V}_{h'}^k(s_{h'}^k)$ form an unbiased estimate of the optimal value function $Q^\star$.

**(c) Ensuring Martingale Difference Condition.** We ensure the validity of Azuma–Hoeffding inequality by centering the two estimators $\hat{Q}_h^{k,d}(s,a)$ and $\overline{V}_{h'}^k(s_{h'}^k)$ in multi-step bootstrapping around their conditional expectations, $Q_h^{k,d}(s,a)$ and $Q_h^{k,ud}(s,a)$.

**(d) Tightening confidence bounds.** By jointly analyzing the concentration of the estimators $\hat{Q}_h^{k,d}(s,a)$ and $\overline{V}_{h'}^k(s_{h'}^k)$, we reduce the bonus $b_n'$ by half, leading to better empirical performance.

We detail our Refined AMB algorithm in the following subsection.

## G.2    REFINED AMB ALGORITHM

We present the Refined AMB algorithm in Algorithm 4 and Algorithm 5, which preserves the overall structure of Xu et al. (2021).

To recover valid upper and lower confidence bounds for the $Q$-estimators, we slightly modify the update rules by shifting the truncation from the $Q$-estimates to the corresponding $V$-estimates:

$$\overline{Q}_h^k(s,a) = (1-\eta_n)\overline{Q}_h^{k-1}(s,a) + \eta_n\left(\hat{Q}_h^{k,d}(s,a) + \overline{V}_{h'}^k(s_{h'}^k) + b_n\right),$$

$$\underline{Q}_h^k(s,a) = (1-\eta_n)\underline{Q}_h^{k-1}(s,a) + \eta_n\left(\hat{Q}_h^{k,d}(s,a) + \underline{V}_{h'}^k(s_{h'}^k) - b_n\right),$$

$$\overline{V}_h^{k+1}(s) = \min\left\{H, \max_{a'\in A_h^k(s)} \overline{Q}_h^{k+1}(s,a')\right\},$$

$$\underline{V}_h^{k+1}(s) = \max\left\{0, \max_{a'\in A_h^k(s)} \underline{Q}_h^{k+1}(s,a')\right\}.$$

Here, the refined bonus is defined as $b_n = b_n'/2$, exactly half of the bonus used in the original AMB algorithm. These modifications enable us to establish the following theorem:

**Theorem G.1** (Formal statement of Theorem 4.3.). *With high probability (under the event $\mathcal{H}$ in Lemma G.1), the following conclusions hold simultaneously for all $(s,a,h,k) \in \mathcal{S} \times \mathcal{A} \times [H] \times [K]$:*

$$\overline{V}_h^k(s) \geq V_h^\star(s) \geq \underline{V}_h^k(s) \quad and \quad \overline{Q}_h^k(s,a) \geq Q_h^\star(s,a) \geq \underline{Q}_h^k(s,a). \tag{58}$$

*Moreover, the following decomposition holds:*

$$Q_h^{k,d}(s,a) + Q_h^{k,ud}(s,a) = Q_h^\star(s,a). \tag{59}$$

The proof is provided in Appendix G.3, where the optimism and pessimism properties of the $Q$-estimators are formally established. By adapting the remaining arguments from Xu et al. (2021) along with the simplifications in Appendix G.4, we show that the Refined AMB algorithm achieves the following fine-grained gap-dependent expected regret upper bound:

$$O\left(\sum_{h=1}^H \sum_{\Delta_h(s,a)>0} \frac{H^5 \log(SAT)}{\Delta_h(s,a)} + \frac{H^5 |Z_{\text{mul}}| \log(SAT)}{\Delta_{\min}}\right).$$

Here, for any $h \in [H]$, we have $|Z_{\text{opt},h}(s)| = \{a \in \mathcal{A} | \Delta_h(s,a) = 0\}$ and

$$|Z_{\text{mul}}| = \{(s,a,h) \in \mathcal{S} \times \mathcal{A} \times [H] | \Delta_h(s,a) = 0, |Z_{\text{opt},h}(s)| > 1\}.$$

---

**Algorithm 4** Refined Adaptive Multi-step Bootstrap (Refined AMB)

---

1: **Input:** $p \in (0,1)$ (failure probability), $H, A, S, K \geq 1$
2: **Initialization:** For any $\forall (s,a,h) \in \mathcal{S} \times \mathcal{A} \times [H]$, initialize $\overline{Q}_h^1(s,a) \leftarrow H, \underline{Q}_h^1(s,a) \leftarrow 0,$
   $G_h^1 = \emptyset, A_h^1(s) \leftarrow \mathcal{A}$ and $\overline{V}_h^1(s) = \underline{V}_h^1(s) = 0.$
3: **for** $k = 1, 2, \ldots, K$ **do**
4:     **Step 1: Collect data:**
5:     Rollout from a random initial state $s_1^k \sim \mu$ using policy $\pi_k = \{\pi_h^k\}_{h=1}^H$, defined as:

$$\pi_h^k(s) \triangleq \begin{cases} \arg\max_{a \in A_h^k(s)} \overline{Q}_h^k(s,a) - \underline{Q}_h^k(s,a), & \text{if } |A_h^k(s)| > 1 \\ \text{the element in } A_h^k(s), & \text{if } |A_h^k(s)| = 1 \end{cases}$$

6:     and obtain an episode $\left\{ (s_h^k, a_h^k, r_h^k = r_h(s_h^k, a_h^k)) \right\}_{h=1}^H$..
7:     **Step 2: Update Q-function:**
8:     **for** $h = H, H-1, \ldots, 1$ **do**
9:         **if** $s_h^k \notin G_h^k$ **then**
10:            UPDATE$(s_h^k, a_h^k, k, h)$.
11:        **end if**
12:    **end for**
13:    **for** $(s,a,h) \in \mathcal{S} \times \mathcal{A} \times [H] \setminus \{(s_h^k, a_h^k) | 1 \leq h \leq H, s_h^k \notin G_h^k\}_{h=1}^H$ **do**
14:        $\overline{Q}_h^{k+1}(s,a) = \overline{Q}_h^k(s,a), \underline{Q}_h^{k+1}(s,a) = \underline{Q}_h^k(s,a),$
15:        $\overline{V}_h^{k+1}(s) = \overline{V}_h^k(s), \underline{V}_h^{k+1}(s) = \underline{V}_h^k(s).$
16:    **end for**
17:    **Step 3: Eliminate the sub-optimal actions:**
18:    $\forall s \in \mathcal{S}, h \in [H]$, set $A_h^{k+1}(s) = \left\{ a \in A_h^k(s) : \overline{Q}_h^k(s,a) \geq \underline{V}_h^k(s) \right\}$
19:    $\forall s \in \mathcal{S}$, set $G_h^{k+1} = \left\{ s \in \mathcal{S} : |A_h^{k+1}(s)| = 1 \right\}.$
20: **end for**

---

**Algorithm 5** UPDATE$(s, a, k, h)$

---

1: Set $\overline{V}_{H+1}^k = \underline{V}_{H+1}^k(s) = 0.$
2: $\forall n$, set $\eta_n = \frac{H+1}{H+n}.$
3: Let $n = N_h^{k+1}(s,a)$ be the number of visits to $(s,a)$ at step $h$ in the first $k$ episodes.
4: Let $h' = h'(h,k)$ be the first index after step $h$ in episode $k$ such that $s_{h'}^k \notin G_{h'}^k$. (If such a state does not exist, set $h' = H+1$.)
5: Compute bonus: $b_n = 2\sqrt{H^3 \log(2SAT/p)/n}.$
6: Compute partial return: $\hat{Q}_h^{k,d}(s,a) = \sum_{h \leq i < h'} r_i^k.$
7: $\overline{Q}_h^{k+1}(s,a) = (1 - \eta_n)\overline{Q}_h^k(s,a) + \eta_n \left( \hat{Q}_h^{k,d}(s,a) + \overline{V}_{h'}^k(s_{h'}^k) + b_n \right).$
8: $\underline{Q}_h^{k+1}(s,a) = (1 - \eta_n)\underline{Q}_h^k(s,a) + \eta_n \left( \hat{Q}_h^{k,d}(s,a) + \underline{V}_{h'}^k(s_{h'}^k) - b_n \right).$
9: $\overline{V}_h^{k+1}(s) = \min \left\{ H, \max_{a' \in A_h^k(s)} \overline{Q}_h^{k+1}(s,a') \right\}.$
10: $\underline{V}_h^{k+1}(s) = \max \left\{ 0, \max_{a' \in A_h^k(s)} \underline{Q}_h^{k+1}(s,a') \right\}.$

---

### G.3 PROOF OF THEOREM G.1

We first prove some probability events to facilitate our proof.

**Lemma G.1.** *Let $\iota = \log(2SAT/p)$ for any failure probability $p \in (0,1)$. Then with probability at least $1-p$, the following event $\mathcal{H}$ holds:*

$$\left| \sum_{i=1}^{N_h^k} \eta_i^{N_h^k} \left( \left( \hat{Q}_h^{k^i,d} - Q_h^{k^i,d} \right)(s,a) + V_{h'}^\star(s_{h'}^{k^i}) - Q_h^{k^i,ud}(s,a) \right) \right| \leq 2\sqrt{\frac{H^3\iota}{N_h^k(s,a)}}, \quad \forall(s,a,h,k).$$

*Proof.* The sequence

$$\left\{ \sum_{i=1}^{N} \eta_i^N \left( \left( \hat{Q}_h^{k^i,d} - Q_h^{k^i,d} \right)(s,a) + V_{h'}^\star(s_{h'}^{k^i}) - Q_h^{k^i,ud}(s,a) \right) \right\}_{N \in \mathbb{N}^+}$$

is a martingale sequence with

$$\left| \eta_i^N \left( \left( \hat{Q}_h^{k^i,d} - Q_h^{k^i,d} \right)(s,a) + V_{h'}^\star(s_{h'}^{k^i}) - Q_h^{k^i,ud}(s,a) \right) \right| \le \eta_i^N H.$$

Then according to Azuma-Hoeffding inequality and (b) of Lemma D.2, for any $p \in (0,1)$, with probability at least $1 - \frac{p}{SAT}$, it holds for given $N_h^k(s,a) = N \in \mathbb{N}_+$ that:

$$\left| \sum_{i=1}^{N} \eta_i^N \left( \left( \hat{Q}_h^{k^i,d} - Q_h^{k^i,d} \right)(s,a) + V_{h'}^\star(s_{h'}^{k^i}) - Q_h^{k^i,ud}(s,a) \right) \right| \le 2\sqrt{\frac{H^3 \iota}{N}}.$$

For any $(s,a,h,k) \in \mathcal{S} \times \mathcal{A} \times [H] \times [K]$, we have $N_h^k(s,a) \in [\frac{T}{H}]$. Considering all the possible combinations $(s,a,h,N) \in \mathcal{S} \times \mathcal{A} \times [H] \times [\frac{T}{H}]$, with probability at least $1-p$, it holds simultaneously for all $(s,a,h,k) \in \mathcal{S} \times \mathcal{A} \times [H] \times [K]$ that:

$$\left| \sum_{i=1}^{N_h^k} \eta_i^{N_h^k} \left( \left( \hat{Q}_h^{k^i,d} - Q_h^{k^i,d} \right)(s,a) + V_{h'}^\star(s_{h'}^{k^i}) - Q_h^{k^i,ud}(s,a) \right) \right| \le 2\sqrt{\frac{H^3 \iota}{N_h^k(s,a)}}.$$

$\square$

Now we use mathematical induction on $k$ to prove Theorem G.1 under the event $\mathcal{H}$.

*Proof.* **Part 1: Proof for $k = 1$.**

For $k = 1$, the Equation (58) holds based on the initialization in line 2 of Algorithm 4.

Now we prove Equation (59) for $k = 1$ by induction on $h = H, ..., 1$.

For $h = H$, we have $h'(1, H) = H + 1$. Equation (59) holds in this case since $Q_H^\star(s,a) = r_H(s,a) = Q_H^{1,d}(s,a)$ and $Q_H^{1,ud}(s,a) = 0$. Now assume that Equation (59) holds for $H, ..., h+1$. We will also show it holds for step $h$.

First, we expand $Q_h^{1,d}(s,a)$ as follows:

$$Q_h^{1,d}(s,a) = \mathbb{E}\left[ \sum_{l=h}^{h'-1} r_l(s_l^1, \pi_l^\star(s_l^1)) \mid \mathcal{F}_{h,1}, (s_h^1, a_h^1) = (s,a) \right]$$

$$= \left( \sum_{s' \notin G_{h+1}^1} + \sum_{s' \in G_{h+1}^1} \right) \mathbb{E}\left[ \sum_{l=h}^{h'-1} r_l(s_l^1, \pi_l^\star(s_l^1)) \mid \mathcal{F}_{h,1}, (s_h^1, a_h^1) = (s,a), s_{h+1}^1 = s' \right]$$

$$\times \mathbb{P}\left( s_{h+1}^1 = s' \mid (s_h^1, a_h^1) = (s,a) \right) \tag{60}$$

$$= \sum_{s' \notin G_{h+1}^1} r_h(s,a) \mathbb{P}\left( s_{h+1}^1 = s' \mid (s_h^1, a_h^1) = (s,a) \right)$$

$$+ \sum_{s' \in G_{h+1}^1} \left( r_h(s,a) + Q_{h+1}^{1,d}(s', \pi_{h+1}^\star(s')) \right) \mathbb{P}\left( s_{h+1}^1 = s' \mid (s_h^1, a_h^1) = (s,a) \right) \tag{61}$$

$$= r_h(s,a) + \sum_{s' \in G_{h+1}^1} Q_{h+1}^{1,d}(s', \pi_{h+1}^\star(s')) \mathbb{P}\left( s_{h+1}^1 = s' \mid (s_h^1, a_h^1) = (s,a) \right). \tag{62}$$

The Equation (60) is obtained by applying the law of total expectation with respect to $s_{h+1}^1$, and leveraging the Markov property of the process. Equation (61) is because:

If $s_{h+1}^1 \notin G_{h+1}^1$, then $h' = h'(k, h) = h + 1$ and

$$\mathbb{E}\left[\sum_{l=h}^{h'-1} r(s_l^1, \pi_l^\star(s_l^1)) \mid \mathcal{F}_{h,1}, (s_h^1, a_h^1) = (s, a), s_{h+1}^1 = s'\right] = r_h(s, a);$$

If $s_{h+1}^1 \in G_{h+1}^1$, then $h' = h'(k, h) = h'(k, h+1)$. In this case, since $\overline{Q}_{h+1}^1 \geq Q_{h+1}^\star \geq \underline{Q}_{h+1}^1$, $a_{h+1}^1 = \pi_h^1(s_{h+1}^1)$ is the unique optimal action $\pi_{h+1}^\star(s_{h+1}^1)$. Therefore we have

$$\mathbb{E}\left[\sum_{l=h}^{h'-1} r(s_l^1, \pi_l^\star(s_l^1)) \mid \mathcal{F}_{h,1}, (s_h^1, a_h^1) = (s, a), s_{h+1}^1 = s'\right]$$

$$= r_h(s, a) + \mathbb{E}\left[\sum_{l=h+1}^{h'-1} r(s_l^1, \pi_l^\star(s_l^1)) \mid \mathcal{F}_{h+1,1}, (s_{h+1}^1, a_{h+1}^1) = (s', \pi_{h+1}^\star(s'))\right]$$

$$= r_h(s, a) + Q_{h+1}^{1,d}(s', \pi_{h+1}^\star(s')).$$

Similarly, we also have

$$Q_h^{1,ud}(s, a) = \mathbb{E}\left[V_{h'}^\star(s_{h'}^1) \mid \mathcal{F}_{h,1}, (s_h^1, a_h^1) = (s, a)\right]$$

$$= \sum_{s' \notin G_{h+1}^1} \mathbb{E}\left[V_{h'}^\star(s_{h'}^1) \mid \mathcal{F}_{h,1}, (s_h^1, a_h^1) = (s, a), s_{h+1}^1 = s'\right] \mathbb{P}\left(s_{h+1}^1 = s' | (s_h^1, a_h^1) = (s, a)\right)$$

$$+ \sum_{s' \in G_{h+1}^1} \mathbb{E}\left[V_{h'}^\star(s_{h'}^1) \mid \mathcal{F}_{h,1}, (s_h^1, a_h^1) = (s, a), s_{h+1}^1 = s'\right] \mathbb{P}\left(s_{h+1}^1 = s' | (s_h^1, a_h^1) = (s, a)\right)$$

$$= \sum_{s' \notin G_{h+1}^1} V_{h+1}^\star(s') \mathbb{P}\left(s_{h+1}^1 = s' | (s_h^1, a_h^1) = (s, a)\right)$$

$$+ \sum_{s' \in G_{h+1}^1} Q_{h+1}^{1,ud}(s', \pi_{h+1}^\star(s')) \mathbb{P}\left(s_{h+1}^1 = s' | (s_h^1, a_h^1) = (s, a)\right). \tag{63}$$

Here Equation (63) is because if $s_{h+1}^1 \notin G_{h+1}^1$, then $h' = h'(k, h) = h + 1$ and

$$\mathbb{E}\left[V_{h'}^\star(s_{h'}^1) \mid \mathcal{F}_{h,1}, (s_h^1, a_h^1) = (s, a), s_{h+1}^1 = s'\right] = V_{h+1}^\star(s');$$

If $s_{h+1}^1 \in G_{h+1}^1$, then $h' = h'(k, h) = h'(k, h+1)$ and

$$\mathbb{E}\left[V_{h'}^\star(s_{h'}^1) \mid \mathcal{F}_{h,1}, (s_h^1, a_h^1) = (s, a), s_{h+1}^1 = s'\right] = Q_{h+1}^{1,ud}(s', \pi_{h+1}^\star(s')).$$

Combining the results of Equation (62) and Equation (63), we reach:

$$Q_h^{1,d}(s, a) + Q_h^{1,ud}(s, a)$$

$$= r_h(s, a) + \sum_{s' \notin G_{h+1}^1} V_{h+1}^\star(s') \mathbb{P}\left(s_{h+1}^1 = s' | (s_h^1, a_h^1) = (s, a)\right)$$

$$+ \sum_{s' \in G_{h+1}^1} \left(Q_{h+1}^{1,d}(s', \pi_{h+1}^\star(s')) + Q_{h+1}^{1,ud}(s', \pi_{h+1}^\star(s'))\right) \mathbb{P}\left(s_{h+1}^1 = s' | (s_h^1, a_h^1) = (s, a)\right)$$

$$= r_h(s, a) + \sum_{s' \notin G_{h+1}^1} V_{h+1}^\star(s') \mathbb{P}\left(s_{h+1}^1 = s' | (s_h^1, a_h^1) = (s, a)\right)$$

$$+ \sum_{s' \in G_{h+1}^1} V_{h+1}^\star(s') \mathbb{P}\left(s_{h+1}^1 = s' | (s_h^1, a_h^1) = (s, a)\right) \tag{64}$$

$$= r_h(s, a) + \sum_{s'} V_{h+1}^\star(s') \mathbb{P}\left(s_{h+1}^1 = s' | (s_h^1, a_h^1) = (s, a)\right)$$

$$= Q_h^\star(s, a) \tag{65}$$

Equation (64) is because by induction, we have

$$Q_{h+1}^{1,d}(s', \pi_{h+1}^\star(s')) + Q_{h+1}^{1,ud}(s', \pi_{h+1}^\star(s')) = Q_{h+1}^\star(s', \pi_{h+1}^\star(s')) = V_{h+1}^\star(s').$$

Equation (65) uses Bellman Optimality Equation in Equation (2).

**Part 2.1: Proof of Equation (58) for $k+1$.**

Assuming that the conclusions Equation (58) and Equation (59) hold for all $1, 2, ..., k$, we will prove the conclusions for $k+1$.

If $(s, a, h) \in \mathcal{S} \times \mathcal{A} \times [H] \setminus \{(s_h^k, a_h^k) | 1 \le h \le H, s_h^k \notin G_h^k\}_{h=1}^H$, then we have

$$\overline{V}_h^{k+1}(s) = \overline{V}_h^k(s) \ge V_h^\star(s) \ge \underline{V}_h^k(s) = \underline{V}_h^{k+1}(s).$$

and

$$\overline{Q}_h^{k+1}(s, a) = \overline{Q}_h^k(s, a) \ge Q_h^\star(s, a) \ge \underline{Q}_h^k(s, a) = \underline{Q}_h^{k+1}(s, a).$$

For $(s_h^k, a_h^k, h)$ with $s_h^k \notin G_h^k$, based on the update rule in line 6 and line 7 in Algorithm 5, we have

$$\overline{Q}_h^{k+1}(s_h^k, a_h^k) = \eta_0^{N_h^{k+1}} H + \sum_{i=1}^{N_h^{k+1}} \eta_i^{N_h^{k+1}} \left( \hat{Q}_h^{k^i,d}(s, a) + \overline{V}_{h'(k^i,h)}^{k^i}(s_{h'(k^i,h)}^{k^i}) + b_i \right)$$

$$\ge \eta_0^{N_h^{k+1}} H + \sum_{i=1}^{N_h^{k+1}} \eta_i^{N_h^{k+1}} \left( \hat{Q}_h^{k^i,d}(s, a) + \overline{V}_{h'(k^i,h)}^{k^i}(s_{h'(k^i,h)}^{k^i}) \right) + 2\sqrt{\frac{H^3\iota}{N_h^{k+1}}}, \quad (66)$$

and

$$\underline{Q}_h^{k+1}(s_h^k, a_h^k) = \sum_{i=1}^{N_h^{k+1}} \eta_i^{N_h^{k+1}} \left( \hat{Q}_h^{k^i,d}(s, a) + \underline{V}_{h'(k^i,h)}^{k^i}(s_{h'(k^i,h)}^{k^i}) - b_i \right).$$

$$\le \sum_{i=1}^{N_h^{k+1}} \eta_i^{N_h^{k+1}} \left( \hat{Q}_h^{k^i,d}(s, a) + \underline{V}_{h'(k^i,h)}^{k^i}(s_{h'(k^i,h)}^{k^i}) \right) - 2\sqrt{\frac{H^3\iota}{N_h^{k+1}}}. \quad (67)$$

These two inequalities are because

$$\sum_{i=1}^{N_h^{k+1}} \eta_i^{N_h^{k+1}} b_i = 2 \sum_{i=1}^{N_h^{k+1}} \eta_i^{N_h^{k+1}} \sqrt{\frac{H^3\iota}{i}} \ge 2\sqrt{\frac{H^3\iota}{N_h^{k+1}}}$$

by (c) of Lemma D.2. Furthermore, by Equation (59) for $k^i \le k$, it holds that:

$$Q_h^\star(s_h^k, a_h^k) = Q_h^{k^i,d}(s, a) + Q_h^{k^i,ud}(s, a).$$

Combining with Equation (66) and Equation (67), we can derive the following conclusion:

$$\left( \overline{Q}_h^{k+1} - Q_h^\star \right)(s_h^k, a_h^k)$$

$$\ge \sum_{i=1}^{N_h^{k+1}} \eta_i^{N_h^{k+1}} \left( \hat{Q}_h^{k^i,d}(s, a) + \overline{V}_{h'(k^i,h)}^{k^i}(s_{h'(k^i,h)}^{k^i}) - Q_h^\star(s_h^k, a_h^k) \right) + 2\sqrt{\frac{H^3\iota}{N_h^{k+1}}}$$

$$= \sum_{i=1}^{N_h^{k+1}} \eta_i^{N_h^{k+1}} \left( \overline{V}_{h'}^{k^i} - V_{h'}^\star \right)(s_{h'}^{k^i})$$

$$+ \sum_{i=1}^{N_h^k} \eta_i^{N_h^k} \left( \hat{Q}_h^{k^i,d}(s, a) - Q_h^{k^i,d}(s, a) + V_{h'}^\star(s_{h'}^{k^i}) - Q_h^{k^i,ud}(s, a) \right) + 2\sqrt{\frac{H^3\iota}{N_h^{k+1}}} \ge 0.$$

The last inequality holds because $\overline{V}_{h+1}^{k^i}(s_{h+1}^{k^i}) \geq V_{h+1}^{\star}(s_{h+1}^{k^i})$ for all $k^i \leq k$ and the event $\mathcal{H}$ in Lemma G.1. Similarly, we can prove the pessimism of $\underline{Q}_h^{k+1}$:

$$\left(\underline{Q}_h^{k+1} - Q_h^{\star}\right)(s_h^k, a_h^k)$$

$$\leq \sum_{i=1}^{N_h^{k+1}} \eta_i^{N_h^{k+1}} \left(\hat{Q}_h^{k^i,d}(s,a) + \underline{V}_{h'(k^i,h)}^{k^i}(s_{h'(k^i,h)}^{k^i}) - Q_h^{\star}(s_h^k,a_h^k)\right) + 2\sqrt{\frac{H^3\iota}{N_h^{k+1}}}$$

$$= \sum_{i=1}^{N_h^{k+1}} \eta_i^{N_h^{k+1}} \left(\underline{V}_{h'}^{k^i} - V_{h'}^{\star}\right)(s_{h'}^{k^i})$$

$$+ \sum_{i=1}^{N_h^k} \eta_i^{N_h^k} \left(\hat{Q}_h^{k^i,d}(s,a) - Q_h^{k^i,d}(s,a) + V_{h'}^{\star}(s_{h'}^{k^i}) - Q_h^{k^i,ud}(s,a)\right) - 2\sqrt{\frac{H^3\iota}{N_h^{k+1}}} \leq 0.$$

The last inequality holds because $\underline{V}_{h+1}^{k^i}(s_{h+1}^{k^i}) \leq V_{h+1}^{\star}(s_{h+1}^{k^i})$ for all $k^i \leq k$ and the event $\mathcal{H}$. With this, we have shown that $\overline{Q}_h^{k+1}(s,a) \geq Q_h^{\star}(s,a) \geq \underline{Q}_h^{k+1}(s,a)$. Therefore, by noting that

$$\overline{V}_h^{k+1}(s) = \min\left\{H, \max_{a\in A_h^k(s)} \overline{Q}_h^{k+1}(s,a)\right\} \geq \max_{a\in A_h^k(s)} Q_h^{\star}(s,a) = V_h^{\star}(s)$$

and

$$\underline{V}_h^{k+1}(s) = \max\left\{0, \max_{a\in A_h^k(s)} \underline{Q}_h^{k+1}(s,a)\right\} \leq \max_a Q_h^{\star}(s,a) = V_h^{\star}(s),$$

we complete the proof of the Equation (58) for $k+1$.

**Part 2.2: Proof of Equation (59) for $k+1$.**

Next we prove Equation (59) for $k+1$ by induction on $h = H, ..., 1$.

For $h = H$, we have $h'(k,H) = H+1$. Equation (59) holds in this case since $Q_H^{\star}(s,a) = r_H(s,a) = Q_H^{1,d}(s,a)$ and $Q_H^{1,ud}(s,a) = 0$. Assume that the conclusion holds for $H, ..., h+1$. For step $h$, similar to Equation (62) and Equation (63) for $k=1$, we obtain:

$$Q_h^{k+1,d}(s,a) = r_h(s,a) + \sum_{s'\in G_{h+1}^{k+1}} Q_{h+1}^{k+1,d}(s', \pi_{h+1}^{\star}(s'))\mathbb{P}\left(s_{h+1}^{k+1} = s'|(s_h^{k+1}, a_h^{k+1}) = (s,a)\right)$$

and

$$Q_h^{k+1,ud}(s,a) = \sum_{s'\notin G_{h+1}^{k+1}} V_{h+1}^{\star}(s')\mathbb{P}\left(s_{h+1}^{k+1} = s'|(s_h^{k+1}, a_h^{k+1}) = (s,a)\right)$$

$$+ \sum_{s'\in G_{h+1}^{k+1}} Q_{h+1}^{k+1,ud}(s', \pi_{h+1}^{\star}(s'))\mathbb{P}\left(s_{h+1}^{k+1} = s'|(s_h^{k+1}, a_h^{k+1}) = (s,a)\right).$$

By combining these two equations, as in Equation (65), we establish Equation (59) at step $h$ for $k+1$, which completes the inductive process and thus proves Theorem G.1. $\qquad\square$

This theorem successfully establishes the optimism and pessimism properties of the $Q$-estimators. Leveraging the remaining arguments in Xu et al. (2021), we can recover the same gap-dependent expected regret upper bound presented in Equation (11).

### G.4 RESULT SIMPLIFICATION

By adapting the remaining arguments from Xu et al. (2021), we can recover the following bound for Refined AMB algorithm:

$$O\left(\sum_{h=1}^{H}\sum_{\Delta_h(s,a)>0} \frac{H^5 \log(SAT)}{\Delta_h(s,a)} + \frac{H^5|Z_{\mathrm{mul}}|\log(SAT)}{\Delta_{\min}} + SAH^2\right) \qquad (68)$$

Define $Z_{\text{sub}} = \{(s, a, h) : \Delta_h(s, a) > 0\}$ and recall that $Z_{\text{opt}} = \{(s, a, h) : \Delta_h(s, a) = 0\}$ and $Z_{\text{mul}} = \{(s, a, h) : \Delta_h(s, a) = 0, |Z_{\text{opt},h}(s)| > 1\}$, where $Z_{\text{opt},h}(s) = \{a : \Delta_h(s, a) = 0\}$. Then we have

$$|Z_{\text{sub}}| + |Z_{\text{mul}}| = |Z_{\text{sub}}| + |Z_{\text{opt}}| - (|Z_{\text{opt}}| - |Z_{\text{mul}}|) \geq S(A - 1)H \geq \frac{SAH}{2},$$

because $|Z_{\text{sub}}| + |Z_{\text{opt}}| = HSA$ and

$$|Z_{\text{opt}}| - |Z_{\text{mul}}| = |Z_{\text{opt}} \setminus Z_{\text{mul}}| = |\{(s, a, h) : \Delta_h(s, a) = 0, |Z_{\text{opt},h}(s)| = 1\}| \leq HS$$

since for each $(s, a, h) \in Z_{\text{opt}} \setminus Z_{\text{mul}}$, we have $|Z_{\text{opt},h}(s)| = 1$, which implies that the optimal action $a$ is unique for each state–step pair $(s, h)$. Therefore,

$$\sum_{h=1}^{H} \sum_{\Delta_h(s,a)>0} \frac{H^5 \log(SAT)}{\Delta_h(s,a)} + \frac{H^5 |Z_{\text{mul}}| \log(SAT)}{\Delta_{\min}}$$

$$\geq \sum_{h=1}^{H} \sum_{\Delta_h(s,a)>0} H^4 \log(SAT) + H^4 |Z_{\text{mul}}| \log(SAT)$$

$$= H^4 (|Z_{\text{sub}}| + |Z_{\text{mul}}|) \log(SAT)$$

$$\geq \frac{SAH^5}{4},$$

where we used $0 < \Delta_h(s, a), \Delta_{\min} \leq H$ in the first inequality and $\log(SAT) \geq 1/2$ for $S, T \geq 1$ and $A \geq 2$ in the last inequality.

Thus, the Refined AMB result in Equation (68) can be equivalently written as

$$O\left(\sum_{h=1}^{H} \sum_{\Delta_h(s,a)>0} \frac{H^5 \log(SAT)}{\Delta_h(s,a)} + \frac{H^5 |Z_{\text{mul}}| \log(SAT)}{\Delta_{\min}}\right).$$

