# OpenReview forum: "Q-Learning with Fine-Grained Gap-Dependent Regret"
_ICLR.cc/2026/Conference — ICLR 2026 Poster_

### Official Review · Reviewer_omHP · 2025-10-30

**Soundness:** 3
**Presentation:** 3
**Contribution:** 3
**Rating:** 8
**Confidence:** 4

**Summary:**

This paper presents the  fine-grained, gap-dependent regret bounds for model-free reinforcement learning in episodic tabular Markov Decision Processes. While previous algorithms achieved minimax worst-case guarantees, their gap-dependent analyses were coarse, depending on the smallest suboptimality gap rather than individual ones. The authors introduce a new analytical framework that explicitly separates optimal and suboptimal state-action pairs, enabling fine-grained regret analysis. They apply this framework to the well-known UCB-Hoeffding algorithm, deriving a tighter bound that matches known lower bounds up to polynomial factors, and propose a simplified variant, ULCB-Hoeffding, which achieves similar theoretical guarantees with improved empirical performance. The paper also revisits the non-UCB-based AMB algorithm, identifying key theoretical flaws and proposing a refined version that restores correctness, ensures valid concentration analysis, and achieves a rigorous fine-grained regret bound in this class. Experimental results on synthetic MDPs confirm the theoretical findings, showing that the refined algorithms outperform prior methods. Overall, the work provides a unified framework that advances theoretical understanding of model-free reinforcement learning by bridging worst-case and instance-dependent analyses.

**Strengths:**

1. Introduces a fine-grained decomposition that separately analyzes optimal and suboptimal pairs, tightening gap-dependent bounds.
2. Framework applies to both UCB-based and non-UCB-based algorithms.
3. Provides the first fine-grained regret bound for model-free RL; matches known lower bounds up to polynomial factors.
4.  Identifies and corrects subtle theoretical flaws in prior work (AMB).
5. Experiments confirm theoretical improvements and show scalability across MDP sizes.

**Weaknesses:**

1. Results do not extend to function approximation or continuous-state RL.

2.  The theoretical derivations are mathematically heavy and might be difficult for practitioners to follow or generalize.

3.  Experiments are conducted on randomly generated MDPs rather than benchmark environments.

4.  The bounds, while asymptotically tight, may not yield practical improvements in all regimes.

5. The experimental comparison includes few competing algorithms beyond AMB and its variants.

**Questions:**

1.  Can the proposed fine-grained framework extend to function approximation (e.g., linear or neural models)?
2.  How do fine-grained regret improvements translate to real-world tasks (e.g., navigation, games)?
3.   Are the polynomial factors in H (e.g., H^5 or H^6) necessary, or could further refinement reduce them?
4.  Does the fine-grained analysis provide insight into exploration dynamics, beyond theoretical improvement?
5.  Could this framework be combined with variance-dependent or adaptive analyses to yield sharper or adaptive regret bounds?
6. Could ULCB-Hoeffding’s structure be adapted to other algorithms (e.g., Q-learning with linear approximation)?

---

> ### Author Response · Authors · 2025-11-20
> **Response to Reviewer omHP (part one)**
>
> We thank you for your careful reading and thoughtful comments. Below are our point-by-point responses, and we hope our responses address your concerns.
>
> **Weakness 1 and Question 1: Extension to function approximation**
>
> To address this comment, we first review the existing gap-dependent results with function approximation in the RL literature. To our knowledge, for linear MDPs, only three works [1,2,3] have established gap-dependent guarantees. However, all of these results depend solely on the global minimum gap $\Delta_{\min}$, and thus do not provide fine-grained performance bounds. For general function approximation, even such coarse-grained guarantees are still unavailable.
>
> Extending gap-dependent analyses from tabular RL to RL with function approximation is an important yet challenging direction. The main difficulties arise from the continuous state--action space. Although a suboptimality gap can be defined for each state--action pair, how to define a fine-grained result in this setting remains unclear, since the summation $\sum_{s,a:\Delta_h(s,a)>0} \frac{1}{\Delta_h(s,a)}$, which is central to our tabular fine-grained results, can no longer be applied to a continuous state-action space. Whether an integral counterpart serves as an appropriate analogue is an open question.
>
> From a technical standpoint, extending our framework requires relating the regret to the integral of the weighted estimation error $(Q_h^k - Q_h^\star)(s,a)$ over the continuous state--action space. How to establish such a relationship is unknown in the literature, and developing it requires new techniques that go beyond current gap-dependent analyses.
>
> For these reasons, extending fine-grained results from tabular MDPs to linear MDPs or general function approximation is an important but technically challenging open direction for the entire literature, rather than a limitation specific to our approach.
>
>
> **Weakness 2: Proof sketch**
>
> To make it easier for practitioners to follow, we have highlighted the key ideas of our framework in lines 291-302 of the revised manuscript, as presented below:
>
> (1) We first establish Lemma 3.1, which upper-bounds the regret by the expectation of the cumulative weighted visitation counts
> $\sum_{h=1}^H \sum_{s,a} \Delta_h(s,a)N_h^{K+1}(s,a)$
> and further relates this term to the cumulative weighted estimation errors
> $\sum_{k=1}^K \omega_h^k (Q_h^k - Q_h^\star)(s_h^k, a_h^k)$.
>
> (2) We then bound the cumulative weighted estimation errors by establishing a recursive relationship between consecutive steps (Lemma 3.2) and propagating it to the final step $H$ (Lemma 3.3).
>
> (3) Using Lemmas 3.2 and 3.3, we derive a recursive relation for the cumulative weighted visitation counts $\sum_{h=1}^H \sum_{s,a} \Delta_h(s,a)N_h^{K+1}(s,a)$ across steps, which enables an inductive argument to derive a fine-grained upper bound and subsequently bounds the expected regret via Lemma 3.1.
>
> **Weakness 3: Experimental Environment**
>
> For all prior model-free gap-dependent works, [4,5] did not include experiments, and [6] conducted experiments on randomly generated MDPs. Therefore, we follow the same experimental setup as [6] in our work. Currently, there is no other established benchmark environment for comparison.
>
>
>
> **Weakness 4, Questions 2 and Question 4: New Exploration Insights and Practical Implication**
>
> The fine-grained analysis does provide additional insight into the exploration dynamics beyond merely improving the theoretical bound.
>
> Prior gap-dependent results rely solely on the minimum gap, which cannot distinguish between MDPs that share the same minimum gap but have very different overall structures. In contrast, our fine-grained bound depends on the full set of suboptimality gaps of all suboptimal state–action pairs. This reveals a new phenomenon in exploration: even when the minimum gap remains the same, an MDP with larger suboptimality gaps on other states and actions can lead the same algorithm to achieve smaller regret and therefore faster learning. This behavior aligns with intuitive exploration dynamics: large suboptimality gaps make it easier for the algorithm to eliminate suboptimal actions early, yet it cannot be captured by previous analyses that used only the minimum gap.
>
> Beyond theoretical interest, this observation also offers guidance for practical applications. For example, in reward design for decision-making systems, such as game environments, practitioners can intentionally assign smaller rewards to clearly suboptimal actions (or equivalently, increase the reward contrast between optimal and suboptimal actions). Doing so effectively enlarges their suboptimality gaps, allowing learning algorithms to discard these suboptimal actions more rapidly and converge to high-performing behaviors more efficiently.

---

> ### Author Response · Authors · 2025-11-20
> **Response to Reviewer omHP (part two)**
>
> **Weakness 5: Experimental performances of other algorithms**
>
> Thank you for the comment. Our experimental goal is to compare model-free algorithms with fine-grained guarantees and empirically verify the theoretical claims in our paper. In particular, our focus is to demonstrate that UCB-Hoeffding, ULCB-Hoeffding, and Refined-AMB indeed achieve logarithmic regret in practice, and that all three algorithms exhibit consistently better performance compared to the original AMB. For this purpose, including additional algorithms would not offer further insight into the phenomena we aim to validate. The comprehensive comparisons among a wider range of model-free algorithms have been provided in [7].
>
> **Question 3 and Question 5: $H$-dependence and potential improvements via variance-aware algorithms**
>
> The high dependence on $H$ arises because UCB-Hoeffding uses a relatively loose Hoeffding-type bonus, which is highly $ H$-dependent. It is also related to the recursive argument over steps to bound the cumulative weighted estimation errors $Q_h^k-Q_h^\star$ in the standard gap-dependent framework for model-free methods. The same $H$-dependency is also observed in the coarse-grained gap-dependent analysis [4] for UCB-Hoeffding.
>
> Improving the dependency on $H$ is possible in principle by incorporating variance-aware techniques and reference-advantage decompositions into our framework. Algorithms such as UCB-Bernstein [8], UCB-Advantage [9], Q-EarlySettled-Advantage [10], and Q-EarlySettled-LowCost [7] all employ variance-based bonuses, which are tighter than those of UCB-Hoeffding, and the latter three additionally use reference-advantage decomposition. As a result, they achieve better worst-case regret [7,8,9,10] and coarse-grained gap-dependent regret [6,7] than UCB-Hoeffding [4,8], and may potentially lead to improved fine-grained results with better $H$-dependency.
>
> However, incorporating variance-based bonuses and reference-advantage decomposition into our framework introduces nontrivial technical challenges. To achieve improved $H$-dependency, we need to handle different bonuses for different state--action--step triples. In contrast, for UCB-Hoeffding, different state--action--step triples with the same number of visits share the same bonus. This difference requires additional analysis to build the recursive structure we established for $\sum_{s,a} \Delta_h(s,a) N_h^{K+1}(s,a)$ at different steps in Lemma D.4. Moreover, [6,7] derive a coarse-grained gap-dependent regret for algorithms with reference-advantage decomposition by introducing a surrogate reference function. Incorporating this surrogate reference function into our framework also requires additional nontrivial technical effort.
>
> **Question 6: Possible extension of ULCB-Hoeffding**
>
> The structural ideas behind ULCB-Hoeffding could potentially be adapted to other algorithms, including Q-learning with linear function approximation. Both optimism (UCB) and pessimism (LCB) have already appeared separately in the linear RL literature. UCB-based exploration is standard in online linear MDP algorithms [11,12], and LCB or pessimistic value estimates are widely used in offline or safe linear RL [13,14]. However, to the best of our knowledge, no existing linear function approximation method combines UCB and LCB simultaneously for action elimination, which is the key design principle of ULCB-Hoeffding. Whether such a combination would help improve theoretical guarantees in the linear MDP setting remains an open question.

---

> ### Author Response · Authors · 2025-11-20
> **Response to Reviewer omHP (part three)**
>
> **Reference:**
>
> [1] He, Jiafan, Dongruo Zhou, and Quanquan Gu. "Logarithmic regret for reinforcement learning with linear function approximation." International Conference on Machine Learning. PMLR, 2021.
>
> [2] Papini, Matteo, et al. "Reinforcement learning in linear mdps: Constant regret and representation selection." Advances in Neural Information Processing Systems 34 (2021): 16371-16383.
>
> [3] Zhang, Weitong, et al. "Achieving constant regret in linear Markov decision processes." Advances in Neural Information Processing Systems 37 (2024): 130694-130738.
>
> [4] Yang, Kunhe, Lin Yang, and Simon Du. "Q-learning with logarithmic regret." International Conference on Artificial Intelligence and Statistics. PMLR, 2021.
>
> [5] Xu, Haike, Tengyu Ma, and Simon Du. "Fine-grained gap-dependent bounds for tabular mdps via adaptive multi-step bootstrap." Conference on Learning Theory. PMLR, 2021.
>
> [6] Zheng, Zhong, Haochen Zhang, and Lingzhou Xue. "Gap-Dependent Bounds for Q-Learning using Reference-Advantage Decomposition." The Thirteenth International Conference on Learning Representations.
>
> [7] Zhang, Haochen, Zhong Zheng, and Lingzhou Xue. "Regret-Optimal Q-Learning with Low Cost for Single-Agent and Federated Reinforcement Learning." arXiv preprint arXiv:2506.04626 (2025).
>
> [8] Jin, Chi, et al. "Is Q-learning provably efficient?." Advances in neural information processing systems 31 (2018).
>
> [9] Zhang, Zihan, Yuan Zhou, and Xiangyang Ji. "Almost optimal model-free reinforcement learning via reference-advantage decomposition." Advances in Neural Information Processing Systems 33 (2020): 15198-15207.
>
> [10] Li, Gen, et al. "Breaking the sample complexity barrier to regret-optimal model-free reinforcement learning." Advances in Neural Information Processing Systems 34 (2021): 17762-17776.
>
> [11] Jin, Chi, et al. "Provably efficient reinforcement learning with linear function approximation." Conference on learning theory. PMLR, 2020.
>
> [12] He, Jiafan, et al. "Nearly minimax optimal reinforcement learning for linear markov decision processes." International Conference on Machine Learning. PMLR, 2023.
>
> [13] Yin, Ming, et al. "Near-optimal offline reinforcement learning with linear representation: Leveraging variance information with pessimism." arXiv preprint arXiv:2203.05804 (2022).
>
> [14] Shi, Laixi, et al. "Pessimistic q-learning for offline reinforcement learning: Towards optimal sample complexity." International conference on machine learning. PMLR, 2022.

---

> ### Author Response · Authors · 2025-11-27
> **Following up on the rebuttal**
>
> Dear Reviewer omHP,
>
> We hope that our responses and revised manuscript have adequately addressed your concerns regarding (1) the extension to function approximation, (2) the heavy mathematical derivations, (3) the experiment design, (4) the practical significance, (5) the $H$-dependency and potential improvement, and (6) the extension of ULCB-Hoeffding. If you have additional questions or concerns, please let us know, and we would be happy to provide further clarification. We truly appreciate your feedback and look forward to hearing from you soon.
>
> Best regards,
>
> Authors of Submission 11791

---

### Official Review · Reviewer_Er67 · 2025-10-31

**Soundness:** 3
**Presentation:** 3
**Contribution:** 2
**Rating:** 6
**Confidence:** 4

**Summary:**

This paper aims to establish fine-grained, gap-dependent regret bounds for model-free algorithms in episodic tabular MDPs. While such bounds exist for model-based methods, model-free approaches have been limited to coarse bounds dependent on the global minimum gap, $\Delta_{min}$. The authors provide a two-part affirmative answer:

1.  For UCB-based algorithms, they develop an analytical framework that distinguishes between optimal and suboptimal state-action pairs. Using this, they derive the first fine-grained, gap-dependent regret bound for the classic UCB-Hoeffding algorithm.
2.  For non-UCB-based algorithms, they revisit the AMB algorithm (Xu et al., 2021), identifying and correcting significant algorithmic and analytical flaws to propose a Refined AMB. They then provide the first rigorous fine-grained bound for a non-UCB-based method.

Empirical results on synthetic MDPs validate the theory, showing that the refined algorithms outperform the original AMB.

**Strengths:**

1.  A key strength is the identification and correction of critical flaws in the AMB algorithm (Xu et al., 2021). The identification of improper truncation and violation of martingale difference conditions is a valuable and clear-cut contribution.
2.  The paper successfully applies a fine-grained analytical framework, separating optimal/suboptimal pairs,  to the model-free setting, yielding the first fine-grained, gap-dependent regret bound for the widely-known UCB-Hoeffding algorithm.
3.  The experiments clearly support the theory. The flawed AMB algorithm performs poorly, while the Refined AMB and UCB-Hoeffding all perform well and exhibit the logarithmic regret behavior predicted by the new theory.

**Weaknesses:**

1.  The paper's primary weakness is its failure to position its analytical framework relative to recent model-based works that also achieve fine-grained bounds, e.g., Dann et al., (2021); Chen et al., (2025). The paper cites these works but does not discuss the technical challenges of adapting their analysis to the model-free setting. Without this comparison, the core technical contribution in Section 3.3 appears to be an incremental adaptation rather than a novel framework.
2.  The derived bounds include $H^5$ and $H^6$ terms. While the focus is on the gap-dependence, this looseness in $H$ is a significant limitation of the current analysis and makes the bounds less tight, even if they match the lower bound except for the factors in $H$.
3.  The inclusion of the ULCB-Hoeffding algorithm feels unnecessary. It achieves the same theoretical bound as UCB-Hoeffding in Theorem 3.3, but performs noticeably worse in the experiments shown in Figure 1, distracting from the two stronger, clearer contributions.

**Questions:**

1. Please explicitly compare your analytical framework in Section 3.3 to the techniques used in model-based papers like Chen et al. (2025). What are the key technical novelties required to make this style of fine-grained analysis work for model-free Q-learning? What new challenges arise from this difference that your analysis overcomes?
2.  Can you elaborate on the source of the large polynomial dependence on $H$? Is this an artifact of the analysis, e.g., from recursively applying bounds, and do you see a path to tightening it?
3.  Could you provide a direct comparison between your final bound for Refined AMB (Eq 10, depending on $|Z_{mul}|$) and the bound for UCB-Hoeffding (depending on $|Z_{opt}|$)? Which bound is tighter, and under what conditions?

---

> ### Author Response · Authors · 2025-11-20
> **Response to Reviewer Er67 (part one)**
>
> We thank you for your careful reading and thoughtful comments. Below are our point-by-point responses, and we hope our responses address your concerns.
>
> **Weakness 1 and Question 1: Technical differences and challenges compared to model-based works:**
>
> All model-based works [1,2,3] first use the following relationship to bound the regret:
> $$\mathbb{E}(\textnormal{Regret}(T)) \leq \mathbb E\left[\sum_{k=1}^K\sum_{h=1}^H E_h^k(s,a)\right],$$
> where
> $$E_h^k(s,a)=Q_h^k(s,a)-(r_h(s,a)+\mathbb E_{s'\sim P_{s,a,h}}[V_{h+1}^k(s')])$$
> is the surplus at $(s,a,h,k)\in \mathcal{S} \times\mathcal{A}\times[H]\times[K]$. They then bound this surplus using the suboptimality gap $\Delta_h(s,a)$.
>
> For all these model-based works, the Q-estimates are updated as
> $$Q_h^k(s,a) = r_h(s,a) + \mathbb{E}\_{s'\sim \hat{P}\_{s,a,h}^k}[V_{h+1}^k(s')] + b_h^k(s,a),$$
> where $\hat{P}\_{s,a,h}^k$ is the empirical estimate of the transition kernel $P_{s,a,h}$ at the beginning of episode $k$. Plugging in this update rule gives
> $$E_h^k(s,a)=b_h^k(s,a)+\mathbb{E}\_{s'\sim \hat{P}\_{s,a,h}^k}[V_{h+1}^k(s')]-\mathbb E_{s'\sim P_{s,a,h}}[V_{h+1}^k(s')]),$$
> where last two terms $\mathbb{E}\_{s'\sim \hat{P}\_{s,a,h}^k}[V_{h+1}^k(s')]$, and $\mathbb E_{s'\sim P_{s,a,h}}[V_{h+1}^k(s')])$  have the same input $V_{h+1}^k$ and $\hat{P}\_{s,a,h}^k$ serves as an unbiased estimate of $P_{s,a,h}$. Therefore, the difference can be bounded by concentration inequalities and the empirical process.
>
> The situation is different for model-free methods, since they do not maintain an empirical estimate $\hat{P}\_{s,a,h}^k$ of the transition kernel. For UCB-Hoeffding, the $Q$-estimates are updated as
> $$Q_h^k(s,a) = r_h(s,a) + \sum_{i = 1}^{N_h^k(s_h^{k}, a_h^{k})}\eta_i^{N_h^k(s_h^{k}, a_h^{k})}V_{h+1}^{k^i(s_h^{k}, a_h^{k},h)} (s_{h+1}^{k^i}) + \beta_{N_h^k(s_h^{k}, a_h^{k})},$$
> and accordingly,
> $$E_h^k(s,a)=b_h^k(s,a)+\sum_{i = 1}^{N_h^k(s_h^{k}, a_h^{k})}\eta_i^{N_h^k(s_h^{k}, a_h^{k})}\left(V_{h+1}^{k^i(s_h^{k}, a_h^{k},h)} (s_{h+1}^{k^i})-\mathbb E_{s'\sim P_{s,a,h}}[V_{h+1}^k(s')]\right).$$
> The inputs of the two terms of the difference above are different: one uses the historical value estimate $V_{h+1}^{k^i}$ and the other uses the current estimate $V_{h+1}^k$. This mismatch introduces additional bias, which prevents the framework used in model-based works from being directly adapted to the model-free setting. Thus, the subsequent analysis in those works does not carry over, and a completely different framework is required for the model-free setting. Next, we explain each component of our framework.
>
> 1. We first establish Lemma 3.1, which bounds the regret by the cumulative weighted visitation counts $\sum_{h=1}^H \sum_{s,a} \Delta_h(s,a) N_h^{K+1}(s,a)$, and we further relate this quantity to the cumulative weighted estimation errors $\sum_{k=1}^K \omega_h^k (Q_h^k - Q_h^\star)(s_h^k, a_h^k)$.
>
> 2. We then bound the cumulative weighted estimation errors by establishing a recursive relationship between consecutive steps (Lemma 3.2) and propagating it to the final step $H$ (Lemma 3.3).
>
> 3. Using Lemmas 3.2 and 3.3, we construct a recursive relationship for the cumulative weighted visitation counts $\sum_{h=1}^H \sum_{s,a} \Delta_h(s,a) N_h^{K+1}(s,a)$ across steps, which enables an inductive argument to derive a fine-grained upper bound and subsequently bound the expected regret via Lemma 3.1.
>
> These innovations form the core of our fine-grained analytical framework and can be applied to a broad class of model-free RL algorithms.

---

> ### Author Response · Authors · 2025-11-20
> **Response to Reviewer Er67 (part two)**
>
> **Weakness 2 and Question 2: High dependency on $H$ and potential improvement:**
>
> The high dependence on $H$ arises because UCB-Hoeffding uses a relatively loose Hoeffding-type bonus, which is highly $ H$-dependent. It is also related to the recursive argument over steps to bound the cumulative weighted estimation errors $Q_h^k-Q_h^\star$ in the standard gap-dependent framework for model-free methods. The same $H$-dependency is also observed in the coarse-grained gap-dependent analysis [4] for UCB-Hoeffding.
>
> Improving the dependency on $H$ is possible in principle by incorporating variance-aware techniques and reference-advantage decompositions into our framework. Algorithms such as UCB-Bernstein [5], UCB-Advantage [6], Q-EarlySettled-Advantage [7], and Q-EarlySettled-LowCost [8] all employ variance-based bonuses, which are tighter than those of UCB-Hoeffding, and the latter three additionally use reference-advantage decomposition. As a result, they achieve better worst-case regret [5,6,7,8] and coarse-grained gap-dependent regret [8,9] than UCB-Hoeffding [4,5], and may potentially lead to improved fine-grained results with better $H$-dependency.
>
> However, incorporating variance-based bonuses and reference-advantage decomposition into our framework introduces nontrivial technical challenges. To achieve improved $H$-dependency, we need to handle different bonuses for different state--action--step triples. In contrast, for UCB-Hoeffding, different state--action--step triples with the same number of visits share the same bonus. This difference requires additional analysis to build the recursive structure we established for $\sum_{s,a} \Delta_h(s,a) N_h^{K+1}(s,a)$ at different steps in Lemma D.4. Moreover, [8,9] derive a coarse-grained gap-dependent regret for algorithms with reference-advantage decomposition by introducing a surrogate reference function. Incorporating this surrogate reference function into our framework also requires additional nontrivial technical effort.
>
> **Weakness 3: Significance of ULCB-Hoeffding**
>
> To fully highlight the significance of incorporating ULCB-Hoeffding, we have rewritten the introduction in lines 55-60 and 100-110 of the revised manuscript, as summarized in the discussion below.
>
> The key novelty in the design of the AMB algorithm lies in two components: ULCB and multi-step bootstrapping. In particular, ULCB refers to leveraging both UCB and Lower Confidence Bound (LCB) techniques to select actions by maximizing the width of the confidence interval rather than maximizing the $Q$-estimates.
>
> UCB-Hoeffding cannot determine whether the optimal action has been learned, as it only maintains an upper-bound estimate of the optimal $Q$-function. Compared to UCB-Hoeffding, the action elimination technique (see line 14 of Algorithm 2 in our paper or line 23 in the original AMB algorithm), aided by the lower-bound Q-estimates in ULCB, enables the agent to determine, at each step and for each state, whether the optimal action has been learned. This not only improves the explainability of the algorithm but may also be of independent interest (see Section 1.2.2 in [10] for further discussion).
>
> In this paper, we incorporate ULCB-Hoeffding to demonstrate that, even without multi-step bootstrapping, which was incorrectly handled in the proof for the original AMB, our framework can be used to show that an algorithm employing the ULCB technique alone can achieve a fine-grained regret, thereby illustrating the generality of our framework. ULCB-Hoeffding can also be regarded as a UCB-based refinement of the original AMB algorithm.

---

> ### Author Response · Authors · 2025-11-20
> **Response to Reviewer Er67 (part three)**
>
> **Question 3: Result Comparison for UCB-Hoeffding and Refined AMB**
>
> After a careful analysis in Section F.4 of the revised manuscript, we derive a equivalent simplified version of Equation (10) that preserves the same scale while removing the last term, $SAH^2$. We use this new Equation (10) in the following comparison.
>
> **Comparison between Equation (2) for UCB-Hoeffding and Equation (10) for Refined AMB.**
>
> Each result exhibits advantages in different scenarios:
>
> - In the case where the MDP has only one suboptimal state-action-step triple $(s,a,h)$ with $h = H$, Equation (2) reduces to $\tilde{O}(H^5 / \Delta_{\min} + SAH^3)$, whereas Equation (10) gives a worse result of $\tilde{O}(H^6 SA / \Delta_{\min})$.
>
> - For an MDP with a unique optimal action for each state-step pair, Equation (2) can scale as poorly as
> $$
> O\left(\sum_{h=1}^H \sum_{\Delta_h(s,a)>0} \frac{H^5 \log(SAT)}{\Delta_h(s,a)} + \frac{H^6 S\log(SAT)}{\Delta_{\min}}+ SAH^3\right),
> $$
> whereas Equation (10) reduces to a better bound of
> $$
> O\left(\sum_{h=1}^H \sum_{\Delta_h(s,a)>0} \frac{H^5 \log(SAT)}{\Delta_h(s,a)}\right)
> $$
> since in this case $|Z_{\textnormal{mul}}| = 0$. Notably, the latter result removes the dependency on $S / \Delta_{\min}$.
>
> **Comparison between the simplified result for UCB-Hoeffding and Equation (10) for Refined AMB.**
>
> In this subsection, we compare the simplified result for UCB-Hoeffding in line 200 of the original manuscript (line 213 of the revised manuscript)
> $$O\left( \sum_{h = 1}^H\sum_{\Delta_{h}(s,a)>0}\frac{H^5\log(SAT)}{\Delta_{h}(s,a)}+  \frac{H^5|Z_{\textnormal{opt}}|\log(SAT)}{\Delta_{\textnormal{min}}} + SAH^3\right)$$
> with Equation (10) for Refined AMB. To conclude, Equation (10) for Refined AMB is no worse than the simplified result for UCB-Hoeffding since $|Z_{\textnormal{mul}}| \leq |Z_{\textnormal{opt}}|$.
>
> However, when there are multiple optimal state-action-step triples, such that $|Z_{\textnormal{opt}}| \geq 2HS$, we have $|Z_{\textnormal{mul}}| \geq |Z_{\textnormal{opt}}| / 2$. This is because
> $$
> |Z_{\textnormal{opt}}| - |Z_{\textnormal{mul}}| = |Z_{\textnormal{opt}} / Z_{\textnormal{mul}}|
> = |\{(s,a,h): \Delta_h(s,a) = 0, |Z_{\textnormal{opt},h}(s)| = 1\}| \le HS,
> $$
> since for each $(s,a,h) \in Z_{\textnormal{opt}} / Z_{\textnormal{mul}}$, we have $|Z_{\textnormal{opt},h}(s)| = 1$, which implies that the optimal action $a$ is unique for each state-step pair $(s,h)$. In this case, the simplified result for UCB-Hoeffding coincides with Equation (10) of Refined AMB.
>
> **Reference:**
>
> [1] Simchowitz, Max, and Kevin G. Jamieson. "Non-asymptotic gap-dependent regret bounds for tabular mdps." Advances in Neural Information Processing Systems 32 (2019).
>
> [2] Dann, Christoph, et al. "Beyond value-function gaps: Improved instance-dependent regret bounds for episodic reinforcement learning." Advances in Neural Information Processing Systems 34 (2021): 1-12.
>
> [3] Chen, Shulun, et al. "Sharp Gap-Dependent Variance-Aware Regret Bounds for Tabular MDPs." arXiv preprint arXiv:2506.06521 (2025).
>
> [4] Yang, Kunhe, Lin Yang, and Simon Du. "Q-learning with logarithmic regret." International Conference on Artificial Intelligence and Statistics. PMLR, 2021.
>
> [5] Jin, Chi, et al. "Is Q-learning provably efficient?." Advances in neural information processing systems 31 (2018).
>
> [6] Zhang, Zihan, Yuan Zhou, and Xiangyang Ji. "Almost optimal model-free reinforcement learning via reference-advantage decomposition." Advances in Neural Information Processing Systems 33 (2020): 15198-15207.
>
> [7] Li, Gen, et al. "Breaking the sample complexity barrier to regret-optimal model-free reinforcement learning." Advances in Neural Information Processing Systems 34 (2021): 17762-17776.
>
> [8] Zhang, Haochen, Zhong Zheng, and Lingzhou Xue. "Regret-Optimal Q-Learning with Low Cost for Single-Agent and Federated Reinforcement Learning." arXiv preprint arXiv:2506.04626 (2025).
>
> [9] Zheng, Zhong, Haochen Zhang, and Lingzhou Xue. "Gap-Dependent Bounds for Q-Learning using Reference-Advantage Decomposition." The Thirteenth International Conference on Learning Representations.
>
> [10] Xu, Haike, Tengyu Ma, and Simon Du. "Fine-grained gap-dependent bounds for tabular mdps via adaptive multi-step bootstrap." Conference on Learning Theory. PMLR, 2021.

---

> ### Author Response · Authors · 2025-11-27
> **Following up on the rebuttal**
>
> Dear Reviewer Er67,
>
> We hope that our responses and revised manuscript have adequately addressed your concerns regarding (1) the comparison with model-based works, (2) the $H$-dependency and potential improvement, (3) the significance of ULCB-Hoeffding, and (4) the comparison between the UCB-Hoeffding and AMB results. If you have additional questions or concerns, please let us know, and we would be happy to provide further clarification. We truly appreciate your feedback and look forward to hearing from you soon.
>
> Best regards,
>
> Authors of Submission 11791

---

### Official Review · Reviewer_F9mz · 2025-10-31

**Soundness:** 2
**Presentation:** 2
**Contribution:** 4
**Rating:** 4
**Confidence:** 4

**Summary:**

This paper studies the online tabular RL problem.
The authors show that UCB-Hoeffding (Jin et al., 2018) achieves a fine-grained, gap-dependent logarithmic regret bound.
Such a result is the first among model-free algorithms.
While there is an exception of AMB (Xu et al., 2021), the authors point out errors in the analysis of AMB and propose how to fix it, providing a correct fine-grained regret bound and its proof for the algorithm.

**Strengths:**

The paper proposes a general framework for obtaining fine-grained, gap-dependent logarithmic regret bounds of model-free algorithms.

The raised issues for AMB and the corrected version seem valid.

**Weaknesses:**

Some core definitions for the analysis are not properly given.

- Is $\eta_ i^{N}$, introduced in line 310, $\eta_ i \Pi_ {j=i+1}^N (1 - \eta_ j)$ or the $N$-th power of $\eta_ i$? It seems like it's the former but I could not find the definition.

- What is the input state-action pair of $N_ h^k$ when written without one? It appears multiple times including in the definition of $\omega_ {h'+1}^k(h)$, but it is not clear from the context.

- It seems like the definition of $\omega_ {h'+1}^k(h)$ requires a state-action pair as $k^i$ requires a state-action pair in its definition. However, the notation does not reflect the fact. Also, there is a problem of using $N_ h^k$ without definition, so I have no idea what $\omega_ {h'}^k(h)$ is supposed to represent.
I hope there is a description about what $\omega_ {h'+1}^k(h)$ represents as it is hard to understand it intuitively.
Partially due to these ambiguities, Lemma C.3 is not trivial. Also, it is really hard to see why the equation in line 372 is true. Could the authors explain how these equations are established?
In addition, as I see that $\omega_ h^k = \mathbb{I}\lbrace (s_ h^k, a_ h^k) \in Z_ {\text{sub}, h}\rbrace$ in the end, which is a simple function, wouldn't plugging in this value from the beginning simplify the analysis a lot without needing to define multiple series of $\omega$?

I will be happy to raise my score once these points are clarified.

**Questions:**

1. While a general framework for fine-grained, gap-dependent bounds for model-based algorithms is known (Simchowitz & Jamieson, 2019; Dann et al., 2021; Chen et al., 2025), I see that the analysis in this paper is different from the one in Simchowitz & Jamieson (2019). What challenges are there in applying the techniques in these papers to the model-free setting?

2. I don't understand why UCLB is proposed when it is no better than UCB-Hoeffding both theoretically and empirically. If the goal is to show the generality of the framework, couldn't any other model-free algorithm be used, for instance, UCB-Bernstein or UCB-Advantage?

3. In Section 4, it is mentioned that the analysis of Xu et al. (2021) violates the martingale property. It is due to the fact that $h'$ in their work is also random? It is not clear what this part is trying to claim. For instance, what are the values of the expectation and conditional expectation described in this paper?

4. Under what MDP was the experiment conducted?

---

> ### Author Response · Authors · 2025-11-20
> **Response to Reviewer F9mz (part one)**
>
> We thank you for your careful reading and thoughtful comments. Below are our point-by-point responses; we hope they address your concerns.
>
> **Weakness 1: Clarification of the definition $\eta_i^N$**
>
> For $\eta_i = \frac{H+1}{H+i}$, denote $\eta_0^0 = 1$, $\eta^N_0 = 0$ for $N\geq 1,$ and $\eta^N_i = \eta_i\prod_{i'=i+1}^N(1-\eta_{i'}), \forall \ 1\leq i\leq N$. This definition corresponds to the first form you mentioned and appears on line 765 in our original manuscript. In our revised manuscript, we have moved it to the main paper at line 181 to make the definition clearer.
>
> **Weaknesses 2 and 3: Clarification of the recursive weights $\omega_{h'+1}^k(h)$ (part one)**
>
> **Full definitions of the recursive weights**
>
> Due to page limitations, the original paper uses the shorthand notations $k^i$ and $N_h^k$. We appreciate your comment and provide further clarification of these shorthand notations and the recursively defined weights as follows. These clarifications are also included in lines 338–359 of our revised manuscript.
>
> For any given step $h$ and non-negative weight sequence $\\{\omega\_h^k\\}\_{k=1}^K$, we recursively define
> $$ \omega_h^k(h) := \omega_h^k;\
>     \omega_{h'+1}^k(h) := \sum_{k'=1}^K \sum_{i=1}^{N_{h'}^{k'}(s_{h'}^{k'},a_{h'}^{k'})} \omega_{h'}^{k'}(h) \eta_i^{N_{h'}^{k'}(s_{h'}^{k'},a_{h'}^{k'})}  \mathbb{I}\left[k^i(s_{h'}^{k'},a_{h'}^{k'},h') = k\right], \ \forall k \in [K],\ h \leq h' < H \quad (1)$$
> and the norms
> $$
> \|\omega(h)\|\_{\infty, h'} := \max_{k \in [K]} \omega_{h'}^k(h), \quad
> \|\omega(h)\|\_{1, h'} := \sum_{k=1}^K \omega_{h'}^k(h).$$
> Here, $k^i(s_{h'}^{k'},a_{h'}^{k'},h')$, with its shorthand $k^i$, denotes the episode index of the $i$-th visit to $(s_{h'}^{k'},a_{h'}^{k'},h')$, and $ N_{h'}^{k'}(s_{h'}^{k'},a_{h'}^{k'})$, with its shorthand $N_{h'}^{k'}$, denotes the number of visits to $(s_{h'}^{k'},a_{h'}^{k'},h')$ before episode $k'$. Based on this full definition, $\omega_{h'}^k(h)$ is defined solely based on the initial weight sequence $\\{\omega_h^k\\}\_{k=1}^K$ at step $h$, the current step index $h'$, and the episode indices $k = 1,2\ldots K$.

---

> ### Author Response · Authors · 2025-11-20
> **Response to Reviewer F9mz (part two)**
>
> **Weaknesses 2 and 3: Clarification of the recursive weights $\omega_{h'+1}^k(h)$ (part two)**
>
> **Intuition behind the recursive weights**
>
> This recursive weight is extracted from our regret analysis when we try to control the cumulative weighted $Q$-estimation errors $\sum_{k = 1}^K \omega_{h}^{k} \left(Q_h^k - Q_h^\star\right)(s_h^{k}, a_h^{k})$ by recursions between steps. The detailed derivation is provided in Lemma D.3, and we also include a brief sketch below to illustrate the key idea.
>
> As provided in Lemma D.1, UCB-Hoeffding establishes the following relationship for the estimation error $(Q_h^k-Q_h^\star)(s_h^{k}, a_h^{k})$ of the $Q$-estimate $Q_h^k(s_h^k,a_h^k)$ with the cumulative bonus $\beta_{N_h^k(s_h^{k}, a_h^{k})}$ with high probability:
> $$(Q_h^k-Q_h^\star)(s_h^{k}, a_h^{k})\leq \sum_{i = 1}^{N_h^k(s_h^{k}, a_h^{k})}\eta_i^{N_h^k(s_h^{k}, a_h^{k})}\left(V_{h+1}^{k^i(s_h^{k}, a_h^{k},h)} (s_{h+1}^{k^i})- V_{h+1}^\star\right)(s_{h+1}^{k^i}) + \beta_{N_h^k(s_h^{k}, a_h^{k})}.$$
> This leads to the following relationship
> $$\sum_{k = 1}^K \omega_{h}^{k} \left(Q_h^k - Q_h^\star\right)(s_h^{k}, a_h^{k})\leq  \sum_{k = 1}^K \omega_{h}^{k} \sum_{i = 1}^{N_h^k(s_h^{k}, a_h^{k})}\eta_i^{N_h^k(s_h^{k}, a_h^{k})}(V_{h+1}^{k^i(s_h^{k}, a_h^{k},h)} - V_{h+1}^\star)(s_{h+1}^{k^i}) + \sum_{k = 1}^K \omega_{h}^{k}(\eta_0^{N_h^k}H + \beta_{N_h^k(s_h^{k}, a_h^{k})}).\quad (2)$$
> Focusing on the first term of the RHS of (2), we further have
> $$
> \begin{aligned}
>     \sum_{k = 1}^K \omega_{h}^{k} \sum_{i = 1}^{N_h^k(s_h^{k}, a_h^{k})}\eta_i^{N_h^k(s_h^{k}, a_h^{k})}(V_{h+1}^{k^i(s_h^{k}, a_h^{k},h)} - V_{h+1}^\star)(s_{h+1}^{k^i(s_h^{k}, a_h^{k},h)})\\
>     = \sum_{k = 1}^K \omega_{h}^{k} \sum_{i = 1}^{N_h^k(s_h^{k}, a_h^{k})}\eta_i^{N_h^k(s_h^{k}, a_h^{k})}(V_{h+1}^{k^i(s_h^{k}, a_h^{k},h)} - V_{h+1}^\star)(s_{h+1}^{k^i}) \left(\sum_{k' = 1}^K \mathbb{I}[k^i(s_h^{k}, a_h^{k},h) = k']\right) \nonumber\\
>     = \sum_{k' = 1}^K (V_{h+1}^{k'} - V_{h+1}^\star)(s_{h+1}^{k'})\left(\sum_{k = 1}^K\sum_{i = 1}^{N_h^k}\omega_{h}^{k}\eta_i^{N_h^k(s_h^{k}, a_h^{k})}\mathbb{I}[k^i(s_h^{k}, a_h^{k},h) = k'] \right)\\
>     \leq \sum_{k' = 1}^K (Q_{h+1}^{k'} - Q_{h+1}^\star)(s_{h+1}^{k'},a_{h+1}^{k'})\left(\sum_{k = 1}^K\sum_{i = 1}^{N_h^k(s_h^{k}, a_h^{k})}\omega_{h}^{k}\eta_i^{N_h^k(s_h^{k}, a_h^{k})}\mathbb{I}[k^i(s_h^{k}, a_h^{k},h) = k'] \right)\\
>     = \sum_{k' = 1}^K \omega_{h + 1}^{k'}(h) (Q_{h+1}^{k'} - Q_{h+1}^\star)(s_{h+1}^{k'},a_{h+1}^{k'})
> \end{aligned}
> $$
> with
> $$\omega_{h + 1}^{k'}(h) := \sum_{k = 1}^K\sum_{i = 1}^{N_h^k(s_h^{k}, a_h^{k})}\omega_{h}^{k}\eta_i^{N_h^k(s_h^{k}, a_h^{k})}\mathbb{I}[k^i(s_h^{k}, a_h^{k},h) = k'].$$
> The inequality here is because $Q_{h+1}^{k'}(s_{h+1}^{k'},a_{h+1}^{k'}) \geq V_{h+1}^{k'}(s_{h+1}^{k'})$ by the update rule in line 7 of Algorithm 1 and $Q_{h+1}^{\star}(s_{h+1}^{k'},a_{h+1}^{k'}) \leq V_{h+1}^{\star}(s_{h+1}^{k'})$ by Bellman optimality equation. Thus, we have
> $$\sum_{k = 1}^K \omega_{h}^{k} \left(Q_h^k - Q_h^\star\right)(s_h^{k}, a_h^{k})\leq  \sum_{k' = 1}^K \omega_{h + 1}^{k'}(h) (Q_{h+1}^{k'} - Q_{h+1}^\star)(s_{h+1}^{k'},a_{h+1}^{k'}) + \sum_{k = 1}^K \omega_{h}^{k}(\eta_0^{N_h^k}H + \beta_{N_h^k(s_h^{k}, a_h^{k})}).\quad (3)$$
> Equation (3) establishes a recursive relationship between the cumulative weighted estimation errors at steps $h$ and $h+1$. By the same argument, starting from step $h+1$, we can derive a similar relationship for any pair of consecutive steps $h'$ and $h'+1$, up to step $H$. The recursively defined weight $\omega_{h'+1}^k(h)$ explicitly characterizes the weight of the term $(Q_{h'+1}^{k} - Q_{h'+1}^\star)(s_{h'+1}^{k}, a_{h'+1}^{k})$ in the recursive propagation from step $h'$ to $h'+1$.

---

> ### Author Response · Authors · 2025-11-20
> **Response to Reviewer F9mz (part three)**
>
> **Weaknesses 2 and 3: Clarification of the recursive weights $\omega_{h'+1}^k(h)$ (part three)**
>
> **Explanation of Lemma C.3**
>
> We now use the definitions above to explain Lemma C.3. The explanation has also been added in the revised version at line 856 for Lemma C.3.
>
> For Lemma C.3, (a) is because
> $$
> \begin{aligned}
>     \sum_{s,a} \tilde{\omega}\_{h+1}^k(h, s, a) &= \sum_{s,a \in \mathcal{S} \times \mathcal{A}} \sum_{k'=1}^K \omega_h^{k'} \sum_{i=1}^{N_h^{k'}(s_{h}^{k'},a_{h}^{k'})} \eta_i^{N_h^{k'}}  \mathbb{I}\left[k^i(s_{h}^{k'},a_{h}^{k'},h) = k,\ (s_h^{k'}, a_h^{k'}) = (s, a)\right] \\
>     =   \sum_{k'=1}^K \omega_h^{k'} \sum_{i=1}^{N_h^{k'}(s_{h}^{k'},a_{h}^{k'})} \eta_i^{N_h^{k'}} \mathbb{I}\left[k^i(s_{h}^{k'},a_{h}^{k'},h) = k\right] = \omega_{h+1}^k(h).
> \end{aligned}
> $$
> (b) is because for any $k \in [K]$
> $$\omega_{h'}^k(h, s, a) := \omega_{h'}^k(h) \cdot \mathbb{I}\left[(s_{h'}^k, a_{h'}^k) = (s, a)\right]\leq \omega_{h'}^k(h) \leq \|\omega(h)\|\_{\infty, h'}.$$
> (c) is because
> $$\|\omega(h, s, a)\|\_{1, h'} = \sum_{k=1}^K \omega_{h'}^k(h, s, a) \leq \|\omega(h)\|\_{\infty,h'}\sum_{k=1}^K\mathbb{I}\left[(s_{h'}^k, a_{h'}^k) = (s, a)\right] = \|\omega(h)\|\_{\infty,h'}N_{h'}^{K+1}(s,a).$$
> (d) is because
> $$\sum_{s,a} \|\omega(h, s, a)\|\_{1, h'} = \sum_{s,a} \sum_{k=1}^K \omega_{h'}^k(h) \cdot \mathbb{I}\left[(s_{h'}^k, a_{h'}^k) = (s, a)\right] = \sum_{k=1}^K \omega_{h'}^k(h) = \|\omega(h)\|\_{1, h'}.$$
> We also restate the proof of (e) in Lemma C.3 as follows:
>
> Note that
> $$
> \begin{aligned}
>    \|\omega(h)\|\_{1,h' + 1} &= \sum_{k = 1}^K\omega_{h'+1}^{k}(h) = \sum_{k'=1}^K \sum_{i=1}^{N_{h'}^{k'}(s_{h'}^{k'},a_{h'}^{k'})} \omega_{h'}^{k'}(h) \eta_i^{N_{h'}^{k'}(s_{h'}^{k'},a_{h'}^{k'})} \sum_{k = 1}^K \mathbb{I}\left[k^i(s_{h'}^{k'},a_{h'}^{k'},h') = k\right] \\
>    =\sum_{k' = 1}^K\omega_{h'}^{k'}(h)\left(\sum_{i = 1}^{N_{h'}^{k'}(s_{h'}^{k'},a_{h'}^{k'})}\eta_i^{N_{h'}^{k'}(s_{h'}^{k'},a_{h'}^{k'})}\right)\leq \sum_{k' = 1}^K\omega_{h'}^{k'}(h) = \|\omega(h)\|_{1,h'}.
> \end{aligned}
> $$
> The inequality is due to the property of $\eta_i^N$ in line 823 of the revised manuscript.
>
> For any $k \in [K]$, we also have
> $$
> \begin{aligned}
>    \omega_{h' + 1}^{k}(h) &= \sum_{k' = 1}^K\omega_{h'}^{k'}(h)\sum_{i =1}^{N_{h'}^{k'}(s_{h'}^{k'},a_{h'}^{k'})}\eta_i^{N_{h'}^{k'}(s_{h'}^{k'},a_{h'}^{k'})}\mathbb{I}\left[k^i(s_{h'}^{k'},a_{h'}^{k'},h') = k\right]\\
>    \leq \|\omega(h)\|\_{\infty,h'}\sum_{k' = 1}^K\sum_{i = 1}^{N_{h'}^{k'}(s_{h'}^{k'},a_{h'}^{k'})}\eta_i^{N_{h'}^{k'}(s_{h'}^{k'},a_{h'}^{k'})}\mathbb{I}\left[k^i(s_{h'}^{k'},a_{h'}^{k'},h')= k \right].
> \end{aligned}
> $$
>
> Here $\mathbb{I}\left[k^i(s_{h'}^{k'},a_{h'}^{k'},h') = k\right] = 1$ if and only if $(s_{h'}^{k'},a_{h'}^{k'}) = (s_{h'}^k,a_{h'}^k)$, $k \leq k'-1$ and $i = N_{h'}^{k+1}(s_{h'}^{k},a_{h'}^{k})$. Then for any $k \in [K]$, we have:
> $$
> \begin{aligned}
>     &\sum_{k' = 1}^K\sum_{i = 1}^{N_{h'}^{k'}(s_{h'}^{k'},a_{h'}^{k'})}\eta_i^{N_{h'}^{k'}(s_{h'}^{k'},a_{h'}^{k'})}\mathbb{I}\left[k^i(s_{h'}^{k'},a_{h'}^{k'},h')= k \right]\\
>     \leq \sum_{k'=k+1}^K\eta_{N_{h'}^{k+1}(s_{h'}^{k},a_{h'}^{k})}^{{N_{h'}^{k'}}(s_{h'}^{k},a_{h'}^{k})}\mathbb{I}\left[(s_{h'}^{k'},a_{h'}^{k'}) = (s_{h'}^{k},a_{h'}^{k}) \right] \leq \sum_{t=N_{h'}^{k'+1}}^{\infty}\eta_{N_{h'}^{k+1}(s_{h'}^{k},a_{h'}^{k})}^{t} \leq 1+\frac{1}{H}.
> \end{aligned}
> $$
> The last inequality is by (a) of Lemma C.2. Therefore, it holds that
> $$\|\omega(h)\|\_{\infty,h' + 1} \leq \left(1+\frac{1}{H}\right)\|\omega(h)\|\_{\infty,h'}.$$
> This proves (e) of Lemma C.3.

---

> ### Author Response · Authors · 2025-11-20
> **Response to Reviewer F9mz (part four)**
>
> **Weaknesses 2 and 3: Clarification of the recursive weights $\omega_{h'+1}^k(h)$ (part four)**
>
> **Explanation of the equation in line 372**
>
> We now explain the equation in line 372 of the original manuscript (line 414 in the revised manuscript). The explanation has also been added in the revised version at line 856 at line 1184.
>
> For this equation, we aim to bound the term
> $$\sum_{k' = 1}^K \omega_{h + 1}^{k'}(h) (Q_{h+1}^{k'} - Q_{h+1}^\star)(s_{h+1}^{k'},a_{h+1}^{k'})$$ in line 418 of the revised manuscript. Here, for any $k \in [K]$, $\omega_{h+1}^k(h)$ is defined by the sequence in Equation (1) at step $h+1$, given any initial weight sequence $\\{\omega_h^k\\}\_{k=1}^K$ at step $h$.
>
> We then use Lemma 3.3 at step $h+1$ to bound the term
> $\sum_{k' = 1}^K \omega_{h + 1}^{k'}(h) (Q_{h+1}^{k'} - Q_{h+1}^\star)(s_{h+1}^{k'},a_{h+1}^{k'})$.
> For any $k \in [K]$, the recursive weight sequence considered in this process starts from step $h+1$ with initial weight $\omega_{h+1}^k(h)$:
> $$ \omega_{h+1}^k(h+1) := \omega_{h+1}^k(h); \quad
>     \omega_{h'+1}^k(h+1)  := \sum_{k'=1}^K \sum_{i=1}^{N_{h'}^{k'}(s_{h'}^{k'},a_{h'}^{k'})} \omega_{h'}^{k'}(h+1)\eta_i^{N_{h'}^{k'}(s_{h'}^{k'},a_{h'}^{k'})} \mathbb{I}\left[k^i(s_{h'}^{k'},a_{h'}^{k'},h') = k\right], \ \forall k \in [K],\ h+1 \leq h' < H.$$
> For any $k \in [K]$, since the initial weights satisfy $\omega_{h+1}^k(h+1) = \omega_{h+1}^k(h)$ and the sequences
> $\\{\omega_{h'}^k(h+1)\\}\_{h' = h+1}^H$ and $\\{\omega_{h'}^k(h)\\}\_{h' = h+1}^H$ follow the same recursive relation, it follows by induction that
> $$ \omega_{h'}^k(h+1) = \omega_{h'}^k(h), \quad h+1 \leq h' \leq H$$
> for these two sequences. Thus, we have that for any $h+1 \leq h' \leq H$:
> $$\|\omega(h+1)\|\_{\infty,h'} = \max_{k \in [K]} \omega_{h'}^k(h+1) = \max_{k \in [K]} \omega_{h'}^k(h)= \|\omega(h)\|\_{\infty,h'}$$
> and
> $$\|\omega(h+1,s,a)\|\_{1,h'} = \sum_{k=1}^K\omega_{h'}^k(h+1) \cdot \mathbb{I}\left[(s_{h'}^k, a_{h'}^k) = (s,a)\right] = \sum_{k=1}^K\omega_{h'}^k(h) \cdot \mathbb{I}\left[(s_{h'}^k, a_{h'}^k) = (s,a)\right] = \|\omega(h,s,a)\|\_{1,h'}.$$
>
> **Incorporating the true value**
>
> Thank you for the comment. First, our framework defines recursive weights. Even when we plug in the “true” values, it does not eliminate the need to compute the weights at each step using the recursive relationship, and we still cannot obtain a simpler expression. Using the true values would also obscure the recursive structure that we highlighted in the Intuition section above. Second, the lemmas (Lemma 3.2 and 3.3) we develop regarding these general recursive weights are of independent interest and can be applied to other RL problems. In other works [1, 2, 3], the weights are not necessarily set to these specific values. Therefore, our framework provides a general tool that goes beyond the single instantiation used in this paper.

---

> ### Author Response · Authors · 2025-11-20
> **Response to Reviewer F9mz (part five)**
>
> **Question 1: Technical Challenges in adapting from model-based work**
>
> All model-based works [4,5,6] first use the following relationship to bound the regret:
> $$\mathbb{E}(\textnormal{Regret}(T)) \leq \mathbb E\left[\sum_{k=1}^K\sum_{h=1}^H E_h^k(s,a)\right],$$
> where
> $$E_h^k(s,a)=Q_h^k(s,a)-(r_h(s,a)+\mathbb E_{s'\sim P_{s,a,h}}[V_{h+1}^k(s')])$$
> is the surplus at $(s,a,h,k)\in \mathcal{S} \times\mathcal{A}\times[H]\times[K]$. They then bound this surplus using the suboptimality gap $\Delta_h(s,a)$.
>
> For all these model-based works, the Q-estimates are updated as
> $$Q_h^k(s,a) = r_h(s,a) + \mathbb{E}\_{s'\sim \hat{P}\_{s,a,h}^k}[V_{h+1}^k(s')] + b_h^k(s,a),$$
> where $\hat{P}\_{s,a,h}^k$ is the empirical estimate of the transition kernel $P_{s,a,h}$ at the beginning of episode $k$. Plugging in this update rule gives
> $$E_h^k(s,a)=b_h^k(s,a)+\mathbb{E}\_{s'\sim \hat{P}\_{s,a,h}^k}[V_{h+1}^k(s')]-\mathbb E_{s'\sim P_{s,a,h}}[V_{h+1}^k(s')]),$$
> where last two terms $\mathbb{E}\_{s'\sim \hat{P}\_{s,a,h}^k}[V_{h+1}^k(s')]$, and $\mathbb E_{s'\sim P_{s,a,h}}[V_{h+1}^k(s')])$  have the same input $V_{h+1}^k$ and $\hat{P}\_{s,a,h}^k$ serves as an unbiased estimate of $P_{s,a,h}$. Therefore, the difference can be bounded by concentration inequalities and the empirical process.
>
> The situation is different for model-free methods, since they do not maintain an empirical estimate $\hat{P}\_{s,a,h}^k$ of the transition kernel. For UCB-Hoeffding, the $Q$-estimates are updated as
> $$Q_h^k(s,a) = r_h(s,a) + \sum_{i = 1}^{N_h^k(s_h^{k}, a_h^{k})}\eta_i^{N_h^k(s_h^{k}, a_h^{k})}V_{h+1}^{k^i(s_h^{k}, a_h^{k},h)} (s_{h+1}^{k^i}) + \beta_{N_h^k(s_h^{k}, a_h^{k})},$$
> and accordingly,
> $$E_h^k(s,a)=b_h^k(s,a)+\sum_{i = 1}^{N_h^k(s_h^{k}, a_h^{k})}\eta_i^{N_h^k(s_h^{k}, a_h^{k})}\left(V_{h+1}^{k^i(s_h^{k}, a_h^{k},h)} (s_{h+1}^{k^i})-\mathbb E_{s'\sim P_{s,a,h}}[V_{h+1}^k(s')]\right).$$
> The inputs of the two terms of the difference above are different: one uses the historical value estimate $V_{h+1}^{k^i}$ and the other uses the current estimate $V_{h+1}^k$. This mismatch introduces additional bias, which prevents the framework used in model-based works from being directly adapted to the model-free setting. Thus, the subsequent analysis in those works does not carry over, and a completely different framework is required for the model-free setting. Next, we explain each component of our framework.
>
> 1. We first establish Lemma 3.1, which bounds the regret by the cumulative weighted visitation counts $\sum_{h=1}^H \sum_{s,a} \Delta_h(s,a) N_h^{K+1}(s,a)$, and we further relate this quantity to the cumulative weighted estimation errors $\sum_{k=1}^K \omega_h^k (Q_h^k - Q_h^\star)(s_h^k, a_h^k)$.
>
> 2. We then bound the cumulative weighted estimation errors by establishing a recursive relationship between consecutive steps (Lemma 3.2) and propagating it to the final step $H$ (Lemma 3.3).
>
> 3. Using Lemmas 3.2 and 3.3, we construct a recursive relationship for the cumulative weighted visitation counts $\sum_{h=1}^H \sum_{s,a} \Delta_h(s,a) N_h^{K+1}(s,a)$ across steps, which enables an inductive argument to derive a fine-grained upper bound and subsequently bound the expected regret via Lemma 3.1.
>
> These innovations form the core of our fine-grained analytical framework and can be applied to a broad class of model-free RL algorithms.

---

> ### Author Response · Authors · 2025-11-20
> **Response to Reviewer F9mz (part six)**
>
> **Question 2: Significance of ULCB-Hoeffding and potential extension to other algorithms**
>
> To fully highlight the significance of incorporating ULCB-Hoeffding, we have rewritten the introduction in lines 55-60 and 100-110 of the revised manuscript, as summarized in the discussion below.
>
> The key novelty in the design of the AMB algorithm lies in two components: ULCB and multi-step bootstrapping. In particular, ULCB refers to leveraging both UCB and Lower Confidence Bound (LCB) techniques to select actions by maximizing the width of the confidence interval rather than maximizing the $Q$-estimates.
>
> UCB-Hoeffding cannot determine whether the optimal action has been learned, as it only maintains an upper-bound estimate of the optimal $Q$-function. Compared to UCB-Hoeffding, the action elimination technique (see line 14 of Algorithm 2 in our paper or line 23 in the original AMB algorithm), aided by the lower-bound Q-estimates in ULCB, enables the agent to determine, at each step and for each state, whether the optimal action has been learned. This not only improves the explainability of the algorithm but may also be of independent interest (see Section 1.2.2 in [7] for further discussion).
>
> In this paper, we incorporate ULCB-Hoeffding to demonstrate that, even without multi-step bootstrapping, which was incorrectly handled in the proof for the original AMB, our framework can be used to show that an algorithm employing the ULCB technique alone can achieve a fine-grained regret, thereby illustrating the generality of our framework. ULCB-Hoeffding can also be regarded as a UCB-based refinement of the original AMB algorithm.
>
> We may extend our framework to more complicated algorithms to show generality, such as UCB-Bernstein [8], UCB-Advantage [9], Q-EarlySettled-Advantage [10], and Q-EarlySettled-LowCost [3]. All of them employ variance-based bonuses, which are tighter than those of UCB-Hoeffding, and the latter three additionally use reference-advantage decomposition. As a result, they achieve better worst-case regret [3,8,9,10] and coarse-grained gap-dependent regret [2,3] than UCB-Hoeffding [1,8], and may potentially lead to improved fine-grained results.
>
> However, incorporating variance-based bonuses and reference-advantage decomposition into our framework introduces technical challenges. To achieve improved results, we need to handle different bonuses for different state--action--step triples. In contrast, for UCB-Hoeffding, different state--action--step triples with the same number of visits share the same bonus. This difference requires additional analysis to build the recursive structure we established for $\sum_{s,a} \Delta_h(s,a) N_h^{K+1}(s,a)$ at different steps in Lemma D.4.
>
> Moreover, [2,3] derive a coarse-grained gap-dependent regret for algorithms with reference-advantage decomposition by introducing a surrogate reference function. Incorporating this surrogate reference function into our framework also requires additional technical effort.

---

> ### Author Response · Authors · 2025-11-20
> **Response to Reviewer F9mz (part seven)**
>
> **Question 3: Violation of Martingale Condition**
>
> The main issue lies in certain flawed definitions in the proof of AMB, which result in a violation of the martingale condition. Correcting this introduces additional challenges due to the randomness associated with the bootstrapped step $h'$. See Appendix F.1 and the discussion below for details.
>
> AMB maintains upper and lower bounds $\overline{Q}_h^k(s, a)$ and $\underline{Q}_h^k(s, a)$ for each state-action-step triple $(s, a, h)$ at the beginning of episode $k$. The policy $\pi^k$ is selected by maximizing the confidence interval length $\overline{Q} - \underline{Q}$. Based on these bounds, for each state $s$ and step $h$, AMB constructs a set of candidate optimal actions, denoted by $A_h^k(s)$, by eliminating any action $a$ whose upper bound is lower than the lower bound of some other action. If $|A_h^k(s)| = 1$, the optimal action is identified, denoted by $\pi_h^\star(s)$, and $s$ is referred to as a decided state; otherwise, $s$ is called an undecided state. Let $G_h^k = \{ s \mid |A_h^k(s)| = 1 \}$ denote the set of all decided states at step $h$ in episode $k$.
>
> Let $h' = h'(k,h) > h$ denote the first index such that $s_{h'}^k \notin G_{h'}^k$. AMB constructs upper and lower bounds of the $Q$-function by decomposing the $Q$-function into two parts: the rewards accumulated within the decided states and those starting from the undecided states. To estimate each part, AMB uses the sum of empirical rewards in episode $k$:
> $$\hat{Q}_h^{k,d}(s, a) = \sum_{l=h}^{h'-1} r_l(s_l^k, a_l^k).$$
> together with the existing upper-bound $V$-estimate $\overline{V}\_{h'}^k(s_{h'}^k)$.
>
> To bound the estimation error, the original AMB incorrectly centers each estimate around their expectation as follows (See Equation 4.2 and Lemma 4.1 in [1]):
> $$\sum_{i = 1}^{N_h^{k}}\eta_i^{N_h^{k}}\left(\hat{Q}\_h^{k^i,d}(s_h^k, a_h^k) - \mathbb{E}(\hat{Q}\_h^{k^i,d}(s_h^k, a_h^k))\right) \quad \textnormal{and} \quad \sum_{i = 1}^{N_h^{k}}\eta_i^{N_h^{k}}\left(V_{h'}^\star(s_{h'}^{k^i}) - \mathbb{E}(V_{h'}^\star(s_{h'}^{k^i})\right).$$
> Here, we use the shorthands $k^i = k^i(s_h^k,a_h^k,h)$ and $N_h^k = N_h^k(s_h^k, a_h^k)$.
>
> To address this issue, we replace the expectations with the corresponding conditional expectations and verify that these conditional expectations collectively form a decomposition of the optimal $Q$-function $Q^\star$, which is the key property assumed by multi-step bootstrapping. While this property naturally holds for one-step Bellman updates due to the Markov property, it remains nontrivial for multi-step bootstrapping because of the randomness introduced by the bootstrapped steps $h'$.
>
> Let $\mathcal{F}\_{h,k}$ denote the filtration generated by the trajectory up to and including step $h$ in episode $k$. In particular, $\mathcal{F}\_{h,k}$ contains the policy $\pi^k$ and the current state-action pair $(s_h^k, a_h^k)$. We bound the estimation error by centering each estimate around its conditional expectation:
> $$\sum_{i = 1}^{N_h^{k}}\eta_i^{N_h^{k}}\left(\hat{Q}\_h^{k^i,d}(s_h^k, a_h^k) - Q_h^{k^i,d}(s_h^k, a_h^k)\right) \quad \textnormal{and} \quad \sum_{i = 1}^{N_h^{k}}\eta_i^{N_h^{k}}\left(V_{h'}^\star(s_{h'}^{k^i}) - Q_h^{k^i,ud}(s_h^k, a_h^k)\right),$$
> where
> $$Q_h^{k,d}(s_h^k, a_h^k) \triangleq \mathbb{E}\ \left[ \sum_{l=h}^{h'-1} r_l(s_l^k, \pi_l^\star(s_l^k)) \mid \mathcal{F}\_{h,k} \right], \quad Q_h^{k,ud}(s_h^k, a_h^k) \triangleq \mathbb{E}\ \left[ V_{h'}^\star(s_{h'}^k) \mid \mathcal{F}\_{h,k}\right].$$
>
> Additionally, we verify that the multi-step bootstrapping induces an unbiased estimate in Lemma F.1:
>     $$Q_h^{k, d}(s_h^k, a_h^k) + Q_h^{k, ud}(s_h^k, a_h^k) = Q_h^\star(s_h^k, a_h^k).$$
>
>
> **Question 4: Experimental Environment**
>
> We conduct our experiments in synthetic environments with randomly generated MDPs, as specified in lines 457-462 of the original manuscript and now in lines 505-511 of the revised manuscript. We consider four \textbf{MDP scales} with $(H,S,A,K) = (2,3,3,10^5), (5,5,5,6 \times 10^5), (7,8,6,5 \times 10^6)$, and $(10,15,10,2 \times 10^7)$. For each $(s, a, h)$, rewards $r_h(s, a)$ are sampled independently from the uniform distribution over $[0,1]$, and transition kernels $\mathbb{P}_h(\cdot \mid s, a)$ are drawn uniformly from the $S$-dimensional probability simplex. The initial state of each episode is selected uniformly at random from the state space.

---

> ### Author Response · Authors · 2025-11-20
> **Response to Reviewer F9mz (part eight)**
>
> **Reference:**
>
> [1] Yang, Kunhe, Lin Yang, and Simon Du. "Q-learning with logarithmic regret." International Conference on Artificial Intelligence and Statistics. PMLR, 2021.
>
> [2] Zheng, Zhong, Haochen Zhang, and Lingzhou Xue. "Gap-Dependent Bounds for Q-Learning using Reference-Advantage Decomposition." The Thirteenth International Conference on Learning Representations.
>
> [3] Zhang, Haochen, Zhong Zheng, and Lingzhou Xue. "Regret-Optimal Q-Learning with Low Cost for Single-Agent and Federated Reinforcement Learning." arXiv preprint arXiv:2506.04626 (2025).
>
> [4] Simchowitz, Max, and Kevin G. Jamieson. "Non-asymptotic gap-dependent regret bounds for tabular mdps." Advances in Neural Information Processing Systems 32 (2019).
>
> [5] Dann, Christoph, et al. "Beyond value-function gaps: Improved instance-dependent regret bounds for episodic reinforcement learning." Advances in Neural Information Processing Systems 34 (2021): 1-12.
>
> [6] Chen, Shulun, et al. "Sharp Gap-Dependent Variance-Aware Regret Bounds for Tabular MDPs." arXiv preprint arXiv:2506.06521 (2025).
>
> [7] Xu, Haike, Tengyu Ma, and Simon Du. "Fine-grained gap-dependent bounds for tabular mdps via adaptive multi-step bootstrap." Conference on Learning Theory. PMLR, 2021.
>
> [8] Jin, Chi, et al. "Is Q-learning provably efficient?." Advances in neural information processing systems 31 (2018).
>
> [9] Zhang, Zihan, Yuan Zhou, and Xiangyang Ji. "Almost optimal model-free reinforcement learning via reference-advantage decomposition." Advances in Neural Information Processing Systems 33 (2020): 15198-15207.
>
> [10] Li, Gen, et al. "Breaking the sample complexity barrier to regret-optimal model-free reinforcement learning." Advances in Neural Information Processing Systems 34 (2021): 17762-17776.

---

> ### Comment · Reviewer_F9mz · 2025-11-23
>
> I sincerely appreciate the authors' effort in providing a detailed explanation regarding the issues I raised. I went through the analysis again with the provided definitions, and now I understand it much better.
>
> ---
>
> **Definition of $\omega_ {h'}^{k}(h)$**
> Thank you for clarifying the definitions. I now fully understand what $\omega_ {h'}^{k}(h)$ is. However, I still feel like there are some unnecessary complications in the notations and the analysis.
>
> If I understand correctly, $\omega_ {h'}^{k'}(h)$ represents the influence (or coefficient) of $(Q_ {h'}^{k'} - Q_ {h'}^{\ast})(s_ {h'}^{k'}, a_ {h'}^{k'})$ in $\sum_ {k=1}^K \omega_ h^{k} (Q_ h^{k} - Q_ h^{\ast})(s_ h^{k}, a_ h^{k})$ when decomposed.
> I suggest adding this kind of written description of $\omega_ {h'}^{k}(h)$, which should greatly help in understanding it intuitively.
> Also, looking into the revised definition of $\omega_ h'^{k}(h)$, it seems like the definition of $\omega_ {h'+1}^{k}$ only regards $(s, a) = (s_ {h'}^{k}, a_ {h'}^{k})$, and is $\omega_ {h'+1}^{k}(h) = \sum_ {i=N_ {h'}^{k}(s, a)}^{N_ {h'}^{K+1}(s, a)} \omega_ {h'}^{k^i(s, a, h')} \eta_ {N^{k}_ {h'}(s, a)}^i$.
> Is this correct? I think the current presentation of using a double sum and an indicator function is unnecessarily complicated and obscures this observation.
>
> Regarding Line 372 (now 414): I now see that $\omega_ {h+1}^{k}$ is defined as $\omega_ {h+1}^{k}(h)$ when analyzing the $h$-th time step, which caused a confusion because I expected it to be $1((s_ {h+1}^{k}, a_ {h+1}^{k}) \in \mathcal{Z}_ {\mathrm{sub}, h+1})$ just as $\omega_ h^{k}$.
> This definition seems to be still missing in the proof of Lemma D.5, especially for line 1181 (revised version).
> The definition appears after, saying "This is because...", when the intended definition of $\omega_ {h'}^k(h+1)$ was not introduced before and it is natural to think it inherits the definition of $\omega_ {h'}^k(h)$ in the same way.
>
> That said, redefining $\omega_ {h+1}^{k}$, and followingly $\omega_ {h'}^k(h+1)$, seems unnecessary, as $\omega_ {h'}^{k}(h)$ plays the exact same role with $\omega_ {h'}^k(h+1)$.
> I understand that its purpose is to apply Lemma 3.2 to time step $h+1$.
> However, I think it only adds confusion since the readers have to memorize the recursive definition of $\omega_ {h'}^{k}(h)$ for $h' \ge h + 2$, then it is replaced by $\omega_ {h'}^{k}(h+1)$ with a newly defined $\omega_ {h+1}^{k} := \omega_ {h+1}^{k}(h)$, then again back to $\omega_ {h'}^{k}(h)$ using induction. Wouldn't it be possible to use the recursive formula more directly and make the analysis clearer?
>
> Regarding these points, I find the explanation given in the discussion (Response to Reviewer F9mz (part two)) easier to follow.
> Maybe sketching the analysis in this way would make it easier to understand.
> Especially, I think if the authors further present a relationship between $\sum_ {k=1}^K \omega_ h^{k}(Q_ h^{k} - Q_ h^{\ast})(s_ h^{k}, a_ h^{k})$ and $\sum_ {k=1}^K \omega_ {h'}^{k}(h) (Q_ {h'}^{k} - Q_ {h'}^{\ast})(s_ {h'}^{k}, a_ {h'}^{k})$ for $h' \ge h + 1$, not only for $h' = h+1$, then it would be clearer to the readers what $\omega_ {h'}^{k}(h)$ represents and how the analysis is done at the high level, and there would be no need to redefine $\omega_ {h'}^{k}(h+1)$.
>
> In addition, I think the difficulties in understanding and memorizing the notations is not only due to their complicated definitions, but it feels like it is also coming from that $\omega_ {h'}^{k}(h)$ looks like a function that maps $h$ to $\mathbb{R}$, but $\omega_ {h'}^{k}$ has no meaning as a function, that is, $\omega_ {h'}^{k}(1), \omega_ {h'}^{k}(2), \dots$ have no relationship with each other.
> Also, it is hard to see the relationship between $h'$ \& $h$, and $k'$ \& $k$, and even worse, $k$ and $k'$ are used interchangeably within the analysis.
> For example, the roles of $k$ with $k'$ are swapped in the proofs of Lemma C.3 (e) and Lemma D.3 from what is presented the definition.
> I might have understood it better if the notations were (for instance) $\omega_ h(k', h')$, and $\lVert \omega_ h(\cdot, h') \rVert_ {1}, \lVert \omega_ h(\cdot, h') \rVert_ {\infty}$ instead.
> I mildly suggest revising the notations.
> In addition, it took me some time to spot the difference between $\tilde{\omega}_ {h+1}^{k}(h, s, a)$ and $\omega_ {h+1}(h, s, a)$. Adding a brief note regarding this point in the manuscript may help.

---

> ### Comment · Reviewer_F9mz · 2025-11-23
>
> **Question 3 (Error in Xu et al. (2021))**
> Thank you for the detailed explanation.
> While I fully understand the point made and agree that it is indeed an error, I think it might be simpler (and more correct) to just say that Xu et al. (2021) did not correctly account for the randomness of $h'$ (in their notation).
> The current "expectation vs. conditional expectation" presentation led me to understand it as an error of using total expectation instead of conditional expectation.
> However, from reading the authors' response and their paper, I am convinced that $\mathbb{E}(\hat{Q}_ h^{k^i, d}(s_ h^k, a_ h^k))$ and $\mathbb{E}(V_ {h'}^{\ast}(s_ {h'}^{k^i}))$ are not even the total expectations.
> Wouldn't it be better to say they took the expectation incorrectly by ignoring the randomness of $h'$?
> The only issue I am raising here is the part presenting the previous method as "centering around its expectation," since it does not seem to be true, even though it is still incorrect in a different way.
>
> ---
>
> I also thank the authors for providing details for my other questions, and my concerns regarding them are resolved.
>
> **Minor Typos**
> Additional sub/super-script of $_ h^k$ after $(s, a)$ in the first term of the equation in Lemma 3.3.
> In Eq. (14), I think either $I_ h^{k}(s, a)$ should be within the parentheses or the summation should be taken outside the parentheses.
> The summation over $(s, a)$ is missing on the right-hand side of line 963 (revised version).

---

> ### Author Response · Authors · 2025-11-26
> **Response to the new comment of Reviewer F9mz**
>
> **Part 1: Simplified definitions of $\omega_{h'}^k(h)$ and additional explanation**
>
> Your statement is almost correct after two adjustments: the upper index should be $N_{h'}^K(s,a)$ rather than $N_{h'}^{K+1}(s,a)$, and $k^i(s,a,h')$ should be replaced by $k^{i+1}(s,a,h')$. We have used the simplified definitions in the revised manuscript. The equivalence between the original definition and the simplified one is established in Equation (11) at lines 868-888 in the revised manuscript. We also added the additional explanation for the relationship between the recursive weight and the error decomposition from our original response in lines 347-349 of our revised manuscript.
>
>
> **Part 2: Clearer notations for better understanding**
>
> In the revised manuscript, we have also adopted your suggested notation for weights and norms, as shown in section 3.3.2. The corresponding proof has been rewritten using the new notations. Specifically, the notation changes are listed as follows:
> $$\omega\_{h'}^k(h) \rightarrow \omega\_{h}(k, h'),\ \omega\_{h'}^k(h,s,a) \rightarrow \omega\_{h}(k, h',s,a), \ \tilde{\omega}\_{h+1}^k(h,s,a) \rightarrow \tilde{\omega}\_{h}(k, h+1, s, a).$$
> Correspondingly,
> $$\\|\omega(h)\\|\_{\infty, h'} \rightarrow \\|\omega_{h}(\cdot,h')\\|\_{\infty} := \max_{k' \in [K]} \omega_{h}(k',h'),\ \\|\omega(h)\\|\_{1, h'} \rightarrow \\|\omega_{h}(\cdot,h')\\|\_{1} := \sum_{k'=1}^K \omega_{h}(k',h'),$$
> and
> $$
> \\|\omega(h, s, a)\\|\_{\infty, h'} \rightarrow \\|\omega_h(\cdot,h', s, a)\\|\_{\infty} := \max_{{k'} \in [K]} \omega_{h}({k'},h', s, a),\
> \\|\omega(h, s, a)\\|\_{1, h'} \rightarrow \\|\omega_h(\cdot,h', s, a)\\|\_{1} := \sum_{{k'}=1}^K \omega_{h}({k'},h', s, a).$$
>
> To improve the presentation of our fine-grained framework discussed in the following part, we also extend the original definition for $\tilde{\omega}\_{h+1}^k(h,s,a)$ to cover all the subsequent steps $h' \in [h, H]$. The new notation and the new definition in the revised manuscript (line 357) are provided as follows:
> $$\tilde{\omega}\_{h}(k',h'+1, s, a) = \omega_{h}(k',h'+1)\mathbb{I}\left[(s_{h'}^{k'}, a_{h'}^{k'}) = (s, a)\right].$$
> It is equivalent to the original definition when $h' = h$ as shown in lines 1079-1989 and is clearly different from
> $$\omega_{h}(k',h', s, a) := \omega_{h}(k',h') \cdot \mathbb{I}\left[(s_{h'}^{k'}, a_{h'}^{k'}) = (s, a)\right].$$
>
> The weight $\tilde{\omega}\_h(k',h'+1,s,a)$ characterizes the weight attached to the term $(Q_{h'+1}^{k'} - Q_{h'+1}^\star)(s_{h'+1}^{k'}, a_{h'+1}^{k'})$ when bounding the cumulative weighted estimation error $\sum_{k = 1}^K \omega_{h}(k,h',s,a) (Q_{h'}^k - Q_{h'}^{\star})(s_{h'}^{k},a_{h'}^{k})$ for each state-action pair $(s, a)$ at step $h'$, as shown in the first conclusion of Lemma 3.2. We have added this explanation in lines 358-360.
>
> **Part 3: Avoid the double assignment of  $\omega_{h+1}^k$ by rewriting our fine-grained framework**
>
> To the double assignment of the weight $\omega_{h+1}^k$, we have rewritten Lemmas 3.2 and 3.3 of our fine-grained framework. Lemma 3.2 now establishes a recursive relationship for the cumulative weighted estimation error
> $$\sum_{k = 1}^K \omega_{h}(k,h',s,a) \left(Q_{h'}^k - Q_{h'}^\star\right)(s_{h'}^{k},a_{h'}^{k})$$
> for each state-action pair $(s,a)$ between steps $h'$ and $h'+1$, which further induces a recursive relationship for the cumulative weighted estimation error
> $$\sum_{k = 1}^K \omega_{h}(k,h') \left(Q_{h'}^k - Q_{h'}^\star\right)(s_{h'}^{k},a_{h'}^{k})$$
> between steps $h'$ and $h'+1$. With this complete recursive structure for any step $h'$, Lemma~3.3 bounds
> $$\sum_{k = 1}^K \omega_{h}(k,h') \big(Q_{h'}^k - Q_{h'}^\star\big)(s_{h'}^{k},a_{h'}^{k})$$
> for all $h' \in [h,H]$, which includes the step $h+1$. It thus eliminates the need to bound this error at step $h+1$ by assigning $\omega_{h+1}^k$ twice. The corresponding proof has also been rewritten to support this claim.
>
>
>
> **Part 4: Error in AMB**
>
> Thank you for the helpful suggestion. In the revised manuscript, we removed the discussion on “conditional expectation vs. total expectation” and, following your suggestion, directly stated that the violation of the martingale condition arises from ignoring the randomness of the bootstrap term in line 477.
>
> **Part 5: Minor issues**
>
> Thank you for your comments. Regarding the first issue, in the original manuscript, we used the shorthand $(s,a)_h^k := (s_h^k, a_h^k)$ due to the page limit; we have now included this explanation in line 365 of the revised manuscript. The other two issues have also been corrected.

---

> > ### Comment · Reviewer_F9mz · 2025-11-28
> >
> > I sincerely appreciate the authors for revision and comments. I have no further concerns or questions (other than a minor one at the end) about the paper. I think this paper is a meaningful contribution in the theory of RL and rate it positively.
> >
> > ---
> >
> > I have one remaining question regarding the last response. I am having trouble understanding why $k^{i+1}$ should be in the definition of $\omega_ h(k', h'+1) $ instead of $k^i$. For instance, isn't $k^{N^K(s, a) + 1}(s, a)$ not well-defined if $(s, a)$ was not taken at the $K$-th episode? Unless index $i$ starts from $0$, (so $k^0, k^1, ...$), I think having $i$ is more natural. Could the authors clarify this point?

---

> > > ### Author Response · Authors · 2025-11-28
> > > **Response to follow-up comment of Reviewer F9mz**
> > >
> > > Thank you very much for your thoughtful comment and for carefully checking the indexing! We agree that the previous upper limit of the summation was confusing. After re-examining the construction, we have corrected the definition by changing the upper index of the summation from $N_{h'}^{K}(s_{h'}^{k'},a_{h'}^{k'})$ to $N_{h'}^{K+1}(s_{h'}^{k'},a_{h'}^{k'})-1$ while keeping the notation $k^{i+1}$. The changes are provided in line 340 of our revised manuscript. For your convenience, the corrected definition is provided as follows:
> > > $$\omega_h(k',h) := \omega_h^{k'};\ \omega_{h}(k',h'+1) := \sum_{i=N_{h'}^{k'+1}(s_{h'}^{k'},a_{h'}^{k'})}^{N_{h'}^{K+1}(s_{h'}^{k'},a_{h'}^{k'})-1}  \omega_{h}(k^{i+1}(s_{h'}^{k'},a_{h'}^{k'},h'),h')\eta_{N_{h'}^{{k'}+1}(s_{h'}^{k'},a_{h'}^{k'})}^{i}.$$
> > > Now, since $N_h^{K+1}(s,a)$ represents the total number of visits to $(s,a,h)$, $k^{i+1}(s,a,h)$ is well defined for $ 0\leq i \leq N_h^{K+1}(s,a) -1$. A detailed proof of the equivalence between the original two-summation form and the revised definition is now provided in lines 864-890 of the revised manuscript.
> > >
> > > In addition, we provide an intuitive explanation for why we keep the notation $k^{i+1}$. We consider the recursion of the weighted estimation error $\omega_h(k,h)(Q_h^k - Q_h^\star)(s_h^k,a_h^k)$ from step $h$ to step $h+1$.
> > > We know that
> > > $$Q_h^k(s_h^{k}, a_h^{k}) = r_h(s_h^{k}, a_h^{k}) + \sum_{j = 1}^{N_h^k(s_h^{k}, a_h^{k})}\eta_j^{N_h^k(s_h^{k}, a_h^{k})}V_{h+1}^{k^j(s_h^{k}, a_h^{k},h)} (s_{h+1}^{k^j}) + \beta_{N_h^k(s_h^{k}, a_h^{k})},$$
> > > and episode $k$ corresponds to the $(N_h^k(s_h^k,a_h^k) + 1)-$th visit for $(s_h^k,a_h^k,h)$ because $N_h^k(s_h^k,a_h^k)$ represents the visiting number before the start of episode $k$. Therefore, using the decomposition above, for any $j\in \{1,2\ldots, N_h^k(s_h^{k}, a_h^{k})\}$, the received weight for $(Q_{h+1}^{k'} - Q_{h+1}^{\star})$ with $k' = k^j(s_h^k,a_h^k,h)$ in $\omega_h(k,h)(Q_h^k - Q_h^\star)(s_h^k,a_h^k)$ will be $\omega_h(k,h) \eta_j^{N_h^k(s_h^{k}, a_h^{k})}$, which further equals
> > > $$\omega_h(k^{N_h^k(s_h^{k}, a_h^{k})+1}(s_h^{k}, a_h^{k},h),h) \eta_j^{N_h^k(s_h^{k}, a_h^{k})}.$$
> > > By substituting $i = N_h^k(s_h^{k}, a_h^{k})$, this clarifies the appearance of $\omega_h(k^{i+1},h')$ and the $\eta_{N_{h'}^{k'+1}}^{i}$ in Equation (5) of our revised manuscript. Thus, the indexing  $k^{i+1}$
> > >  naturally reflects the recursion of weighted errors along successive visits.
> > >
> > > We hope this resolves the concern, and we thank you again for catching the indexing issue.

---

### Official Review · Reviewer_HMT1 · 2025-10-31

**Soundness:** 3
**Presentation:** 2
**Contribution:** 3
**Rating:** 8
**Confidence:** 3

**Summary:**

This work establishes gap-dependent regret bounds for UCB-based model-free RL regarding individual suboptimality gaps $\Delta_h(s,a)$ instead of global one $\Delta_{\min}$. Besides, this work identifies the issues in analyzing the AMB algorithm, which is non-UCB-based model-free algorithm, on incorrectly applying concentration inequalities, and refines the algorithm and the analysis.

**Strengths:**

1 This work establishes the first and tight individual-gap-dependent regret bounds for model--free RL, which improves coarse global-gap-dependent coarse bound in the prior work. The core technical contribution lies in separating the analysis of of optimal and suboptimal state-action pairs.

2 This work refines and fixed the issues in the AMB algorithm and associated analysis, which provides a rigorous fine-grained regret bound for non-UCB-based algorithm.

**Weaknesses:**

1 The analysis is based on episodic tabular MDP. The theoretical guarantees do not extend to complex settings such as linear MDP, MDP with function approximation.

2 Model-based algorithms has shown to achieve fine-grained gap-dependent regret bound. This work addresses a theoretical gap where model-free algorithms were lagging behind model-based algorithms in this setting. The contribution may not be significant.

3 The regret bound has high dependency on the horizon $H$.

**Questions:**

1 There lacks comparison between this work and model-based RL. For example: a) What is the technical difficulty in adapting techniques in model-based algorithms for deriving fine-grained gap-dependent regret bound? b) Comparison on the dependency on $H$, and potential improvement on $H$.

2 What is the advantage of ULCB-Hoeffding over UCB-Hoeffding? They achieve the same fine-grained gap-dependent regret bound and their analysis are also similar. Given that ULCB is considerably more complex, I don't see the necessity of introducing ULCB algorithm in the main context.

3 Comparing the last term in Eqn. (10) and Eqn (2), the non-UCB-based algorithm AMB achieves a better sample complexity in $H$. Is it due to the sharper analysis of AMB?

Minor issues:

1* Line 226 "line 15 in Algorithm 2": line 15 should be replaced.

2* Line 310 Eqn. (5): inside indicator function, should $k$ be $k'$? The recursive definition is hard follow. It would be great if the authors could come up with intuitive way to explain the intuition if exists.

3* Line 951 "(1),." -> "(1)."

---

> ### Author Response · Authors · 2025-11-20
> **Response to Reviewer HMT1 (part one)**
>
> We thank you for your careful reading and thoughtful comments. Below are our point-by-point responses, and we hope our responses address your concerns.
>
> **Weakness 1: Extension to function approximation**
>
> To address this comment, we first review the existing gap-dependent results with function approximation in the RL literature. To our knowledge, for linear MDPs, only three works [1,2,3] have established gap-dependent guarantees. However, all of these results depend solely on the global minimum gap $\Delta_{\min}$, and thus do not provide fine-grained performance bounds. For general function approximation, even such coarse-grained guarantees are still unavailable.
>
> Extending gap-dependent analyses from tabular RL to RL with function approximation is an important yet challenging direction. The main difficulties arise from the continuous state--action space. Although a suboptimality gap can be defined for each state--action pair, how to define a fine-grained result in this setting remains unclear, since the summation $\sum_{s,a:\Delta_h(s,a)>0} \frac{1}{\Delta_h(s,a)}$, which is central to our tabular fine-grained results, can no longer be applied to a continuous state-action space. Whether an integral counterpart serves as an appropriate analogue is an open question.
>
> From a technical standpoint, extending our framework requires relating the regret to the integral of the weighted estimation error $(Q_h^k - Q_h^\star)(s,a)$ over the continuous state--action space. How to establish such a relationship is unknown in the literature, and developing it requires new techniques that go beyond current gap-dependent analyses.
>
> For these reasons, extending fine-grained results from tabular MDPs to linear MDPs or general function approximation is an important but technically challenging open direction for the entire literature, rather than a limitation specific to our approach.

---

> ### Author Response · Authors · 2025-11-20
> **Response to Reviewer HMT1 (part two)**
>
> **Weakness 2, Question 1a and Question 1b.1: Comparison of $H$-dependence and technical challenges to model-based work**
>
> **Comparison of dependency on $H$**
>
> In model-based works, the state-of-the-art gap-dependent bound in [4] achieves a regret of
> $$\hat{O}\left(\sum_{h=1}^H \sum_{\Delta_h(s,a)>0} \frac{H^2}{\Delta_h(s,a)}+\frac{H^2 |Z_{\textnormal{opt}}|}{\Delta_{\min}}+SAH^4 \max\\{S,H\\}\right).$$
> Here, $\hat{O}$ hides variance-dependent terms and logarithmic terms. This result shows better $H$-dependency on gap-dependent terms and similar $H$-dependency in the last term. However, the last term introduces an $S^2$-dependence, which is prohibitive in large-scale MDPs.
>
> This improved $H$-dependency stems from the strong performance of the model-based algorithm MVP [5]. MVP achieves better regret by employing variance-based bonuses and leveraging estimates of the transition kernel for more accurate value function estimates. However, as is typical for model-based methods, using transition-kernel estimates comes at the cost of an increased memory usage $O(S^2AH)$, compared with the $O(SAH)$ of model-free methods.
>
> **Technical difficulties in adapting model-based techniques**
>
> All model-based works [4,6,7] first use the following relationship to bound the regret:
> $$\mathbb{E}(\textnormal{Regret}(T)) \leq \mathbb E\left[\sum_{k=1}^K\sum_{h=1}^H E_h^k(s,a)\right],$$
> where
> $$E_h^k(s,a)=Q_h^k(s,a)-(r_h(s,a)+\mathbb E\_{s'\sim P_{s,a,h}}[V_{h+1}^k(s')])$$
> is the surplus at $(s,a,h,k)\in \mathcal{S} \times\mathcal{A}\times[H]\times[K]$. They then bound this surplus using the suboptimality gap $\Delta_h(s,a)$.
>
> For all these model-based works, the Q-estimates are updated as
> $$Q_h^k(s,a) = r_h(s,a) + \mathbb{E}\_{s'\sim \hat{P}\_{s,a,h}^k}[V_{h+1}^k(s')] + b_h^k(s,a),$$
> where $\hat{P}\_{s,a,h}^k$ is the empirical estimate of the transition kernel $P_{s,a,h}$ at the beginning of episode $k$. Plugging in this update rule gives
> $$E_h^k(s,a)=b_h^k(s,a)+\mathbb{E}\_{s'\sim \hat{P}\_{s,a,h}^k}[V_{h+1}^k(s')]-\mathbb E_{s'\sim P_{s,a,h}}[V_{h+1}^k(s')]),$$
> where last two terms $\mathbb{E}\_{s'\sim \hat{P}\_{s,a,h}^k}[V_{h+1}^k(s')]$, and $\mathbb E\_{s'\sim P_{s,a,h}}[V_{h+1}^k(s')])$ have the same input $V_{h+1}^k$ and $\hat{P}\_{s,a,h}^k$ serves as an unbiased estimate of $P_{s,a,h}$. Therefore, the difference can be bounded by concentration inequalities and the empirical process.
>
> The situation is different for model-free methods, since they do not maintain an empirical estimate $\hat{P}\_{s,a,h}^k$ of the transition kernel. For UCB-Hoeffding, the $Q$-estimates are updated as
> $$Q_h^k(s,a) = r_h(s,a) + \sum_{i = 1}^{N_h^k(s_h^{k}, a_h^{k})}\eta_i^{N_h^k(s_h^{k}, a_h^{k})}V_{h+1}^{k^i(s_h^{k}, a_h^{k},h)} (s_{h+1}^{k^i}) + \beta_{N_h^k(s_h^{k}, a_h^{k})},$$
> and accordingly,
> $$E_h^k(s,a)=b_h^k(s,a)+\sum_{i = 1}^{N_h^k(s_h^{k}, a_h^{k})}\eta_i^{N_h^k(s_h^{k}, a_h^{k})}\left(V_{h+1}^{k^i(s_h^{k}, a_h^{k},h)} (s_{h+1}^{k^i})-\mathbb E_{s'\sim P_{s,a,h}}[V_{h+1}^k(s')]\right).$$
> The inputs of the two terms of the difference above are different: one uses the historical value estimate $V_{h+1}^{k^i}$ and the other uses the current estimate $V_{h+1}^k$. This mismatch introduces additional bias, which prevents the framework used in model-based works from being directly adapted to the model-free setting. Thus, the subsequent analysis in those works does not carry over, and a completely different framework is required for the model-free setting. Next, we explain each component of our framework.
>
> 1. We first establish Lemma 3.1, which bounds the regret by the cumulative weighted visitation counts $\sum_{h=1}^H \sum_{s,a} \Delta_h(s,a) N_h^{K+1}(s,a)$, and we further relate this quantity to the cumulative weighted estimation errors $\sum_{k=1}^K \omega_h^k (Q_h^k - Q_h^\star)(s_h^k, a_h^k)$.
>
> 2. We then bound the cumulative weighted estimation errors by establishing a recursive relationship between consecutive steps (Lemma 3.2) and propagating it to the final step $H$ (Lemma 3.3).
>
> 3. Using Lemmas 3.2 and 3.3, we construct a recursive relationship for the cumulative weighted visitation counts $\sum_{h=1}^H \sum_{s,a} \Delta_h(s,a) N_h^{K+1}(s,a)$ across steps, which enables an inductive argument to derive a fine-grained upper bound and subsequently bound the expected regret via Lemma 3.1.
>
> These innovations form the core of our fine-grained analytical framework and can be applied to a broad class of model-free RL algorithms.

---

> ### Author Response · Authors · 2025-11-20
> **Response to Reviewer HMT1 (part three)**
>
> **Weakness 3 and Question 1b.2: High dependency on $H$ and potential improvement**
>
> The high dependence on $H$ arises because UCB-Hoeffding uses a relatively loose Hoeffding-type bonus, which is highly $ H$-dependent. It is also related to the recursive argument over steps to bound the cumulative weighted estimation errors $Q_h^k-Q_h^\star$ in the standard gap-dependent framework for model-free methods. The same $H$-dependency is also observed in the coarse-grained gap-dependent analysis [8] for UCB-Hoeffding.
>
>
> Improving the dependency on $H$ is possible in principle by incorporating variance-aware techniques and reference-advantage decompositions into our framework. Algorithms such as UCB-Bernstein [9], UCB-Advantage [10], Q-EarlySettled-LowCost [11], and Q-EarlySettled-LowCost [12], all employ variance-based bonuses, which are tighter than those of UCB-Hoeffding, and the latter three additionally use reference-advantage decomposition. As a result, they achieve better worst-case regret [9,10,11,12] and coarse-grained gap-dependent regret [12,13] than UCB-Hoeffding, and may potentially lead to improved fine-grained results with better $H$-dependency.
>
> However, incorporating variance-based bonuses and reference-advantage decomposition into our framework introduces nontrivial technical challenges. To achieve improved $H$-dependency, we need to handle different bonuses for different state--action--step triples. In contrast, for UCB-Hoeffding, different state--action--step triples with the same number of visits share the same bonus. This difference requires additional analysis to build the recursive structure we established for $\sum_{s,a} \Delta_h(s,a) N_h^{K+1}(s,a)$ at different steps in Lemma D.4. Moreover, [12,13] derives a coarse-grained gap-dependent regret for UCB-Advantage by introducing a surrogate reference function. Incorporating this surrogate reference function into our framework also requires additional nontrivial technical effort.
>
> **Question 2: Significance of ULCB-Hoeffding**
>
> To fully highlight the significance of incorporating ULCB-Hoeffding, we have rewritten the introduction in lines 55-60 and 100-110 of the revised manuscript, as summarized in the discussion below.
>
> The key novelty in the design of the AMB algorithm lies in two components: ULCB and multi-step bootstrapping. In particular, ULCB refers to leveraging both UCB and Lower Confidence Bound (LCB) techniques to select actions by maximizing the width of the confidence interval rather than maximizing the $Q$-estimates.
>
> UCB-Hoeffding cannot determine whether the optimal action has been learned, as it only maintains an upper-bound estimate of the optimal $Q$-function. Compared to UCB-Hoeffding, the action elimination technique (see line 14 of Algorithm 2 in our paper or line 23 in the original AMB algorithm), aided by the lower-bound Q-estimates in ULCB, enables the agent to determine, at each step and for each state, whether the optimal action has been learned. This not only improves the explainability of the algorithm but may also be of independent interest (see Section 1.2.2 in [14] for further discussion).
>
> In this paper, we incorporate ULCB-Hoeffding to demonstrate that, even without multi-step bootstrapping, which was incorrectly handled in the proof for the original AMB, our framework can be used to show that an algorithm employing the ULCB technique alone can achieve a fine-grained regret, thereby illustrating the generality of our framework. ULCB-Hoeffding can also be regarded as a UCB-based refinement of the original AMB algorithm.
>
> **Question 3: Comparison of the last terms in the theoretical results**
>
> The last term is not the bottleneck that determines the overall tightness of the result. In fact, as shown in Section F.4 of the revised version, the first two terms in the original Equation (10) already dominate and implicitly contain an $O(SAH^5)$ contribution. Therefore, the last term in Equation (2) does not affect the tightness compared to Refined AMB, and the seemingly tighter last term $SAH^2$ in the original Equation (10) does not result from a sharper analysis of Refined AMB; the $SAH^5$ dependence is already captured by the first two terms of Equation (10).
>
> Mathematically, as shown in Section F.4, the Refined AMB result in original Equation (10) can be also equivalently written as
> $$
> O\left(\sum_{h=1}^H \sum_{\Delta_h(s,a)>0} \frac{H^5 \log(SAT)}{\Delta_h(s,a)} + \frac{H^5 |Z_{\textnormal{mul}}| \log(SAT)}{\Delta_{\min}} + SAH^5 \right).$$

---

> ### Author Response · Authors · 2025-11-20
> **Response to Reviewer HMT1 (part five)**
>
> **Question 4: Minor issues**
>
> We have addressed all minor issues in our revised manuscript. For the second issue, we have rewritten the definitions of the recursively defined weights in lines 338–359 of the revised manuscript to make them clearer, and we now also include the intuition behind their definitions as follows.
>
> The recursive defined weights are extracted from our regret analysis when we try to control the cumulative weighted $Q$-estimation errors $\sum_{k = 1}^K \omega_{h}^{k} \left(Q_h^k - Q_h^\star\right)(s_h^{k}, a_h^{k})$ by recursions between steps. The detailed derivation is provided in Lemma D.3, and we also include a brief sketch below to illustrate the key idea.
>
> As provided in Lemma D.1, UCB-Hoeffding establishes the following relationship for the estimation error $(Q_h^k-Q_h^\star)(s_h^{k}, a_h^{k})$ of the $Q$-estimate $Q_h^k(s_h^k,a_h^k)$ with the cumulative bonus $\beta_{N_h^k(s_h^{k}, a_h^{k})}$ with high probability:
> $$(Q_h^k-Q_h^\star)(s_h^{k}, a_h^{k})\leq \sum_{i = 1}^{N_h^k(s_h^{k}, a_h^{k})}\eta_i^{N_h^k(s_h^{k}, a_h^{k})}\left(V_{h+1}^{k^i(s_h^{k}, a_h^{k},h)} (s_{h+1}^{k^i})- V_{h+1}^\star\right)(s_{h+1}^{k^i}) + \beta_{N_h^k(s_h^{k}, a_h^{k})}.$$
> This leads to the following relationship
> $$\sum_{k = 1}^K \omega_{h}^{k} \left(Q_h^k - Q_h^\star\right)(s_h^{k}, a_h^{k})\leq  \sum_{k = 1}^K \omega_{h}^{k} \sum_{i = 1}^{N_h^k(s_h^{k}, a_h^{k})}\eta_i^{N_h^k(s_h^{k}, a_h^{k})}(V_{h+1}^{k^i(s_h^{k}, a_h^{k},h)} - V_{h+1}^\star)(s_{h+1}^{k^i}) + \sum_{k = 1}^K \omega_{h}^{k}\left(\eta_0^{N_h^k}H + \beta_{N_h^k(s_h^{k}, a_h^{k})}\right).\quad (1)$$
> Focusing on the first term of the RHS of (1), we further have
> \begin{align*}
>     \sum_{k = 1}^K \omega_{h}^{k} \sum_{i = 1}^{N_h^k(s_h^{k}, a_h^{k})}\eta_i^{N_h^k(s_h^{k}, a_h^{k})}(V_{h+1}^{k^i(s_h^{k}, a_h^{k},h)} - V_{h+1}^\star)(s_{h+1}^{k^i(s_h^{k}, a_h^{k},h)})\\
>     = \sum_{k = 1}^K \omega_{h}^{k} \sum_{i = 1}^{N_h^k(s_h^{k}, a_h^{k})}\eta_i^{N_h^k(s_h^{k}, a_h^{k})}(V_{h+1}^{k^i(s_h^{k}, a_h^{k},h)} - V_{h+1}^\star)(s_{h+1}^{k^i}) \left(\sum_{k' = 1}^K \mathbb{I}[k^i(s_h^{k}, a_h^{k},h) = k']\right)\\
>     = \sum_{k' = 1}^K (V_{h+1}^{k'} - V_{h+1}^\star)(s_{h+1}^{k'})\left(\sum_{k = 1}^K\sum_{i = 1}^{N_h^k}\omega_{h}^{k}\eta_i^{N_h^k(s_h^{k}, a_h^{k})}\mathbb{I}[k^i(s_h^{k}, a_h^{k},h) = k'] \right) \nonumber\\
>     \leq \sum_{k' = 1}^K (Q_{h+1}^{k'} - Q_{h+1}^\star)(s_{h+1}^{k'},a_{h+1}^{k'})\left(\sum_{k = 1}^K\sum_{i = 1}^{N_h^k(s_h^{k}, a_h^{k})}\omega_{h}^{k}\eta_i^{N_h^k(s_h^{k}, a_h^{k})}\mathbb{I}[k^i(s_h^{k}, a_h^{k},h) = k'] \right)\\
>     = \sum_{k' = 1}^K \omega_{h + 1}^{k'}(h) (Q_{h+1}^{k'} - Q_{h+1}^\star)(s_{h+1}^{k'},a_{h+1}^{k'})
> \end{align*}
> with
> $$\omega_{h + 1}^{k'}(h) := \sum_{k = 1}^K\sum_{i = 1}^{N_h^k(s_h^{k}, a_h^{k})}\omega_{h}^{k}\eta_i^{N_h^k(s_h^{k}, a_h^{k})}\mathbb{I}[k^i(s_h^{k}, a_h^{k},h) = k'].$$
> The inequality here is because $Q_{h+1}^{k'}(s_{h+1}^{k'},a_{h+1}^{k'}) \geq V_{h+1}^{k'}(s_{h+1}^{k'})$ by the update rule in line 7 of Algorithm 1 and $Q_{h+1}^{\star}(s_{h+1}^{k'},a_{h+1}^{k'}) \leq V_{h+1}^{\star}(s_{h+1}^{k'})$ by Bellman optimality equation. Thus, we have
> $$\sum_{k = 1}^K \omega_{h}^{k} \left(Q_h^k - Q_h^\star\right)(s_h^{k}, a_h^{k})\leq  \sum_{k' = 1}^K \omega_{h + 1}^{k'}(h) (Q_{h+1}^{k'} - Q_{h+1}^\star)(s_{h+1}^{k'},a_{h+1}^{k'}) + \sum_{k = 1}^K \omega_{h}^{k}\left(\eta_0^{N_h^k}H + \beta_{N_h^k(s_h^{k}, a_h^{k})}\right).\quad (2)$$
> Equation (2) establishes a recursive relationship between the cumulative weighted estimation errors at steps $h$ and $h+1$. By the same argument, starting from step $h+1$, we can derive a similar relationship for any pair of consecutive steps $h'$ and $h'+1$, up to step $H$. The recursively defined weight $\omega_{h'+1}^k(h)$ explicitly characterizes the weight of the term $(Q_{h'+1}^{k} - Q_{h'+1}^\star)(s_{h'+1}^{k}, a_{h'+1}^{k})$ in the recursive propagation from step $h'$ to $h'+1$.

---

> ### Author Response · Authors · 2025-11-20
> **Response to Reviewer HMT1 (part six)**
>
> **Reference:**
>
> [1] He, Jiafan, Dongruo Zhou, and Quanquan Gu. "Logarithmic regret for reinforcement learning with linear function approximation." International Conference on Machine Learning. PMLR, 2021.
>
> [2] Papini, Matteo, et al. "Reinforcement learning in linear mdps: Constant regret and representation selection." Advances in Neural Information Processing Systems 34 (2021): 16371-16383.
>
> [3] Zhang, Weitong, et al. "Achieving constant regret in linear Markov decision processes." Advances in Neural Information Processing Systems 37 (2024): 130694-130738.
>
> [4] Chen, Shulun, et al. "Sharp Gap-Dependent Variance-Aware Regret Bounds for Tabular MDPs." arXiv preprint arXiv:2506.06521 (2025).
>
> [5] Zhang, Zihan, et al. "Settling the sample complexity of online reinforcement learning." The Thirty Seventh Annual Conference on Learning Theory. PMLR, 2024.
>
> [6] Simchowitz, Max, and Kevin G. Jamieson. "Non-asymptotic gap-dependent regret bounds for tabular mdps." Advances in Neural Information Processing Systems 32 (2019).
>
> [7] Dann, Christoph, et al. "Beyond value-function gaps: Improved instance-dependent regret bounds for episodic reinforcement learning." Advances in Neural Information Processing Systems 34 (2021): 1-12.
>
> [8] Yang, Kunhe, Lin Yang, and Simon Du. "Q-learning with logarithmic regret." International Conference on Artificial Intelligence and Statistics. PMLR, 2021.
>
> [9] Jin, Chi, et al. "Is Q-learning provably efficient?." Advances in neural information processing systems 31 (2018).
>
> [10] Zhang, Zihan, Yuan Zhou, and Xiangyang Ji. "Almost optimal model-free reinforcement learning via reference-advantage decomposition." Advances in Neural Information Processing Systems 33 (2020): 15198-15207.
>
> [11] Li, Gen, et al. "Breaking the sample complexity barrier to regret-optimal model-free reinforcement learning." Advances in Neural Information Processing Systems 34 (2021): 17762-17776.
>
> [12] Zhang, Haochen, Zhong Zheng, and Lingzhou Xue. "Regret-Optimal Q-Learning with Low Cost for Single-Agent and Federated Reinforcement Learning." arXiv preprint arXiv:2506.04626 (2025).
>
> [13] Zheng, Zhong, Haochen Zhang, and Lingzhou Xue. "Gap-Dependent Bounds for Q-Learning using Reference-Advantage Decomposition." The Thirteenth International Conference on Learning Representations.
>
> [14] Xu, Haike, Tengyu Ma, and Simon Du. "Fine-grained gap-dependent bounds for tabular mdps via adaptive multi-step bootstrap." Conference on Learning Theory. PMLR, 2021.

---

> ### Author Response · Authors · 2025-11-27
> **Following up on the rebuttal**
>
> Dear Reviewer HMT1,
>
> We hope that our responses and revised manuscript have adequately addressed your concerns regarding (1) the extension to function approximation, (2) the comparison with model-based works, (3) the significance of ULCB-Hoeffding, and (4) the comparison of the last terms in the results of UCB-Hoeffding and AMB. If you have additional questions or concerns, please let us know, and we would be happy to provide further clarification. We truly appreciate your feedback and look forward to hearing from you soon.
>
> Best regards,
>
> Authors of Submission 11791

---

### Meta-Review · Area_Chair_VPdY · 2026-01-06

**Summary:**

The paper provides fine-grained gap-dependent bounds for tabular Q-learning.  They propose several algorithms (or refine existing algorithms) to achieve this: UCB-Hoeffding,  ULCB-Hoeffding, Refined AMB.  A previous work AMB (Xu et al., 2021) has achieved a similar guarantee, though relying more complex multi-step bootstrapping strategy.  The first two proposed algorithms simplify the algorithms, aligning more with the standard Q-learning, while Refined AMB is based on a minor tweak over AMB, but with corrected theoretical analysis.

I believe the main contribution lies in showing that the original UCB-Hoeffding algorithm achieves fine-grained gap-dependent bound, without any modification. The analytical framework can also be used to analyze other algorithms like ULCB-Hoeffding and AMB.

The authors also pointed out there are two technical flaws in the AMB work.  I looked into the claims, and agree that their Q-update rule has to be corrected to a similar form as (Jin et al., 2018); however, the second claim that they misuse Azuma's inequality feels more like a minor clarity issue in their proofs (the conditioning in their (4.2) is not written clearly), but not really a "mistake".  Overall, the two flaws of Xu et al. can actually be both corrected easily and straightforwardly by any experienced reader, and do not really hurt their position as the first to achieve fine-grained gap bounds in model-free learning.   Therefore, I suggest the authors to weaken their claim of being the first to achieve model-free fine-grained bound, and emphasize more on being the first to achieve similar bound with a *simpler and more standard* algorithm.  I believe a more common way to handle this situation is to simply contact Xu et al. (2021), letting them know the minor errors and urging them to fix those in their paper.

Please add the comparison with model-based fine-grained bound in the rebuttal to the paper.

**Reviewer Concerns:**

The reviewers raised the following concerns:
- Loose dependence on H
- Missing comparison with model-based approaches

These are minor concerns compared to the contribution.  The authors are encouraged to incorporate a technical comparison with model-based approach in the paper.

**Reviewer Scores:**

The original scores are 8864, where the score-4 reviewer's comments are mostly some clarification questions.  After the rebuttal, that reviewer seems to be satisfied, and might increases their score to 6.

---

### Decision · Program_Chairs · 2026-01-26

Accept (Poster)